

# Effects of precipitation seasonality, vegetation cycle, and irrigation on enhanced weathering

Giuseppe Cipolla[1], Salvatore Calabrese[2], Amilcare Porporato[3], and Leonardo Noto[1]

[1]Dipartimento di Ingegneria, Università degli Studi di Palermo (Palermo, Italy)
[2]Department of Biological and Agricultural Engineering, Texas A&M University (College Station, TX, USA)
[3]Department of Civil and Environmental Engineering, Princeton University (Princeton, NJ, USA)

**Correspondence:** Giuseppe Cipolla (giuseppe.cipolla04@unipa.it)

**Abstract.** Enhanced Weathering (EW) is a promising strategy for carbon sequestration, but several open questions remain regarding the actual rates of dissolution in conditions of natural hydroclimatic variability in comparison to laboratory experiments. In this context, models play a pivotal role, as they allow exploring and predicting EW dynamics under different environmental conditions. Here a comprehensive hydro-biogeochemical model has been applied to four cropland case studies

(i.e., Sicily and the Padan plain in Italy and California and Iowa in the USA) characterized by different rainfall seasonality, vegetation (i.e., wheat for Sicily and California and corn for Padan plain and Iowa), and soil type to explore their influence on dissolution rates. The results reveal that rainfall seasonality, and irrigation when applied, are crucial in determining EW and carbon sequestration dynamics, given their effect on hydrological fluxes, soil pH, and weathering rate. The carbon sequestration rate was found to be strongly affected also by the pre-EW soil pH, which is one of the main factors controlling soil pH

before the olivine amendment. In the analyzed case studies, Iowa and Sicily sequester the greatest amount of $CO_2$ (4.20 and 0.62 kg $ha^{-1}$ $y^{-1}$, respectively), as compared to California and the Padan plain (2.21 and 0.39 kg $ha^{-1}$ $y^{-1}$, respectively). These low carbon sequestration values suggest that an in-depth analysis at the global scale is required to assess EW efficacy for carbon sequestration.

## 1 Introduction

Enhanced Weathering (EW) is considered one of the most promising Carbon Dioxide Removal (CDR) technologies to mitigate climate change (Ramos et al., 2022; Gomez-Casanovas et al., 2021; Russell et al., 2012). Many studies, mainly based on laboratory experiments and modeling approaches, demonstrated a great carbon sequestration potential when applied over large scales (Beerling et al., 2020; Cipolla et al., 2021a, b; Renforth et al., 2015; Goll et al., 2021; Lewis et al., 2021). EW consists of enhancing naturally occurring weathering reaction rates by adding highly reactive minerals, such as silicates, to the soil

(Hartmann et al., 2013; Lal et al., 2021). Reacting with $CO_2$, silicate grains allow for the formation of dissolved bicarbonate ($HCO_3^-$) and carbonates ($CO_3^{2-}$), which are then leached out of the soil, transported by groundwater, and eventually reach the oceans or precipitate as carbonates. Many studies about EW discuss using forsterite (i.e., $Mg_2SiO_4$), always referred as to olivine in EW applications (Köhler et al., 2010; ten Berge et al., 2012), since it can be extracted from volcanic rocks (i.e.,



basalt or gabbro), widely distributed across the globe, and it is characterized by relatively fast dissolution rates if compared to
other silicate minerals, such as albite and orthoclase (Hartmann et al., 2013).

Despite the increasing attention to EW, a knowledge gap exists about the actual rates and efficiency of EW applications for
carbon sequestration in actual field conditions. The technological readiness of EW is currently limited to pilot-scale experi-
ments (Asibor et al., 2021) and quote the large uncertainties in scaling up laboratory results to field scale (White and Brantley,
2003). The most common elements of uncertainty regard the formation of secondary minerals, the alteration of soil hydro-
logical properties due to silicate amendments (i.e., the formation of the preferential flow path in soil pores), and the effect of
the biotic complements (e.g., fungi and bacteria that, depending on soil pH, may accelerate or slow down the organic matter
decomposition and in turn silicates dissolution rates (Vicca et al., 2022).

To begin to address these uncertainties, several experimental approaches have been carried out to characterize olivine or any
other silicate mineral used for EW, such as basalt or wollastonite, dissolution dynamics. These are mainly based on laboratory
experiments conducted on single mineral particles (Hellmann and Tisserand, 2006; Köhler et al., 2003) or pot and mesocosm
experiments (Renforth et al., 2015; ten Berge et al., 2012; Amann et al., 2020; Kelland et al., 2020; Dietzen et al., 2018) with
conditions that are more similar to the field. Amann et al. (2020) realized a mesocosm experiment adding olivine to agricultural
loamy sand, extracted from a field in Belgium. The experiment considered the most common crops in that area, i.e., wheat and
barley, and focused on both well-watered and water-stressed conditions, secondary mineral formation, cation exchange, and the
formation of preferential flow paths. The effects of olivine amendment were assessed by monitoring the chemical characteristics
of soil water, particularly looking at the pH, magnesium, and silicates concentration as well as Dissolved Inorganic Carbon
(DIC). The achieved weathering rates were on the order of $10^{-13}$ mol $m^{-2}\ s^{-1}$ corresponding to carbon sequestration rates of
23 and 49 $kg_{CO_2}\ ha^{-1}\ y^{-1}$. Weathering rates in actual field conditions are much lower than those derived in laboratory setups
(Palandri and Kharaka, 2004; Rimstidt, 2015) and soil column experiments under controlled conditions (Renforth et al., 2015;
45  ten Berge et al., 2012), underlying the discrepancy between lab and field conditions, which is generally due to an ensemble of
intrinsic and extrinsic factors (White and Brantley, 2003), that in the laboratory are not easily reproducible.

These uncertainties in weathering rates inevitably affect the evaluation of a realistic carbon sequestration rate of EW. Mod-
eling approaches may be useful to explore the role of soil processes that cannot be easily replicated in laboratory, such as the
complexity of cation exchanges or hydroclimatic fluctuations (Cipolla et al., 2021a). Vicca et al. (2022) highlights that only
50  a few models connecting EW dynamics with plants, climate, and biogeochemical processes exist. In the reactive transport
models summarized in Taylor et al. (2017), depending on their degree of complexity, not all these processes are considered
at the same time and, some of them, are sometimes too simplified. Lewis et al. (2021) presents an application of a 1-D reac-
tive transport model to simulate EW carbon sequestration potential and the release of base cations (i.e., $Mg^{2+}$, $Ca^{2+}$, $K^+$ and
$Na^+$) that can be beneficial for plants. Like many other reactive transport models, they assume a constant average infiltration
55  rate, thus excluding rainfall variability effects on EW. In their study, the application of 50 t $ha^{-1}$ of basalt leads to a carbon
sequestration potential between 1.3 and 8.5 $t_{CO_2}\ ha^{-1}$ after 15 years of amendment. These values, however, are much higher
than the one resulting from the experiment of Amann et al. (2020), which represents a situation closer to the field environment.
By introducing stochasticity in rainfall and connecting ecohydrological with biogeochemical processes, the model presented





in Cipolla et al. (2021a) leads to carbon sequestration rates of the same order of magnitude as those in Amann et al. (2020),
suggesting that the model estimates approach a very similar condition to what happens in field.

In this study, the model presented in Cipolla et al. (2021a) is applied to four hypothetical case studies (i.e., agricultural applications of EW), two of them in Italy (i.e., Sicily, in the south, and the Padan plain, in the north) and two in the USA (i.e., California, in the south-west, and Iowa, in the north-central area). Many of the model components are characterized on the base of measurements (i.e., pH and cation exchange), and achieved weathering rates are then compared with those of Amann et al. (2020), which is currently one of the experiments closest to field conditions. Extending the work of Cipolla et al. (2021b), where the influence of statistically stationary precipitation with different MAPs on EW is analyzed, in this study we focus on the effect of the seasonal variation of the meteorological forcing, irrigation, soil type and crop cycle on EW dynamics. These four regions are characterized by very similar MAP values but much different rainfall seasonality. They also present similar soils and different crop types. The impact of different vegetation cycles is also considered in this study, given that crops contribute to the water balance through evapotranspiration losses differently during the year, depending on the crop stage, and in the carbon balance since they release organic matter (i.e., litter (D'Odorico et al., 2003; Porporato et al., 2003) and root exudation (Shen et al., 2020; Wang et al., 2015)) to the soil. Furthermore, differences among the selected places in terms of cation exchange capacity and background weathering flux have been taken into account, given that these factors strongly affect soil pH before olivine amendment (Cipolla et al., 2021a, b). The main goal of this paper is therefore to understand which of the considered factors (i.e., rainfall seasonality, soil type and composition and crop cycle) is mostly connected to changes in EW time dynamics. We also explore the role of irrigation on EW, only when it is required basing on crop type and climatic condition. Moreover, analyses allow one to define the most suitable combined characteristics of climate, soil and vegetation for EW, while providing important hypotheses to be tested experimentally.

The manuscript is structured as follows: Section 2 presents a description of the adopted methodology and the four places under study, along with the estimation of the parameters related to the above-mentioned factors. Here, an in-depth discussion of the differences in terms of rainfall seasonality, soil and vegetation types is reported. In Section 3, EW dynamics at the four considered sites are presented and discussed, highlighting the role of each of the considered factors on EW. Lastly, Section 4 is devoted to some final discussions about the potentials and limitations of the present study with a view to further applications and provides some general conclusions.

## 2 Materials and methods

### 2.1 Methodology

The analyses presented in this manuscript employ a mathematical model of EW dynamics developed and presented in Cipolla et al. (2021a), to which we refer for details, linking ecohydrological and biogeochemical processes (Cipolla et al., 2021a, b) and is externally forced by rainfall fluctuations, which can be in the form of an observed time-series or, for long-term EW dynamics, a stochastic representation. The model is composed of four closely related components. Starting from the organic matter decomposition, which provides one of the sources of $CO_2$ in the system, the model accounts for the mass balance of





$CO_2$ in the gas and dissolved phases, with particular attention to the released $H^+$, bicarbonate (i.e., $HCO_3^-$) and carbonate (i.e., $CO_3^{2-}$) ions in soil water. A specific component considers the balance of soil water cations released by olivine dissolution (i.e., silicates and magnesium) as well as other base cations (i.e., $Mg^{2+}$, $Ca^{2+}$, $K^+$ and $Na^+$) along with their influence on pH.

The cation exchange capacity (CEC) accounts for the process between the dissolved cations and the solid matrix of the soil, composed of organic and inorganic colloids. The model fully accounts for ecohydrological interactions driven by precipitation variability, including the evapotranspiration effect on nutrient cation uptake from vegetation and, in turn, its acidifying effect on the soil, as well as the impacts of leaching rates on ion removal and hence carbon sequestration. All these aspects are translated into an explicit system of eight mass balance differential equations and an implicit system composed of twenty-two algebraic

equations (Cipolla et al., 2021a).

Concerning four specific regions, the case studies presented in the following sections aim to understand which of the factors among rainfall seasonality, soil type and composition, and crop cycle exerts the greatest influence on EW dynamics, in addition to the effects of any irrigation intervention. A flow chart of the methodology employed in this manuscript is presented in Figure 1. For the four case studies, rainfall seasonality was characterized by the temporal variability of the average depth and

frequency of rainfall events. Soil type and vegetation parameters were characterized by using the spatial distribution of some related variables (i.e., the sand, silt and clay content, organic matter, cation exchange capacity, soil pH, crop type and cycle); see Section 2.2. To single out the most important factors on EW dynamics, a combination of numerical experiments was designed to consider different conditions of soil moisture, pH and weathering rates (see Table 1). First, some possible combinations of rainfall seasonality, soil type and crop cycle are taken into account, to explore the role of each of them on EW dynamics.

The last four simulations presented in Section 3.4 are related to the analysis of EW dynamics at the four selected locations, selecting the most frequent combination of soil type and crop cycle with rainfall seasonality.

## 2.2 Study areas and data

The case studies refer to two different locations in Italy (i.e., Sicily, in the south, and the Padan plain, in the north) and in the USA (i.e., California, in the south-west, and Iowa, in the north-central area). Figure 2 shows the cropland areas suitable for po-

tential olivine amendment as indicated by Copernicus land cover maps (Buchhorn et al., 2020). For these areas, the most common crops according to the USDA crop production maps ($https://ipad.fas.usda.gov/rssiws/al/global_cropprod.aspx$), namely the wheat for Sicily and California and the corn for Padan plain and Iowa, are considered. As in Cipolla et al. (2021b), all simulations are related to a unit ground area of homogeneous soil, vegetation and rainfall characteristics, delimited by the active root zone depth of the involved crop.

### 2.2.1 Rainfall seasonality

Following Cipolla et al. (2021a), rainfall is modeled as a marked Poisson process, where the average depth $\alpha$ and the frequency of events $\lambda$ are defined by the mean rainfall depth and the event frequency, respectively, whose product (if expressed in per year) provides the MAP (Rodriguez-Iturbe et al., 1999). The former, $\alpha$, represents the average rainfall depth of occurred daily rainy events, while $\lambda$ is the ratio between the number of rainy days and the total number of days. These parameters have



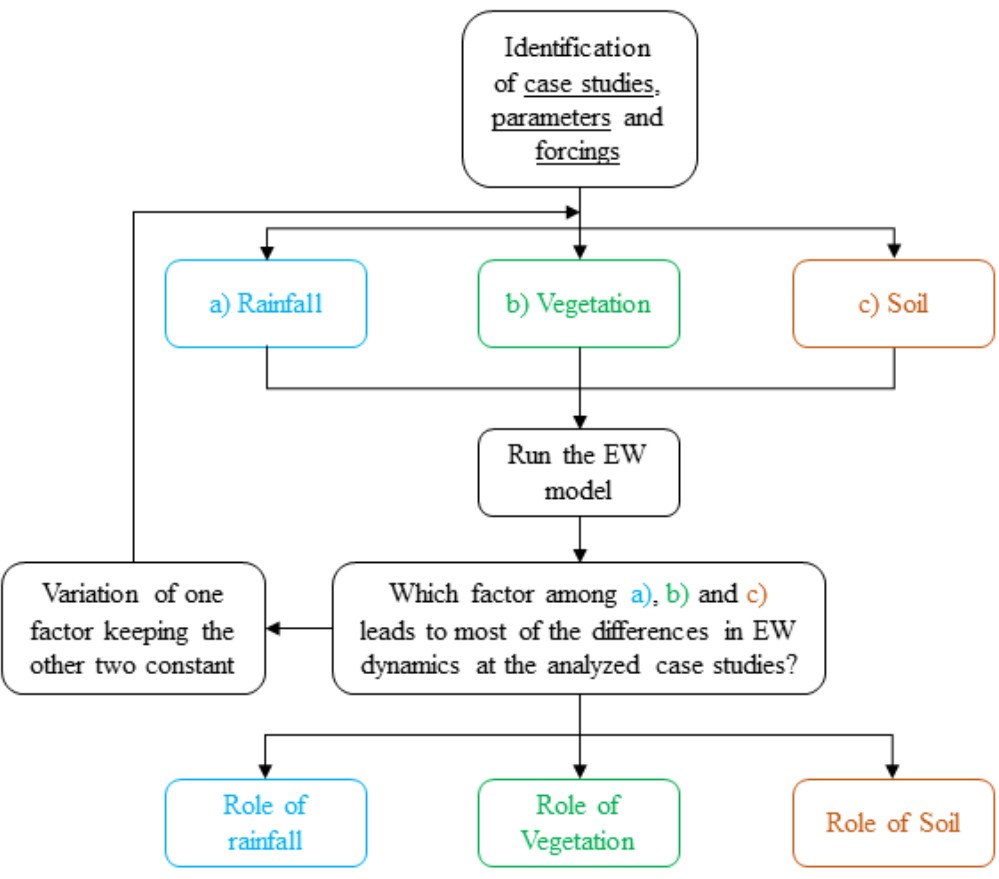

**Figure 1.** Flow chart of the employed procedure for the case study of the EW model described in Cipolla et al. (2021a).

been estimated at a monthly scale from daily rainfall data recorded from 2000 to 2020 by the SIAS (Servizio Informativo Agrometeorologico Siciliano) rain gauge network for Sicily, the Eraclito dataset of ARPA Emilia Romagna for the Padan plain and the USGS (United States Geological Survey) for the Iowa and California states. Given the point scale application of the EW model, data collected by a single rain gauge representative of the area under study have been considered for the analyses. Rainfall parameters $\alpha$ and $\lambda$ have been obtained on a monthly scale, providing their seasonal variation within the year (see

Figure 3). In particular, Sicily (panel a) and California (panel b) are characterized by a similar pattern of $\alpha$ and $\lambda$, since they both reach a minimum during summer months, which are usually rainless, and a peak in winter months, especially between November and January (Cipolla et al., 2020). Regarding the Padan plain (panel c) and Iowa (panel d), rainfall seasonality is less evident than in the other locations, especially in the case of the Padan plain. For the latter, the two parameters are two months





**Table 1.** Simulation outline adopted in this study. Here all the explored combinations of the analyzed factors are reported.

| | Simulations | Crop | Soil | Location | Output (Heatmaps) |
|---|---|---|---|---|---|
| **Rainfall seasonality (Section 3.1)** | 1 | Corn | Clay loam | Sicily (California) | Padan plain / Sicily and Iowa / California |
| **Rainfall seasonality (Section 3.1)** | 2 | Corn | Clay loam | Padan plain (Iowa) | Padan plain / Sicily and Iowa / California |
| **Rainfall seasonality (Section 3.1)** | 3 | Corn | Silty clay loam | Sicily (California) | Padan plain / Sicily and Iowa / California |
| **Rainfall seasonality (Section 3.1)** | 4 | Corn | Silty clay loam | Padan plain (Iowa) | Padan plain / Sicily and Iowa / California |
| **Rainfall seasonality (Section 3.1)** | 5 | Wheat | Silty clay loam | Sicily (California) | Padan plain / Sicily and Iowa / California |
| **Rainfall seasonality (Section 3.1)** | 6 | Wheat | Silty clay loam | Padan plain (Iowa) | Padan plain / Sicily and Iowa / California |
| **Rainfall seasonality (Section 3.1)** | 7 | Wheat | Clay loam | Sicily (California) | Padan plain / Sicily and Iowa / California |
| **Rainfall seasonality (Section 3.1)** | 8 | Wheat | Clay loam | Padan plain (Iowa) | Padan plain / Sicily and Iowa / California |
| | | Location | Crop | Soil | Output (Heatmaps) |
| **Soil type (Section 3.2)** | 9 | Sicily | Wheat | Clay loam | Clay loam / Silty clay loam |
| **Soil type (Section 3.2)** | 10 | Sicily | Wheat | Silty clay loam | Clay loam / Silty clay loam |
| **Soil type (Section 3.2)** | 11 | Padan plain | Corn | Clay loam | Clay loam / Silty clay loam |
| **Soil type (Section 3.2)** | 12 | Padan plain | Corn | Silty clay loam | Clay loam / Silty clay loam |
| **Soil type (Section 3.2)** | 13 | Iowa | Corn | Clay loam | Clay loam / Silty clay loam |
| **Soil type (Section 3.2)** | 14 | Iowa | Corn | Silty clay loam | Clay loam / Silty clay loam |
| **Soil type (Section 3.2)** | 15 | California | Wheat | Clay loam | Clay loam / Silty clay loam |
| **Soil type (Section 3.2)** | 16 | California | Wheat | Silty clay loam | Clay loam / Silty clay loam |
| | | Location | Soil | Crop | Output (Heatmaps) |
| **Crop cycle (Section 3.3)** | 17 | Sicily | Clay loam | Wheat | Corn / Wheat |
| **Crop cycle (Section 3.3)** | 18 | Sicily | Clay loam | Corn | Corn / Wheat |
| **Crop cycle (Section 3.3)** | 19 | Padan plain | Silty clay loam | Wheat | Corn / Wheat |
| **Crop cycle (Section 3.3)** | 20 | Padan plain | Silty clay loam | Corn | Corn / Wheat |
| **Crop cycle (Section 3.3)** | 21 | Iowa | Silty clay loam | Wheat | Corn / Wheat |
| **Crop cycle (Section 3.3)** | 22 | Iowa | Silty clay loam | Corn | Corn / Wheat |
| **Crop cycle (Section 3.3)** | 23 | California | Clay loam | Wheat | Corn / Wheat |
| **Crop cycle (Section 3.3)** | 24 | California | Clay loam | Corn | Corn / Wheat |
| | | Location | Soil | Crop | Output (Heatmaps) |
| **Most frequent scenarios (Section 3.4)** | 25 | Sicily | Clay loam | Wheat | Output (heatmaps) |
| **Most frequent scenarios (Section 3.4)** | 26 | Padan plain | Silty clay loam | Corn | Output (heatmaps) |
| **Most frequent scenarios (Section 3.4)** | 27 | Iowa | Silty clay loam | Corn | Output (heatmaps) |
| **Most frequent scenarios (Section 3.4)** | 28 | California | Clay loam | Wheat | Output (heatmaps) |

out of phase, while for Iowa the highest values of both $\alpha$ and $\lambda$ occur during the spring and summer months. These different

rainfall seasonality regimes have relevant effects on EW dynamics, especially when considered along with crop phenology, especially given the modest MAP differences between sites (i.e., MAP about 600 mm for Sicily and California and 750 mm for the Padan plain and Iowa).

### 2.2.2 Soil type and composition

The soil type for the four sites under study was derived using the USDA soil texture triangle, which requires as input the silt,

sand and clay fractions. These in turn were extracted from the Soil Grids project maps (Hengl et al., 2017) considering the areas covered by croplands and reaching a depth equal to the active root zone of 40 and 60 cm for the corn and wheat, respectively (Fan et al., 2016). The most frequent soil types for the four considered sites were considered. These are the clay loam soil for Sicily and California and the silty clay loam soil for the Iowa state and the Padan plain. Following previous studies (Clapp





**Figure 2.** Locations of the areas possibly devoted to the olivine amendment in the places under study within the Copernicus land cover maps. In particular, cropland areas in the panels of Padan plain and Iowa are related to corn, while those in the panels of Sicily and California to wheat. The central box showing the locations of all the four sites contains the OpenStreetMap as background (© OpenStreetMap contributors. Distributed under the Open Data Commons Open Database License (ODbL) v1.0. ).

and Hornberger, 1978; Laio et al., 2001), the main properties of these two soil types, such as porosity, the soil moisture values
characteristic of the soil water retention curve, the hydraulic conductivity at saturation and the bulk density have been extracted
and presented in Table 2. For the calculation of the soil moisture at the hygroscopic point, the wilting point and at incipient
stress, we used the soil water retention curve defined by Clapp and Hornberger (1978), supposing soil water potentials of -10
MPa, -3 MPa and -0.03 MPa respectively, as done in Laio et al. (2001).

    Regarding the soil organic carbon content, the Global Soil Organic Carbon map (GSOCmap) of the Food and Agriculture
Organization of the United Nations (FAO) (http://54.229.242.119/GSOCmap/) (FAO, 2020), representing a global spatial dis-
tribution of the organic carbon content of the first 30 cm soil layer expressed in ton $ha^{-1}$, was used. This could be assimilated
as the initial carbon concentration in the litter and humus pools ($C_0$) that, expressed in kg $m^{-3}$, is given as input to the carbon
module of the model. Following D'Odorico et al. (2003), the carbon concentration in the biomass pool ($C_b$), which regulates





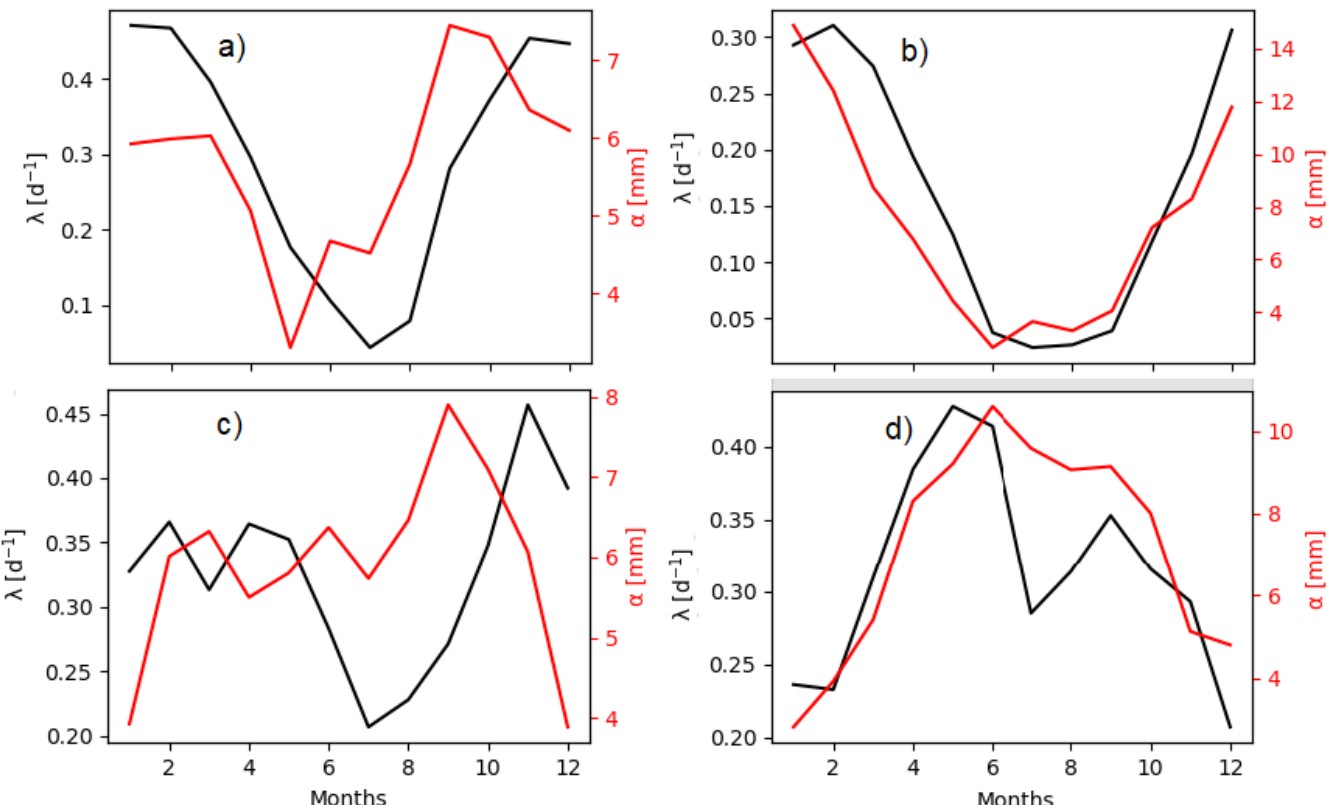

**Figure 3.** Values of $\alpha$ and $\lambda$ rainfall parameters from January (i.e., month indicated with 1) to December (i.e., month indicated with 12) for a) Sicily, b) California, c) Padan plain and d) Iowa.

**Table 2.** Properties of the clay loam and silty clay loam soils used in the model.

|  |  | Clay loam | Silty clay loam |
|---|---|---|---|
| Soil porosity | n | 0.476 | 0.477 |
| Soil moisture at the hygroscopic point | $s_h$ | 0.394 | 0.319 |
| Soil moisture at wilting point | $s_w$ | 0.453 | 0.373 |
| Soil moisture at incipient stress | $s^*$ | 0.64 | 0.56 |
| Soil moisture at field capacity | $s_{fc}$ | 0.821 | 0.75 |
| Saturation hydraulic conductivity | Ks | 0.212 | 0.1469 |
| Pore size distribution index | b | 8.52 | 7.75 |
| Bulk density of soil | $\rho_b$ | 1450 | 1500 |

the biochemical decomposition rate of organic matter, is considered constant and is defined as about 1% of the above-defined

carbon input. Table 3 lists the two considered organic carbon contents for the four sites under study.





**Table 3.** Initial organic carbon content in the litter and humus pools ($C_0$) and the biomass pool ($C_b$) for the four sites under study.

|  | **Sicily** | **Padan plain** | **California** | **Iowa** |
|---|---|---|---|---|
| $C_0[kg\ m^{-3}]$ | 13.19 | 18.19 | 19.74 | 11.58 |
| $C_b[kg\ m^{-3}]$ | 131.9 | 181.9 | 197.4 | 115.8 |

Due to the key role of the cation exchanges between the dissolved ions and soil colloids on pH, the total cation exchange capacity (CEC) per unit mass of the soil has been set based on site-specific values reported in the CEC maps of Ballabio et al. (2019) for the two sites in Italy and by the United States Department of Agriculture (USDA) data for Iowa and California (Hempel et al., 2014; Libohova et al., 2014), resulting in a total CEC equal to 50 cmol $kg^{-1}$ for Sicily and the Padan plain and 160 35 and 25 cmol $kg^{-1}$ for Iowa and California.

The background weathering flux represents the losses of protons due to the chemical weathering process of all the minerals that are naturally present in the soil. In general, locations with the same soil type, vegetation and climate characteristics may lead to different weathering rates according to the minerals within the soil that, with different rates, consume H+ ions available in soil water. This weathering flux can be estimated on the base of the mineral composition of the soil and the type of the 165 existing bedrock. This last information was extracted from the lithological map presented in Hartmann and Moosdorf (2012), representing the nature of bedrock on a global scale. The exploration of the lithological information in correspondence with the cropland areas for the four considered sites, highlights that Sicily and the Padan plain are prevalently characterized by carbonate sedimentary rocks (e.g., calcite), while the other two sites in the USA mainly present siliciclastic sedimentary rocks (e.g., quartz). This results in a very different background weathering rate between places on the two continents since carbonate 170 rocks have a much faster dissolution rate constant than siliciclastic rocks. Indeed, considering Lasaga (1984) and 44 (1979), at 25°C and pH equal to 5, calcite and quartz minerals are characterized by a dissolution rate constant of about $10^{-5}$ and $10^{-13}$ mol $m^{-2}\ s^{-1}$, respectively. This factor has a relevant effect on olivine dissolution dynamics since it affects soil pH dynamics before olivine amendment. However, soil pH may depend on other factors that are not considered in the EW model (i.e., the presence of fertilizers, the action of microbes, fungi and bacteria or the action of other minerals that may release or take up H$^+$ 175 ions).

The dissolution rate constants incorporate these previous aspects and were set to achieve steady-state soil pH values typical of the four sites under study, as reported in Ballabio et al. (2019) for Sicily and the Padan plain (i.e., a value ranging from 7.2 to 8.8) and the USDA (Hempel et al., 2014; Libohova et al., 2014) for California (i.e., a value around 7) and Iowa (i.e., a value around 6). As a result, three values of background weathering flux, a faster one for the case studies in Italy (dissolution rate 180 constant of $10^{-5}$ mol $m^{-2}\ s^{-1}$), a lower value for California (dissolution rate constant set to $10^{-6}$ mol $m^{-2}\ s^{-1}$) and an even slower one for Iowa (dissolution rate constant set to $10^{-8}$ mol $m^{-2}\ s^{-1}$) were obtained.





### 2.2.3 Crop cycle

To investigate the role of the crop cycle on the EW dynamics, the monthly variation of the crop coefficient (Kc) was considered. For each development stage, a single crop coefficient per crop type and the climatic area was obtained following FAO
guidelines (solid lines in Figure 4). Regarding the considered crops, the wheat plant date is in November for Sicily, December for California and October for Padan plain and Iowa. The total crop stage lasts 240, 180 and 335 days respectively. Corn is usually planted in April and the total crop stage is 170 days, in both the Padan plain and Iowa. Regarding Sicily and California, the corn cycle is the same as that of Padan plain and Iowa, except for the end-of-season stage that lasts 30 days instead of 50 and, therefore, the total cropping season lasts for 150 days.
The effects of the seasonal pattern of the crop coefficient on transpiration losses were computed as,

$$
T(s) = \begin{cases} 0 & 0 \leq s < s_w \\ \frac{s-s_w}{s^*-s_w} f(t) ET_0(t) & s_w \leq s < s^* \\ f(t) ET_0(t) & s^* \leq s \leq 1 \end{cases}
\tag{1}
$$

where $ET_0(t)$ is the reference evapotranspiration, which is computed using the Penman-Monteith equation for a reference crop. The annual pattern of this variable, for the four selected locations, has been extracted from the high-resolution database provided by Singer et al. (2021). The term f of equation (2) represents the rate of canopy cover over the entire study area, which
is complementary to the area covered by bare soil. We here assume it as the ratio between the crop coefficient, at a certain time, and the maximum crop coefficient, which is reached at the mid-season stage. The bare soil evaporation is evaluated as,

$$
E(s) = \begin{cases} 0 & 0 \leq s < s_h \\ \frac{s-s_h}{s_{fc}-s_h} (1-f(t)) ET_0(t) & s_h \leq s < s_{fc} \\ (1-f(t)) ET_0(t) & s_{fc} \leq s \leq 1 \end{cases}
\tag{2}
$$

The total evapotranspiration water losses are therefore computed as the sum between bare soil evaporation and crop transpiration. The crop coefficient variation across the year, represented in Figure 4, is determined as,

$$
Kc(t) = \begin{cases} K_{c,ini} & t \leq f_i \\ \frac{K_{c,max}-K_{c,ini}}{f_d-f_{ini}} (t-f_i) + K_{c,ini} & f_i < t \leq f_d \\ K_{c,max} & f_d < t \leq f_{ms} \\ \frac{K_{c,ls}-K_{c,max}}{f_{ls}-f_{ms}} (t-f_{ms}) + K_{c,max} & f_{ms} < t \leq f_{ls} \\ K_{c,ls} & t = f_{ls} \end{cases}
\tag{3}
$$





where $f_i$, $f_d$, $f_{ms}$ and $f_{ls}$ are the period related to the different crop stages (i.e., initial, development, maturity and late seasons, respectively), while $K_{c,ini}$, $K_{c,max}$ and $K_{c,ls}$ are the corresponding crop coefficient values, according to FAO guidelines.

Regarding the added carbon to the soil from vegetation (ADD), we considered the byproducts released as root exudates and the input of litter after crop harvest, which depends on the adopted agricultural practice. Since root exudation products are connected to the vegetation cycle, because they consist of carbon-based compounds deriving from plant metabolic activity that are released from living roots (Shen et al., 2020), their contribution to the carbon input to the soil can be modeled as a slight linear increase from the background ADD (i.e., the starting point of the ADD axis in Figure 4) to a minimum ADD value during the initial growing stage (i.e., from November, $1^{st}$ for 30 days for wheat in Sicily, from December, $1^{st}$ for 20 days for wheat in California, from October $1^{st}$ for 160 days for wheat in Padan plain and Iowa and from April $1^{st}$ for 30 days for corn in all four locations) and with a more relevant growth in the development phase (i.e., from December, $1^{st}$ for 140 days for wheat in Sicily, from December, $20^{th}$ for 60 days for wheat in California, from March $9^{th}$ for 75 days for wheat in Padan plain and Iowa and from May $1^{st}$ for 40 days for corn in all four locations), until reaching a maximum ADD value at which this carbon input remains stable during the maturity period (i.e., from April, $19^{th}$ for 40 days for wheat in Sicily, from February, $17^{th}$ for 70 days for wheat in California, from May $23^{th}$ for 75 days for wheat in Padan plain and Iowa and from June $9^{th}$ for 50 days for corn in all four locations). Lastly, a linear decrease towards the background ADD value has been assumed at the end-of-season stage for vegetation (i.e., from May, $29^{th}$ for 30 days for wheat in Sicily, from April, $28^{th}$ for 30 days for wheat in California, from August $6^{th}$ for 25 days for wheat in Padan plain and Iowa, from July $29^{th}$ for 50 days for corn in the Padan plain and Iowa and from July $29^{th}$ for 30 days for corn in Sicily and California). After the end of the season, in correspondence with crop harvest, the soil is amended with plant litter, leading to a sudden increase in the carbon input, as visible in the dashed lines of Figure 4. This carbon input, then, linearly decreases until reaching the background ADD value at the beginning of the next crop stage since it is progressively degraded by the biomass. The constant background ADD is considered to take into account possible other amendments (e.g., manure) or carbon residues (e.g., death roots or stems) from previous cycles. The ADD values shown in Figure 4, typical of root exudation and added litter, have been defined to achieve the typical carbon concentration in the soils under study (see Table 3) from the organic matter balance module of the EW model and also looking at the typical carbon input from wheat and corn residues presented in Wang et al. (2015).

## 3 Results

Our analysis is devoted to a comparison of EW dynamics in the four sites, with special attention to rainfall seasonality (and irrigation when applied), soil type and composition and crop phenology, which are among the factors that most affect EW dynamics. The following results identify the combinations of environmental factors that are most suitable for olivine amendment.

### 3.1 The role of rainfall seasonality on EW dynamics

To explore the role of rainfall seasonality on EW we present pairwise comparisons of EW dynamics achieved for the case studies in Italy and the USA, given the different rainfall seasonality between Sicily and the Padan plain and California and







**Figure 4.** Seasonal variability of the crop coefficient and the added carbon from vegetation for a) wheat in Sicily, b) wheat in California, c) wheat in the Padan plain and Iowa, d) corn in Padan plain and Iowa and e) corn in Sicily and California.

Iowa. Simulations based on scenarios given in Table 1, have been carried out by fixing the soil and the vegetation type and varying the location of the unit surface area.

Figure 5 displays the heatmaps of the ratio between soil moisture, pH and weathering rate achieved for Iowa (IA) and California (CA). Here, the panels in the first row are related to the corn (C) planted in a clay loam (CL) soil, those in the second row to the corn (C) within the silty clay loam (SCL) soil, panels in the third row to wheat (W) planted in a silty clay loam



soil (SCL) and, lastly, those displayed in the last row to wheat (W) within a clay loam (CL) soil. The panels in the last row of the figure, show the average daily ratios of the three considered variables of the model. In the cases when corn is considered,

during the summer season (i.e., Julian day between 150 to about 250), soil moisture is higher in California than in Iowa. In the rest of the year, by contrast, the soil in Iowa is as wet as in California, except for some days around Julian day 300.

Since corn is planted in April and concentrates the reproductive and maturity phases during the driest periods in the Mediterranean climate (i.e., Sicily and California), we assumed that an irrigation contribution would be necessary to avoid stress conditions for plants, which would negatively affect productivity. Following previous studies (Vico and Porporato, 2011a, b),

stress-avoidance irrigation, a scheme designed to avoid stress conditions for vegetation, has been imposed, activating it when soil moisture becomes less than the point of incipient stress. According to this scheme, an irrigation volume is released to the crop only when the crop coefficient is different than zero and when soil moisture becomes less or equal to the point of incipient stress. The provided irrigation volume $\Delta s$ (i.e., the soil moisture gap) is equal to 0.2, but with the condition that soil moisture remains under the value at field capacity, to avoid wasting water due to percolation through deep soils. Figure 6 displays a

typical stress-avoidance irrigation scheme for corn planted in a silty clay loam soil in California and the resulting yearly soil moisture and pH time-series with and without the irrigation contribution. Here, 14 irrigation interventions, over the 150 days of the vegetation cycle, are needed to bring soil moisture above the point of incipient stress, when there is a lack of rainfall and transpiration losses are significant. The total annual volume of supplied water resulted in about 500 mm. The effects of irrigation are visible in the soil moisture time-series (panel c of Figure 6) since it ranges between the value at incipient stress

and the field capacity in the days ranging from about 100 to almost 250 (i.e., when the crop reproductive and mid-season phases are concentrated). In panel d) it is visible that irrigation results in a significant reduction of soil pH during the summer season, compared with the case in which no irrigation is provided. The effects of multiple irrigation interventions are also visible in the pH ratio heatmaps related to corn in Figure 5 since, during the irrigation period, it is higher than 1, corresponding to more acidic soil in California than in Iowa. These considerations about soil moisture and pH affect weathering rate dynamics, which

is higher in California during summer, due to irrigation. The average daily weathering rate ratio assumes values higher than one only for a short period in the beginning and at the end of the year, around the $300^{th}$ Julian day.

In the case of wheat, Iowa provides significantly wetter and more acidic soil conditions, resulting in greater weathering rates that, around the Julian day 300, may reach values twenty times higher than those achieved in the case of California. Comparing rainfall seasonality (panels b and d of Figure 3), it is worth noticing that there are higher values of precipitation distribution for

Iowa than for California (i.e., similar $\alpha$ but more than double $\lambda$ around the Julian day 300), leading to higher soil water content also given by the low transpiration losses of the crop, that in the above-mentioned days is in the initial growing stage (see panel c of Figure 4). On average, over the considered 10 years, weathering rates derived for Iowa are about seven times higher than those in California under the two considered soil types, resulting from an average soil moisture/pH ratio slightly greater/less than 1.

Regarding the Italian case studies (Figure S1 of the supplementary material), similar considerations may be done for corn since, as in California, it requires to be irrigated in Sicily during summer, given the scarcity of precipitation. During the rest of the year, soil in the Padan plain is as wet as in Sicily, meaning that the ratio remains to values equal to about 1. For wheat, soil





moisture is higher in the Padan plain than in Sicily during summer, given the near absence of rainfall in Sicily in this period. Therefore, soil tends to be more acidic and weathering rate tends to be higher (e.g., average daily weathering rate ratio higher

than 1). For the rest of the year, values of the ratio of the weathering rate between the two places, tend to be slightly less than 1, translating to similar olivine dissolution dynamics, even though more favorable in Sicily.

As a result, considering the same soil and vegetation, for the US selected locations, olivine dissolution is significantly faster in Iowa, given the significantly higher values of precipitation distribution, while for the Italian case studies, rainfall seasonality leads to small differences in EW dynamics, given the similar distribution of precipitation across the year. Irrigation, applied

here only during the summer season for corn planted in Sicily and California, leads to a decrease in soil pH and a faster olivine dissolution. More relevant differences in terms of MAP, indeed, would result in very different EW dynamics (Cipolla et al., 2021b), and may also contribute to emphasizing the effect of rainfall seasonality on olivine dissolution, thus emphasizing the relevant role of climatic conditions on EW.

### 3.2   The role of soil type on EW dynamics

The role of different soil types on EW dynamics is explored by considering the same location and vegetation type and varying the two different soils considered in this study, i.e., clay loam soil and silty clay loam), since they are the most frequent in the four sites.

Figure 7 displays heatmaps of the ratio between soil moisture, pH and weathering rate achieved considering the clay loam soil and those obtained with the silty clay loam one. The results highlight that the clay loam soil leads to a larger soil water

content in most of the cases as compared to the silty clay loam. This is because the latter soil type is characterized by lower soil moisture at the hygroscopic and wilting points and, for this reason, evapotranspiration losses are higher, as these values are exceeded most frequently. Similar conclusions can be drawn for the soil moisture value at the field capacity that involves water losses for leaching. This produces more acidic conditions and a faster olivine dissolution under the clay loam soil. Apart from some spikes, occurring on some specific days, weathering rates obtained with the clay loam soil tends to be about twice

as high as those obtained with the silty clay loam soil at all four locations. This small difference is due to the very similar characteristics of the clay loam and silty clay loam soils (see Table 2 for more details), which result in similar hydrological fluxes (e.g., infiltration and leaching rates), water balance and all the derived variables (i.e., pH and dissolution rate).

### 3.3   The role of vegetation on EW dynamics

Besides its role in the soil water balance through transpiration, vegetation also plays a significant role in EW dynamics due

to its acidifying effect given by the displacement of H+ to balance the cations taken from soil water as nutrients (Weil and Brady, 2017) and also because it provides the organic matter that, once decomposed, is one of the $CO_2$ sources in the system (Porporato et al., 2003). The type of vegetation, therefore, influences the time dynamics of soil moisture, pH and weathering rate given that the different crop phenology determines different dynamics of transpiration and added carbon.

Looking at the panels in Figure 8 related to weathering rate ratio, it is evident that corn leads to a weathering rate on average

about four times higher than the one achieved for wheat when they are planted on clay loam soil in Sicily and California (first





**Figure 5.** Time-series heatmaps of the ratio between soil moisture, pH and weathering rate achieved in the case of Iowa and those related to California computed within 10 years after olivine addition. These are related to corn planted in a clay loam soil (panels in the first row of the figure), corn planted in a silty clay loam soil (panels in the second row of the figure), wheat planted in a silty clay loam soil (panels in the third row of the figure) and wheat planted in a clay loam soil (panels in the fourth row of the figure). The average daily ratios of the three considered variables are shown in the last panel of the figure.

and fourth row of the figure), while olivine dissolution dynamics are about the same comparing the two crops planted in a silty clay loam soil in the Padan plain and Iowa (second and third row of the figure), even though providing a slightly more favorable







**Figure 6.** Stress-avoidance irrigation procedure for corn planted in silty clay loam soil in California. Panel a) represents the crop coefficient, b) the yearly soil moisture time-series with and without irrigation, c) the specific volume of water added by irrigation and d) the yearly pH time-series with and without irrigation.

condition when corn is considered (annual average weathering rate daily ratio equal to about 1.5). The greatest weathering rate considering the corn in Sicily and California is obtained during summer and is certainly due to the irrigation contributions. In the rest of the year, or in the cases of Padan plain and Iowa, where irrigation is not considered, crop phenology does not lead to significant differences in EW dynamics. Indeed, as one can notice from Figure 9, when any of the two crops is in the rest


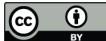


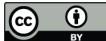

**Figure 7.** Time-series heatmaps of the ratio between soil moisture, pH and weathering rate achieved considering the clay loam soil and those related to the silty clay loam soil. The heatmaps are related to a period of 10 years after olivine addition. These are related to Sicily with wheat (panels in the first row of the figure), Padan plain with corn (panels in the second row of the figure), Iowa with corn (panels in the third row of the figure) and California with wheat (panels in the fourth row of the figure). The average daily ratios of the three considered variables are shown in the last panel of the figure.

phase, apart from leaching, there are water losses due to bare soil evaporation, that are similar in magnitude to transpiration. It, therefore, happens that, when corn is in the rest phase and at the same time wheat is in its initial or mid-season stage, in the first case water losses are mainly governed by bare soil evaporation, while in the other one by crop transpiration. Since they have a



similar magnitude, the fact that wheat and corn cycles are not in phase does not affect much water balance and, in turn, pH and weathering rate dynamics. The only difference for which corn leads to a slightly faster olivine dissolution may be attributed to the smaller active root zone depth (i.e., 0.4 and 0.6 m for corn and wheat, respectively) that causes the soil to reach saturation more easily.

### 3.4  EW case studies

The time dynamics of soil moisture, pH and weathering rate in the two selected locations in the USA are shown in Figure 10, while those related to Italian case studies are reported in the supplementary material. In all scenarios, we considered the application of 10 kg $m^{-2}$ of olivine, which is equivalent to the application of 100 t $ha^{-2}$. California and Iowa are characterized by a slightly different signal in the seasonality of soil moisture, pH and, in turn, weathering rate. The highest soil moisture values for California occur usually in the first (i.e., before the Julian day 100) and the last (i.e., from the Julian day 300 onwards)

part of the year since those days are characterized by the greatest part of the total annual rainfall. Soil moisture assumes low values from the Julian day 100 to about 250 mainly due to the scarcity of rainfall during the summer period. On those days where soil moisture is high, pH and weathering rate reach their minimum and maximum values, respectively, as compared to the rest of the year, confirming the fact that weathering reactions are favored by wet and acid conditions (Cipolla et al., 2021a, b; Hartmann et al., 2013; Calabrese and Porporato, 2020).

For Iowa, the period with generally higher soil moisture covers all winter and part of the autumn and spring months while, during summer, despite a relevant presence of rainfall (Figure 3), soil water content presents low values due to the high transpiration losses, given the occurrence of the corn mid-season (i.e., a peak of the crop coefficient). Comparing the annual average values of the analyzed variables in these two locations, one can observe as Iowa is characterized by a faster olivine dissolution (an annual average weathering rate of 2.13 x $10^{-12}$ mol $m^{-2}$ $s^{-1}$ against the value of 1.61 x $10^{-12}$ mol $m^{-2}$ $s^{-1}$

achieved for California), due to a lower annual average pH (6.61 in Iowa and 7.03 in California) and higher mean annual soil moisture (0.62 in Iowa and 0.57 in California).

A similar situation can be observed from the comparison between Sicily and the Padan plain (Figure S2 of the supplementary material). Because of the similar rainfall seasonality and, of course, the same soil type and vegetation, it is thus possible to make a comparative analysis between Sicily and California and between Padan plain and Iowa regarding the time dynamics of

soil moisture, pH and weathering rate. The greatest difference between Italian and US case studies can be observed in the order of magnitude of soil pH and olivine weathering rate. Indeed, the two sites in Italy are characterized by a higher annual average pH (7.46 in Sicily and 7.55 in the Padan plain) and lower weathering rates (4.79 x $10^{-13}$ mol $m^{-2}$ $s^{-1}$ in Sicily and 3.17 x $10^{-13}$ mol $m^{-2}$ $s^{-1}$ in the Padan plain) than the two sites in the USA. This is mainly due to the different lithological properties of the bedrocks and the mineral composition of the soil that lead to a more significant background weathering flux, hence H[+]

consumption and less acidic soil, in the sites in Italy than those in the USA, before olivine amendment (Ballabio et al., 2019; Hempel et al., 2014; Libohova et al., 2014). The achieved order of magnitude of weathering rate reflects the values presented in the mesocosm experiment of Amann et al. (2020), which presents a condition very similar to the field environment. This





**Figure 8.** Time-series heatmaps of the ratio between soil moisture, pH and weathering rate achieved considering the corn and those related to wheat. The heatmaps are related to a period of 10 years after olivine addition. These are related to Sicily with clay loam (panels in the first row of the figure), Padan plain with silty clay loam (panels in the second row of the figure), Iowa with silty clay loam (panels in the third row of the figure) and California with clay loam soil (panels in the fourth row of the figure). The average daily ratios of the three considered variables are shown in the last panel of the figure.

aspect stresses the importance of a suitable calibration of the background weathering flux, cation exchange capacity and other components of the EW model, allowing to obtain realistic estimates of olivine dissolution dynamics.




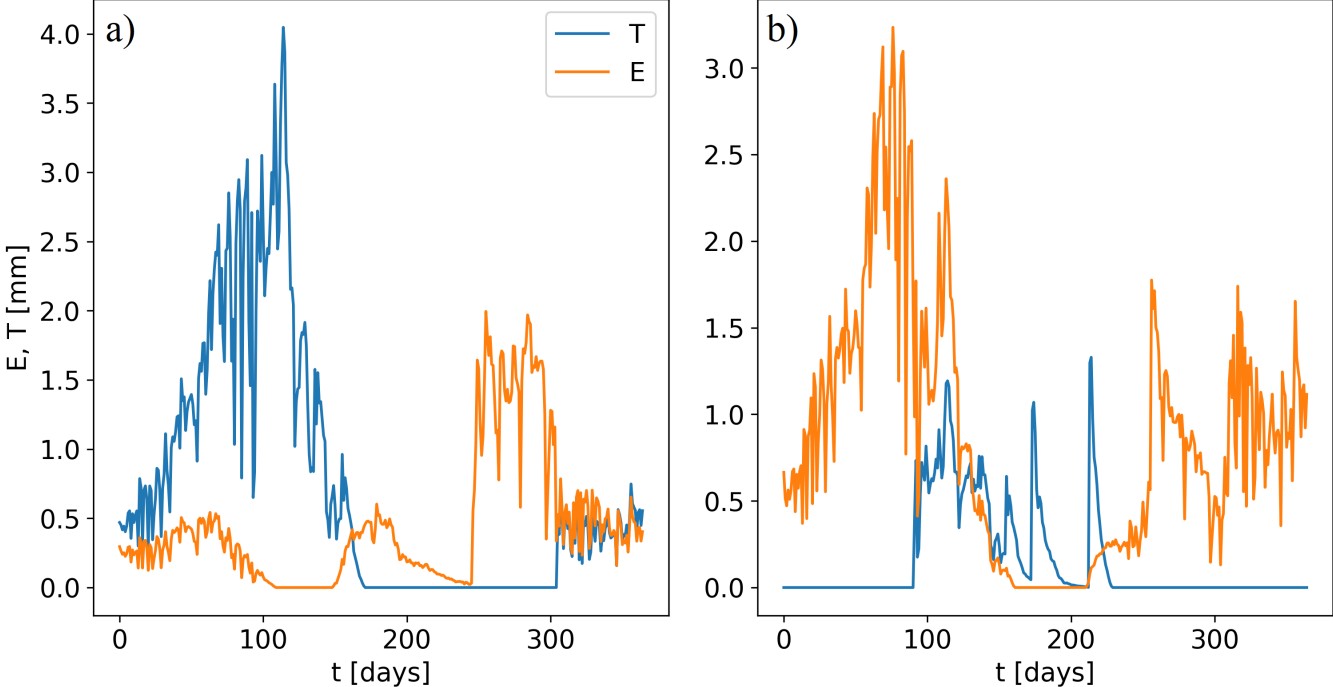

**Figure 9.** Annual time-series for bare soil evaporation and crop transpiration in the case of a) wheat and b) corn planted in a clay loam in Sicily. Corn is not irrigated in this case.

The seasonal variability of the sequestered $CO_2$ related to the four sites under study is presented in Figure 11. This is calculated, as in Cipolla et al. (2021b), as the leaching of the extra $HCO_3^-$ and $CO_3^{2-}$ produced by olivine dissolution, considering a time window of 10 years after olivine amendment. In general, the low monthly values of sequestered $CO_2$ for all case-studies are due to the generally low leaching rate, which is provided by low MAP values for all the considered sites. The annual average sequestered $CO_2$ resulted equal to 0.62 kg $ha^{-1}$ $y^{-1}$ for Sicily, 0.39 kg $ha^{-1}$ $y^{-1}$ for the Padan plain,

2.21 kg $ha^{-1}$ $y^{-1}$ for California and 4.20 kg $ha^{-1}$ $y^{-1}$ for Iowa. The values achieved for the US locations are lower but still comparable with those of Amann et al. (2020), that derived a sequestered $CO_2$ within the range 23-49 kg $ha^{-1}$ $y^{-1}$, amending more than the double of olivine with respect to our study (i.e., 22 kg $m^{-2}$) in his mesocosm experiment and having similar conditions to the field environment.

The seasonal pattern of the sequestered $CO_2$ is very similar to the one of rainfall (Figure 3), highlighting the fact that

hydrological processes play the most relevant role in characterizing EW dynamics. On the other hand, the seasonality of sequestered $CO_2$ is not in phase with crop phenology. A great amount of rainfall, contextually occurring to low transpiration losses, leads to high soil water content, resulting in a high weathering rate, with a correspondence increase of $HCO_3^-$ and $CO_3^{2-}$ concentration produced by olivine reaction with $CO_2$, and a high leaching rate. Apart from that, the order of magnitude of sequestered $CO_2$ depends a lot on the achieved weathering rates, which are strongly connected to soil pH before the olivine





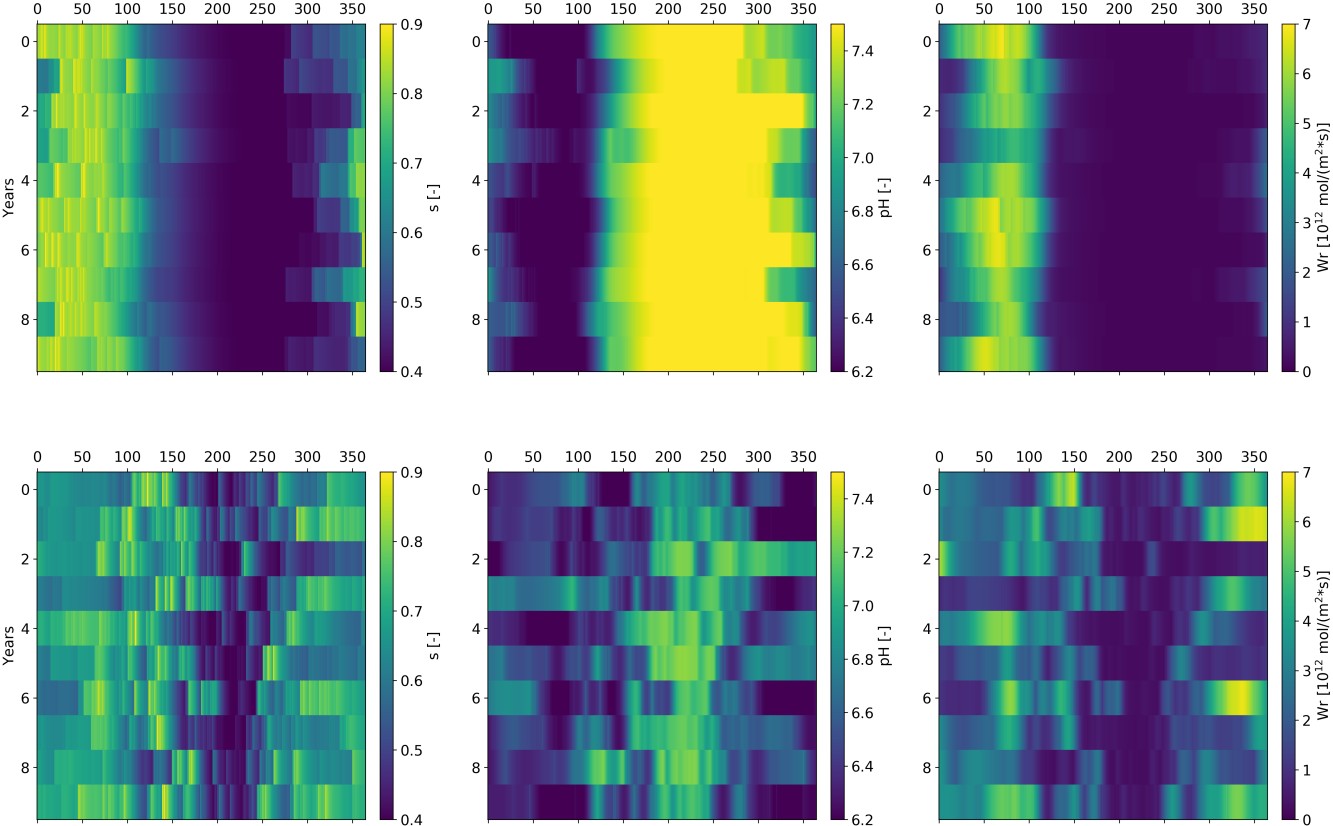

**Figure 10.** Time-series heatmaps of soil moisture, pH and weathering rate computed within 10 years after olivine addition. These are related to wheat planted in clay loam soil for California (panels in the first row of the figure) and corn in silty clay loam soil for Iowa (panels in the second row of the figure).

amendment. In the case studies here presented, this effect is given by the background weathering flux, which provides more acidic conditions within California and Iowa with respect to Sicily and the Padan plain.

## 4 Discussion and conclusions

The introduction of rainfall seasonality, irrigation and crop phenology, in addition to soil properties and composition, in the EW model presented in Cipolla et al. (2021a) is fundamental to characterize EW dynamics in specific cropland areas of the world. In this regard, we here analyzed EW dynamics for two hypothetical case studies in Italy, Sicily and the Padan plain, and two in the USA, California and Iowa.

Analyzing the interactions among rain and crop properties helped better capture the EW process in seasonal cropland areas and its carbon sequestration potential. From the case studies presented in this manuscript, we showed that a fundamental role





**Figure 11.** Box plots representing the seasonality of the $CO_2$ sequestered by leaching of extra $HCO_3^-$ and $CO_3^{2-}$ produced by olivine dissolution, computed over the 10 years after olivine amendment. The plots are related to a) Sicily, b) Padan plain, c) California and d) Iowa.

in EW dynamics is played by rainfall distribution. Comparing EW yields in the case of wheat planted in a clay loam soil for
Sicily and California and corn in a silty clay loam of the Padan plain and Iowa, we found that Sicily in the former case and
Iowa in the latter are the most favorable for a possible EW intervention. Especially in these latter case studies, the sequestered
$CO_2$ is perfectly in phase with rainfall distribution. A great amount of rainfall, contextually occurring during low transpiration
losses, leads to high soil water content, resulting in a high weathering rate, with a correspondence increase of $HCO_3^-$ and



$CO_3^{2-}$ concentration produced by olivine reaction with $CO_2$, and a high leaching rate, according to with these ions are taken
away from the reference domain. Irrigation, when considered, provides a significant change in EW dynamics since it leads to a decrease in soil pH and a consequent increase in weathering rate. However, since the stress-avoidance scheme brings soil moisture to the field capacity at most, it does not affect sequestered $CO_2$ by leaching.

Taking into account the case of Iowa, which resulted in the highest carbon sequestration rate and is characterized by a cropland area covered by the corn of about 56,000 $km^2$, the annual average sequestered $CO_2$ could reach the value of about
0.023 Mt $y^{-1}$, if the whole cropland area were amended with olivine. Sicily, instead, may sequester on average a mass of 0.0002 Mt $y^{-1}$, if amending the total cropland area cultivated with wheat of about 265,000 ha. As stated by Amann et al. (2020), extending the here achieved carbon sequestration rates at the global scale leads to a low carbon sequestration potential of the EW strategy, if compared with the annual global $CO_2$ emissions by fossil fuels. It is clear that, as we demonstrated in our simulations, an accurate estimation of EW carbon sequestration potential requires an in-depth analysis at the global scale,
to take the strong spatial variability of the process into account. This would also allow exploring EW dynamics in areas of the world characterized by a higher MAP, thus with a greater capability to sequester carbon. The rates of carbon sequestration presented in Beerling et al. (2020) at the global scale are about three orders of magnitude higher than those achieved in this study. However, these are derived using a one-dimensional vertical reactive transport model that considers the $CO_2$ captured by EW as the dissolved inorganic carbon due to the weathering reaction (apart from the $CO_2$ emissions due to logistic operation),
which is similar to the one that we called $CO_{2,sw}$ in Cipolla et al. (2021b). In effect, even in this our previous work we obtained about a three order of magnitude difference between the $CO_{2,sw}$ and the $CO_2$ leached, but we believe that this latter is better connected to the actual sequestered $CO_2$, given that some $HCO_3^-$ and $CO_3^{2-}$ dissolved in soil water due to olivine dissolution may react with $H^+$ forming the carbonic acid, thus potentially releasing $CO_2$ back to the atmosphere.

The EW applications here presented can be further improved by taking into account some processes that were neglected
here. For instance, the competition between fertilizers, composed of nitrates and phosphates, and olivine in the use of $H^+$ ions could be an important aspect to be characterized given that it may alter the olivine weathering reaction rate and, in turn, carbon sequestration. These can be added to the soil to enhance soil fertility and plant productivity, given that many areas of the world present limitation in nitrogen and phosphorous (Du et al., 2020). Regarding nitrogen fertilization, Hao et al. (2019) carried out a field experiment to understand its impact on soil acidification in a silty clay loam soil in China, adopting a wheat-maize
crop rotation system. A pH decrease due to the application of nitrogen fertilizer can be due essentially to three factors, i.e., nitrate leaching, plant uptake of cations and the leaching of bicarbonate ions. The authors derived the contribution on soil pH of each of these three factors, finding out that the greater plant nutrients uptake, due to the increase in soil fertility, is the main factor that leads to pH decrease, given their displacement of $H^+$ action for maintaining the charge balance (Weil and Brady, 2017). Furthermore, the high clay fraction of the soil under study resulted in low leaching of nitrogen products (e.g., $NO_{3-}$),
causing consequent high N losses in the air due to a great denitrification rate. Despite this acidification may be beneficial for olivine EW, nitrogen fertilizer application may lead to the presence of nitric acid (i.e., $HNO_3$) in soil water that, reacting with carbonate rocks (e.g., calcite), which in this case represent the nature of bedrock in Sicily and Padan plain, releases $CO_2$ to the atmosphere (Hartmann and Moosdorf, 2012), thus reducing carbon sequestration potential.



Despite we here considered olivine application for EW, our model is flexible to consider the use of any other silicate mineral.

Indeed, many EW experiments have been conducted with wollastonite or basalt, among various aspects to avoid the high Ni and Cr content of olivine. Haque et al. (2020) carried out a wollastonite EW experiment on three farms with different plants, located in three parts of the world. The authors realized that soil inorganic carbon content, in the form of $CaCO_3$, was significantly higher (up to three times) than in the same fields without amendment. Furthermore, Kelland et al. (2020) derived greater carbon sequestration rates (i.e., 2-4 $t_{CO_2}ha^{-1}$) respect to our study and the experiment of Amann et al. (2020) by amending

with basalt powder a clay loam soil planted with sorghum. However, this estimation has been derived by conceptualizing experimental results with a 1-D reactive transport model that does not take into account fluctuations in soil moisture, which are fundamental in EW dynamics. Apart from the leached $HCO_3^-$ and $CO_3^{2-}$, carbon sequestration has been defined as a function of pedogenic carbonate minerals that formed as a result of the combination of calcium and magnesium with bicarbonates of basalt dissolution. Therefore, a possible development of this work may consist of a comparison of EW yields under the

amendment of different silicate minerals in various areas of the world, possibly taking into account the formation of pedogenic carbonate minerals upon silicates dissolution.

Another relevant aspect to consider when planning an EW intervention is its economic feasibility of itself. For this purpose, carbon sequestration potentials have to be compared with costs, to find a better compromise between these factors, such that the country adopting this strategy may afford it without a great economic effort. Beerling et al. (2020), after quantifying the

CDR potential for several countries of the world, set up a cost analysis for each nation, to see if the countries with the highest carbon sequestration potential also present the lowest costs. The authors found an average global cost of about 100 USD per ton of sequestered $CO_2$, with the highest values obtained for the United States, Canada, China, Poland and Spain, especially for a great fraction of the involved cropland area. These costs are however similar to other CDR techniques, such as direct air capture and biochar, leading to considering EW as a reasonable intervention to mitigate climate change. The main differences

in costs depend on many factors, such as the cost of electricity for mineral grinding, transport and spreading operations. An in-depth cost analysis is therefore necessary with the aim to identify those countries providing simultaneously the highest carbon sequestration potential minimizing the related costs. This may be a future enhancement of this work in perspective to apply the EW model at the global scale, deriving a global spatial distribution of $CO_2$ sequestration and costs and providing a tool to decision makers for an actual future application of the EW strategy.

*Code and data availability.*

*Author contributions.*   Giuseppe Cipolla: Conceptualization, Data curation, Formal analysis, investigation, Methodology, Writing – original draft preparation; Salvatore Calabrese: Conceptualization, Investigation, methodology, Supervision, Validation and Writing – review & editing; Amilcare Porporato: Conceptualization, Supervision, Validation and Writing – review & editing; Leonardo Noto: Conceptualization, Investigation, Project administration, methodology, Supervision, Validation and Writing – review & editing.



*Competing interests.* The authors declare that they have no conflict of interest.



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
