# Peer review of "Effects of precipitation seasonality, vegetation cycle, and irrigation on enhanced weathering"

_EGUsphere, 2022_

## Referee Comment (RC1)

**Review of Cipolla et al EGUsphere-2022-196**

**General comments**

Enhanced weathering (EW) is a biogeochemical carbon sequestration strategy which is currently gaining interest in the light of climate change. This paper considers several case studies of EW in the USA and Italy, and considers important aspects of soil moisture and rainfall effects which have been largely neglected in previous studies.  The authors have used an existing EW model (Cipollo et al. 2021, *Adv. Water Res.* **154**:103934) to determine the seasonality of leached bicarbonate and carbonate as the metric of carbon sequestration. Their weathering rates and carbon sequestration rates are within the range of other studies, but water fluxes (precipitation, evapotranspiration) reducing soil moisture greatly reduced leached carbon during the growing season.

**Specific comments**

Generally, this is a good study but there are a number of omissions in the manuscript that need to be rectified.  For example, no details are provided about the olivine application rate or whether the application was repeated. More detail about the tuning of the model, e.g. the background weathering rates, is required; it is unclear whether cation exchange capacity or other parameters were also tuned. The chemistry of rainfall and irrigation affecting key aspects of the model, such as the effect of soil moisture on pH and weathering, needs to be clarified. Comparison of the grain-scale weathering rates to those of Amann et al (2020) should be revisited as it seems the modelled rates are actually higher rather than similar.  Some of the figures do not include enough information or could be improved by better labeling. These weaknesses should be easy for the authors to address; places in the text where these specific aspects should be addressed in the manuscript are detailed below amongst the English language and other technical corrections.

**Technical corrections**

**Figures**

As many readers may look at the figures without reading the text beforehand, it would be helpful to make it easier to understand the main elements of the story at a glance, with an indication of where more information can be found in the text.  As they stand, some of the figures (especially the heatmaps) currently require careful study and references to the text to understand them.

**Figure 3**. Rainfall frequency is noted as a reason why Iowa has higher weathering rates than California, and it would have been useful to see  $\lambda$ for the states superimposed.  The top row could show $\lambda$ for Padan, Sicily and IA, CA and the bottom row could show $\alpha$ for the same cases. This arrangement would still allow readers to compare $\alpha$ and $\lambda$ seasonality for individual sites.   In any case, please make it clear what $\alpha$ and $\lambda$ are.  Y axis labels should read "Mean rainfall  $\alpha$ [mm]" and "Rain frequency $\lambda$" or something similar. In the caption, refer readers to section 2.2.1 for more details.

**Figure 4**. Y axes labels for Kc should read "Crop coefficient Kc".  In the caption, clarify that ADD is added organic carbon.  Refer readers to Section 2.2.3 for more details.

**Figure 5**. It would be helpful if the labels "Corn Clay Loam" etc could appear on the left of each row of this figure, the individual panel colorbars were removed, and a single colorbar with legible label e.g. "soil moisture ratio $S_{IA}/S_{CA}$" appeared at the top of each row.  Not all software makes it easy to do this but it would greatly improve the readability of the whole figure. The bottom row lefthand label would be "Average daily ratio" and the individual  Y-axes labels would be e.g.

"$S_{IA}/S_{CA}$". The label for column 3 should make it clear what scale is being considered, e.g. "Grain scale weathering rate ratio".

**Figure 6**. Please add letter designations to the panels (a, b, c, d). The Kc panel Y axis label should say "Crop coefficient Kc" and that for the bottom left panel should say "Specific soil moisture s".

**Figures 7, 8, 10**: Similar remarks as for Figure 5. Please add column and row labels. Room for these labels will be available if the individual colorbars are removed and master colorbars for each column appear at the top with the column labels. Also, make it clear that the weathering rates are per mineral surface area rather than per land area, e.g., mol/(m$^2$ olivine s) in Figure 10.

**Figure 11**. Are the samples contributing to the boxplots monthly values from the ten years of a single run, or from several runs where some parameter(s) varied? Which crops and soils are involved in these simulations; which simulations from Table 1 are included? Please clarify. Consider adding the place names to the figures so that readers can see this information at a glance.

**Introduction**

line 20: allows

lines 20 and 21: Yes, there are many studies about olivine because it is widely distributed and has relatively fast dissolution rates. However, the sentence is a little awkward and I had to read it several times. Better to say that "many studies discuss using olivine (often modelled as the end-member forsterite $Mg_2SiO_4$) ..." Olivine is a solid solution series between forsterite ($Mg_2SiO_4$) and fayalite ($Fe_2SiO_4$), but the common ones tend to be more Mg-rich and rate laws for forsterite dissolution are freely available.

line 31: "...(i.e., fungi and bacteria) that, ..."

line 42: please make it clear that the weathering rate is per square meter of mineral, not per square meter of land.

lines 50–52: The sentence about the models summarized by Taylor et al is a little bit difficult to understand. The models do indeed vary in their degree of complexity and plant processes may well be absent or oversimplified. A better wording might be: "The reactive transport models summarized by Taylor et al. (2017) vary in their degree of complexity and plant processes may be absent or oversimplified."

**2.1 Methodology**

Please describe the olivine applications: one-time or annual, mass per unit area applied, specific surface area modelled, depth of soil into which the olivine is mixed. This information deserves either a subsection or, if journal guidelines and space permits, a table. The source of the weathering rate law for olivine used in the model should also be cited.

**2.2.2 Soil type and composition**

How was the Hartmann and Moosdorf (2012) lithological map used? Were minerals assigned to the different lithological classes and then weathered individually in the soil, or was a generic rock defined for which the rate constant (rather than the apparent surface area of the rock/mineral) was tuned to the reported pH for the soils? What stoichiometry (base cations, Al, C, Si) was assigned to the native minerals/rocks?

Soil properties which differ for the four sites should be tabulated, perhaps extending Table 3: CEC, mean initial pH, bedrock type from Hartmann and Moosdorf (2012).

**2.2.3 Crop cycle**

Please clarify what the "crop coefficient" represents as soon as it is introduced, i.e., it is a proportionality constant relating actual evapotranspiration to potential evapotranspiration and depends on the crop and stage of growth. Are you using the Kc and/or crop stage length values tabulated in Tables 11 and 12 of the FAO website (https://www.fao.org/3/X0490E/x0490e0b.htm), or following the procedures outlined on that website?  In either case the FAO guidelines and tables deserve a proper citation.

Lines 204–215: This sentence is very long and the beginning of it is awkward.  It is obvious that root exudation products are connected to the vegetation cycle so this does not need to be stated. Reword as follows: "Root exudation products consist of carbon-based compounds ... (Shen et al.,2020). Their contribution ... during the initial growing stage (... all four locations).  During the development phase ..."

**Results**

**3.1 The role of rainfall seasonality on EW dynamics**

line 239: The comma after "... the figure" is unnecessary.

Most of the paragraph about irrigation does not belong here; it deserves its own subsection. A bit more information would help explain why irrigation lowers the pH, and whether rainfall does the same. Are any ions being included in the irrigation water and does this differ from rainfall?  Are rainwater and/or irrigation water in equilibrium with atmospheric $pCO_2$?  Is the saturation state of the olivine playing a role where soil moisture is low? Discussion of the heatmaps and the influence of the irrigation shown there can remain in Results.

lines 259–260: Awkward sentence. Reword: "These considerations about soil moisture and pH affect  weathering rates, which are higher in California in summer due to irrigation."

Lines 265–266: Reword: "... Julian day 300). Combined with low transpiration during the initial growing stage, this leads to higher soil water content."

Line 268: Edit: "... resulting from slightly higher average soil moisture and slightly lower pH."

Lines 270–271: Reword the whole sentence: "Similar considerations apply to corn grown in Italy (Figure S1). In summer, corn requires irrigation in Sicily but not on the Padan plains.  During the rest of the year, the Sicilian and Padan plains soil have similar soil moisture and the soil moisture ratio is near 1."

Line 276: Reword: " Therefore, the Pandan soil tends to be more acidic and weathering rates tend to be higher ...).

Line 277–279: The word "significantly" implies that a statistical test has been done but this does not seem to be the case.  The sentence would also benefit from rewriting, e.g., "For the same soil and vegetation, higher rainfall leads to considerably faster olivine dissolution in Iowa than in California

due to higher soil moisture driven by higher seasonal rainfall frequency (λ). For the Italian case studies ...".

Line 281: What is meant by "More relevant differences..."? Were your selected case studies not relevant in terms of MAP? Do you mean larger differences?

Line 282: Too many instances of "emphasizing"; reword as "2021b), emphasizing the effect of rainfall seaonality and climatic conditions on olivine dissolution and EW."

**3.2 The role of soil type on EW dynamics**

As the range of soil textures for the case studies was somewhat limited, it would have been interesting to see what the model predicts with more extreme textures, such as clay and sandy soils.

**3.3 The role of vegetation on EW dynamics**

Line 311–313: Too many commas and repeated words here. Please edit: "...  when either of the two crops is in the rest phase, water losses due to bare soil evaporation are similar in magnitude to transpiration for the other crop.  The fact that wheat and corn cycles are not in phase ..."

**3.4 EW case studies**

Line 322: Is this a one-time application of 10kg/m$^2$ or is it repeated annually as in some other studies?  As stated above, information about the olivine treatments needs to be either tabulated or presented in Section 2.

Lines 330–332:  Awkward sentence. Reword: "... spring months, but in summer soil moisture is low due to high transpiration losses associated with a peak of the corn crop coefficient."

Line 333: "... one can observe that Iowa ..."

Line 337: The first sentence ("A similar situation ... ") looks like it belongs at the bottom of the previous paragraph as it compares the two Italian sites with similar conclusions to the two American sites. This paragraph is about comparison of Sicily with California, and Padan with Iowa.

Line 338: Remove "thus"

Lines 340–341: Please reword as "... can be observed in soil pH and the order of magnitude of the olivine weathering rate."

Line 347: Amann et al. (2020) used a Belgian soil with pH~6.6 (their table S1) which is very similar to the average annual Iowa soil pH of 6.61 given in line 335.  However, the weathering rates (mol Olivine m$^{-2}$ s$^{-1}$) modelled for Iowa (2.13e-12 = 10$^{-11.67}$), California (1.61e-12 = 10$^{-11.79}$), Padan (3.17e-13 = 10$^{-12.50}$), and Sicily (4.78e-13 = 10$^{-12.32}$) are all at least an order of magnitude faster than Amann et al's rates (10$^{-13.12}$ and 10$^{-13.75}$ for coarse and fine dunite respectively).  This is not necessarily a problem but the model rates do not really "reflect" those of the Amann et al study. Their specific surface areas (their Table 2, m$^2$ g$^{-1}$) were measured with gas adsorption which likely overestimates the actual reactive surface area of the dunite, unfortunately they do not also present more conservative geometric surface areas. What specific surface area was used in the model?  As stated above, basic information about the olivine treatments should be tabulated or described.

Line 348: Section 2.2.2 discusses the calibration of the background weathering, but says that CEC was set based on existing CEC data (Ballabio et al. 2019 and USDA) which does not necessarily imply that CEC was calibrated; it seemed reasonable to assume that the CEC values used were simply means of CEC measurements from the cropland areas in the four regions (pink areas of the maps in Figure 2). If CEC or any other parameter of the weathering model was calibrated, please give details in Section 2.2.2. Then explain how these parameters affect the weathering fluxes and link to the rest of this paragraph comparing the Italian and US case studies.

Line 357: This sentence is ambiguous; it is not clear which study the 22 kg m$^{-2}$ applies to. Amann et al. (2020) seem to have applied 22 kg dunite m$^{-2}$ to their mesocosms; they said the dunite was about 90% olivine of which 92% was forsterite, so they applied 19.8 kg olivine m$^{-2}$ and 18.216 kg forsterite m$^{-2}$. What was the application rate here? it is never stated anywhere in this manuscript as far as I can see.

line 362: Replace "correspondence" with "corresponding".

Line 395: Here olivine-derived $CO_2$ leached and $CO_2$ in soil water are distinguished, with leached $CO_2$ being the preferred metric of carbon sequestration. If EW were rolled out on a large scale and $CO_2$ consumption then calculated based on river water samples, then the leached DIC (dissolved inorganic carbon), alkalinity or $HCO_3$ based on major cations, is indeed relevant rather than the chemistry of the soils. It could also be argued that DIC stored for centuries to millennia in groundwater is actually sequestered at least on those timescales. If those long flowpaths comprise closed systems, the total carbon will be constant and speciation of the carbonate system will not lead to degassing. when that water eventually enters streams or rivers the likelihood of degassing due to turbulence or seasonal mixing in estuaries should be similar to waters entering at similar points on shorter flowpaths. Following the carbon all the way to the ocean where it is believed to be sequestered on $10^5$-year timescales (e.g., Renforth and Henderson 2017, *Rev. Geophys.* **55**:636–674) is not really straightforward! In any case, it is not clear how the Cipolla et al. (2021) model calculates loss to groundwater. Please clarify.

Lines 410–413: Nitric acid weathering may well weather both olivine and carbonate rocks, resulting in loss of $N_2O$ (a potent greenhouse gas) either on site or downstream, as well as $CO_2$ degassing in the latter case . The beginning of this sentence suggests that nitric acid is beneficial for EW. Reading the whole sentence several times, it seems this is not the intention, but rather that nitric acid is not beneficial even though it may increase olivine weathering rates. Please reword, e.g. "Even though acidification may increase olivine weathering rates, nitric acid (i.e., $NHO_3$) from nitrogen fertilizer would react with carbonate rocks such as those comprising the bedrock in Sicily and the Padan plain, releasing $CO_2$ to the atmosphere (reference) and reducing carbon sequestration potential." Please find another reference because Hartmann and Moosdorf (2012) do not mention nitrogen, although several other Hartmann papers do.

Line 414: "Despite we here considered olivine application for EW" is a bit awkward; please consider rewording, e.g.: "Even though we considered olivine application for EW here"

---

## Referee Comment (RC2)

**Review of Cipolla et al EGUsphere-2022-196**

by Ingrid Smet (ingrid.smet@fieldcode.com)

**General comments**

Due to the climate crisis emergency, research into enhanced weathering (EW) as a potential method for carbon dioxide removal (CDR) has increased exponentially over the past decade. Land based application of EW is thereby tested from lab over mesocosm to field scale experiments, representing increasingly more realistic conditions which are however also increasingly more complex - and require increasingly more time (from weeks-months in lab experiments to up to 10 years in field experiments). As time is of the essence when it comes to climate change mitigation, model computations of EW scenarios can play an important role in assessing the potential for $CO_2$ sequestration under specific climate, soil and crop conditions. To achieve this, close collaboration between EW ´lab/field´ and ´computer´ based researchers is necessary to coordinate their research and continuously use insights gained from one field as new input for the research in the other field.

This manuscript thus represents a very relevant study on modelling the effects of rainfall seasonality, irrigation, crop growth cycle and soil type at 4 different cropland sites across the world. The complexity of the authors´ EW model and the as realistic as possible input data for most of their model´s variables make it stand out and represent an important contribution to research into CDR potential of terrestrial EW.

The main weakness of this manuscript is the lack of some relevant background knowledge regarding mineralogy, petrology and soil formation. This is reflected in a rather poor and unrealistic modelling of the mineralogy of the soil, and the absence of necessary information on the ´olivine´ material used as soil amendment for EW. This can, however, certainly be addressed in a revised version of the manuscript.

Comparison of the dissolution and $CO_2$ sequestration rates obtained from the current model with those from the (few) published lab and field experiments could also use further discussion. It would be more valuable when the conditions (crop, olivine amendment, soil type, water availability, …), and the methods to calculate these rates, are also compared for the lab/field and model studies. Further exploration of the plausible reasons for any observed differences between lab/field and model results would also be interesting.

Future further improvements of this excellent EW model could be to introduce a combination of different minerals as EW source material, reflecting the reality of Ca-Mg silicate rock powders proposed for EW. A multi-mineral design of the soil´s mineral composition would also greatly benefit the computed background weathering ratio prior to EW. Using model parameter data from ongoing field experiments could be a next step to overall improve the EW model, which then in turn will provide relevant insights into real life EW experiments.

*Specific comments*

Below follows a list of all my comments, ordered according to the manuscript´s structure. Besides language corrections (yellow) there are "requests and suggestions to rephrase" as well *more explanatory paragraphs to clarify (geological-mineralogical) concepts relevant to EW and this manuscript*.

The changes suggested above to improve the manuscript are presented in more detail within these comments.

**Title**

As the title is now, it suggests that mainly rainfall seasonality, vegetation cycle and irrigation have been studied in detail to assess their effect on EW. The manuscript however also investigates the effect of having two (not too) different soil types. So perhaps include this as a fourth variable, and also point out that this is a model. For example

"Effects of precipitation seasonality, irrigation, vegetation cycle and soil type on enhanced weathering – Modelling of cropland case studies across four sites."

As pointed out above and further discussed below, the mineralogical composition of the soil used in these models is significantly less representative or realistic than the other four model parameters mentioned in the title.

If this is corrected in a revised version of the manuscript, 'soil type' in the above suggested title can be replaced by 'soil composition'.

**Abstract**

- lines 9-10: … strongly affected also by the pre-EW soil pH, which is one of the main factors controlling soil pH before olivine amendment. The same parameter is referred to her: pre-EW soil pH = soil pH before olivine amendment. After having read the rest of the manuscript, it seems that ´pre-EW soil pH´ should be replaced by ´background weathering flux´ or the ´mineral composition of the soil´, which largely determines the background weathering flux.

- lines 10-11: Looking at the numbers presented here for sequestered $CO_2$, and without further explanation on the modelled rainfall seasonality, crop cycle or soil type here in the abstract, this sentence does not make so much sense. How are 4.20 and 0.62 the largest compared to 2.21 and 0.39? Do you mean to compare the two US sites with one another, and the two Italian sites with one another? After reading the manuscript I understand what is meant here, but the abstract should make sense on its own. Please rephrase to make the ´take home´ message more clear.

**1 Introduction**

- line 20: bicarbonates (as on line 21 there is also the plural carbonates)

- lines 21-22: please consider rephrasing "which are then leached out of the soil, transported by groundwater, and eventually reach the oceans or precipitate as carbonates" to clarify that

carbonate precipitation may happen at any stage from (bi)carbonate formation in the soil to transportation into groundwater and transfers via rivers into the ocean.

- lines 22-25: please rewrite/revise the sentence "Many studies… …(Hartmann et al.,2013)." to clarify/correct the following:

*- Olivine is the general name for the solid solution series between the ideal end member minerals forsterite ($Mg_2SiO_4$) and fayalite ($Fe_2SiO_4$), where the Mg richer varieties are more common and also more reactive with $CO_2$ and $H_2O$. (Generally, it is the Mg-Ca-silicates that have the most potential for CDR - forsterite and wollastonite $CaSiO_3$.) So for ease of representation/calculation $Mg_2SiO_4$ is often used to represent an olivine mineral with real formula $(Mg_{1-x},Fe_x)_2SiO_4$.*

*- The mineral olivine is found in igneous rocks: whereas (1) volcanic rocks such as basalt and (2) plutonic rocks such as gabbro typically have up to ca 10-20vol% of olivine; (3) ultramafic rocks such as parts of the earth's mantle exposed on the surface in ophiolite assemblages can have much higher olivine contents up to 95%. So most 'olivine' mines across the world are quarrying ultramafic mantle rocks (for example dunite, peridotite) as they have higher olivine contents, but 'basalt' is also quarried and used for EW as despite its somewhat lower olivine contents it consists of other silicates that provide plant nutrition upon dissolution.*

(Gabbro is NOT a volcanic rock)

- lines 28-29: There is indeed still a discrepancy between silicate dissolution rates observed in labs (where they are more easily measurable) and in the field (where the main challenge is to differentiate the EW signal from the other biogeochemical processes going on).  However a lot of lab, mesocosm and field experiments have been carried out since the reference to this issue in White & Barley (2003). As the research field for EW as CDR method has exponentially grown in the last 1-2 decades, it seems better to provide a more recent reference on this issue.

- line 34: "any other Ca-Mg-silicate mineral" (see comment lines 22-25); basalt is NOT a mineral, it is a rock containing different minerals one of which can be olivine → "such as  wollastonite"

- line 35: the single-mineral particle lab experiments of dissolution you refer to here are not on Ca-Mg-silicates most often considered for EW, but instead on other silicate minerals that are relevant to natural weathering and soil formation (albite feldspar in Hellman and Tisserand, 2006; illite clay mineral in Koehler et al., 2003). Perhaps you can replace these with references to olivine, wollastonite, … dissolution rate experiment studies which are more relevant to this study? (for example: Pokrovsky & Schott, 2000 https://doi.org/10.1016/S0016-7037(00)00434-8 ; Oelkers et al 2018 https://doi.org/10.1016/j.chemgeo.2018.10.008 ...)

- lines 35-36: Please rewrite to clarify and correctly group the different types of EW experiments. *Besides single mineral grain dissolution experiments (see above), terrestrial EW experiments can be classified in the following 3 categories: (1) Laboratory experiments involving soil cores/columns to which silicate rock powder (SRP) is added, under controlled T and irrigation conditions, without biological processes (Renforth et al. 2015, Dietzen et al., 2018). (2) Mesocosm or pot experiments where plants and/or soil organisms are added to larger containers of soil with SRP, representing more closely 'real life' conditions but still closed and*

controlled system (ten Berge et al. 2012, Amann et al. 2020, Kelland et al. 2020). (3) Field trials where SRP is added outdoors to a field, grassland, forest soil representing complex open system of real life conditions (published study results so far only with wollastonite: Haque et al. 2020, Taylor et al, 2021 https://doi.org/10.5194/bg-18-169-2021 ).

- line 41: "magnesium and silica concentrations"

- lines 42-43: Please clarify that weathering rates of $10^{-13}$ mol/(m$^2$.s) refers to the surface of the mineral grain in contrast to sequestration rates in kgCO$_2$/(ha.year) which refers to land surface on which mineral dust is spread on.

- line 46: Although White and Brantley (2003) indeed compare field and laboratory observed dissolution rates, the subject of this study is natural weathering of plagioclase and other non Ca-Mg-silicates present in a granite. Could you perhaps find more recent references pointing out the discrepancy of lab, mesocosm and field derived dissolution rates of Ca-Mg silicate minerals relevant for enhanced weathering?

- line 60: "suggesting that the model estimates approach a condition that is more similar to what happens in the field" (mesocosm experiments still do not represent the full complexity of field trials)

- lines 63-64: "Many of the model components are characterized on the base of measurements (i.e. pH and cation exchange)" perhaps better formulated as "Many of the model parameters are obtained from measurements"?

- line 66: The acronym "MAPs" is used here without introducing/explaining it.

**2.1 Methodology**

- line 88: Long sentence which might be more easily readable by splitting as "… to which we refer for details. It links ecohydrological and …"

- line 90: "The model is composed of four closely related components." After reading this paragraph a number of times it is not clear to me which one are these four. Could you please sum them up here, or number them in the following description?

- line 93: "… of soil water ions released by olivine dissolution…" as you refer both to silicates which are anions and magnesium which forms cations

- line 94: Mg$^{2+}$ can be removed here as base cation as it is already referred to as one of the main ions formed upon olivine dissolution

- line 95: "the … (CEC) accounts for the process between": which process?

**2.2 Study areas and data**

- line 116: there seems to be a mistake with the web link:  there is a ´c´ in subscript https://ipad.fas.usda.gov/rssiws/al/globalcropprod.aspx and I get an error message when trying this link

- line119: "active root zone depth of the involved crop." Could you please already here write these specific depths chosen for the corn and the wheat crops in the models?

**2.2.1 Rainfall seasonality**

- line 123: The acronym MAP is used again without writing it out in full before

- line 126: Since acronyms SIAS and USGS in the previous and next line, respectively, are fully written out perhaps this might also be done here for the acronym ARPA.

- lines 133-134: "… are two months out of phase, …" If ´out of phase´ refers to different trends for α and λ, one increasing and the other decreasing, it seems to me this happens in more than 2 months (from 2 through to 6, and from about 9 to 11).

**2.2.2 Soil type and composition**

- line 151: SOC estimations are derived from the GSOCmap which represents "organic carbon content of the first 30 cm soil layer" – are the retrieved $C_0$ and $C_b$ values in the model applied only to the top 30 cm, or also further down to the root depths of 40 cm and 60 cm for corn and wheat, respectively?

- line 163: … consume $H^+$ ions…

- line 165: "existing bedrock. This last information was extracted from the lithological map presented in Hartmann and Moosdorf (2012)." *Although a very valuable publication, it is too general to derive soil mineralogical input data for these 4 respective regions in comparison to the rainfall input data that are carefully derived from real meteorological measurements at these locations. The here used mineralogical/background lithological input data would compare to using the most common meteorological pattern in south Europe, west and central USA. So either acknowledge that the input data for the soil mineralogy of the four sites is much less representative for the real locations than the rainfall data. Or try to find more accurate data for the local geology of these four areas.*

*In case of the latter, please take into account that soils in plains retrieved a lot of their minerals from the weathering of surrounding mountains and might hence not only reflect the local bedrock of the plain but also the mineralogical composition of surrounding mountains. Furthermore, weathering of bedrock and surrounding rocks creates new minerals that end up in the soil. Eventually, the most accurate model input for the mineralogical composition of a soil is obtained from XRD measurements of that soil.*

- lines 168-170: *Carbonate sedimentary rocks are NOT calcite which is a mineral - carbonate sedimentary rocks (e.g. limestone, dolostone, …) are mainly composed of carbonate minerals (e.g. calcite, dolomite, …). Siliciclastic sedimentary rocks are NOT quartz which is a mineral - siliciclastic sedimentary rocks (e.g. sandstone, conglomerate, siltstone, shales, breccia, …) are mainly composed of silicate minerals (e.g. quartz, feldspars, micas, clay minerals, …).* Please correct this by rewriting the sentence.

-line 170: "Lasaga (1984) and 44 (1979)" Please correct the later reference and perhaps also check more recent references on dissolution rate constants for carbonate and siliciclastic rocks.

- line 171: …"calcite and quartz minerals…" It seem from the text that just these two minerals are used for the modelling of background soil weathering, calcite for the Italian sites and quartz for the US sites? If so, please do mention that this is a big simplification of the real soil´s mineralogy which is highly unlikely to exist only of calcite or only of quartz. In case a more accurate estimation of the soil mineralogy is used, a combination of mineral dissolution rate constants of the main occurring minerals should be taken into account.

**2.2.3 Crop cycle**

- line 184: What does FAO stand for? Reference please?

- lines 190-198: When introducing these important computation calculations (1) and (2), please clarify all the different variables in them, as was done for the next computation calculation (3). For example Crop transpiration loss T(s) where s refers to varying soil moisture - refer to Table 2 – and Bare soil evaporation E(s) where s represents…

- lines 208-218: The details on the crop cycle´s different stages and their length at each of the four sites and for each specific crop is better represented in a figure/graph than written out in detail here, introduced in the first paragraph (183-189). For example with an horizontal axis representing the year and a vertical axis which reflects different sites and crops, showing horizontal bars divided in blocks which represent the different stages, having number of length in days inside and a specific colour for each of the specific crop cycle stage.

**3 Results**

- lines 226-228: Rainfall seasonality, soil type, crop phenology and soil composition are correctly mentioned as some of the factors mostly affecting EW dynamics. And parameter input data of these variables for the model calculations are carefully determined based on real life data from the 4 sites. Except when it comes to the soil's mineral composition, where general, non-site-specific and somewhat unrealistic mineral assemblages are used (quartz for the US sites and calcite for the Italian sites) to derive background weathering fluxes. Please either clearly state that these parameter input values are less site accurate than the other ones. Or better find more accurate mineral assemblages typical for each of the four sites and use these to calculate a background weathering flux based on each mineral's relative presence and dissolution rate constant.

- line 229: *Another major control factor of EW dynamics is the silicate rock dust powder (SRP) applied for EW. Its mineral composition greatly determines $CO_2$ sequestration potential (for example whether it is mostly olivine in ultramafic matle rocks, or olivine along with feldspars and volcanic glass in basalt). $CO_2$ sequestration potential is also influenced by how much the SRP's mineralogy differs from the soil mineralogy (see Swoboda et al, 2022 -* 10.1016/j.scitotenv.2021.150976 *).*

In general, information seems to be missing on the 'olivine amendment' that is used in these EW models. It seems that the same imaginary 100% forsterite rock dust is used across the four sites, keeping this input parameter simple and the same everywhere to allow investigation of the effects of rainfall seasonality (with/without irrigation), soil type and composition, and crop cycle – which is the main aim of this study? Or is real olivine rich rock dust modelled, for example the

one used in the mesocosm experiments of Amann et al. (2020) to which results the outcome of these models are compared?

Besides the mineralogical composition of the applied silicate rock dust powder there is other information that is important to better compare the model results to insights from field scale experiments: what is the grainsize of the rock dust? How much of it is applied per m²? Is it left on top or worked into the soil? If the latter, to which depth is it mixed with the soil? Is this application repeated annually throughout the 10 years, or is it a one time application? These SRP parameters also have an important influence on EW dynamics (see Swoboda et al, 2022) and are therefore usually well defined in lab, pot/mesocosm or field experiments. So in order to allow better comparison of EW models and EW field trials, as well as better communication between the scientists carrying out these two kinds of studies, please also include this information as a separate subsection of 2.2 Study areas and data, for example "2.2.4 olivine amendment".

**3.1 The role of rainfall seasonality and irrigation on EW dynamics**

- line 235: "…between soil moisture (S), pH and weathering rate (Wr) achieved…"

- line 236: Before describing the top 4 rows with heat panels, it would be helpful for scientists not familiar with such diagrams to shortly describe how to interpret them. For example, blue colours indicate higher values for a parameter (soil moisture, pH, weathering rate) in California than in Iowa at a specific time and under specific crop and soil conditions. Red colours indicate that at a given circumstances of soil, crop type and rainfall seasonality the soil moisture, pH or weathering rate is higher in Iowa than in California.

- line 240: Is it necessary to use the computation term 'Julian day' here as the model output data are shown horizontally as a year from day 0 to day 365, so one could say "from day 150 through to about day 250" which is more easily understandable for non-modelling scientists? If 'Julian day' needs to be mentioned perhaps shortly explain what exactly this means?

- line 241: As before, Julian day needed or is "some days around day 300" also ok?

- line 254: Soil moisture time-series in the figure 6 caption is referred to as panel b), not c)

- line 255: Please rewrite as "the field capacity in the days from about day 100 up to day 250".

- lines 242-257 until "… is provided.": This paragraph introducing irrigation for the Mediterranean climates  – the reason why it is necessary and how it is implemented in the model – should be moved to '2.2 Study area and data' as a new subsection right after 2.2.1 Rainfall seasonality. So 2.2.2 Irrigation, 2.2.3 Soil type and composition, 2.2.4 Crop cycle, 2.2.5 Olivine amendment. Figure 6 should then also be moved to this earlier section of the paper. The stress-avoidance irrigation procedure for corn planted in Sicily should also be shown in 2.2.2 Irrigation for one of the two soil types, either added to Figure 6 or as a new Figure.

- line 240: When the irrigation paragraph is moved to an earlier section, you can then refer back to it here "soil moisture is higher in California than in Iowa due to irrigation".

- line 259: What is the reason that the soil pH becomes lower, more acid, with increased soil moisture, irrigation? Please briefly clarify.

- line 261: Please be consistent, in line 255 'Julian' was omitted when describing the period from day 100 to day 250. So perhaps generally remove the word 'Julian' throughout this document.

- line 263: Please replace "the Julian day 300" with "the 300th day" or "around day 300".

- lines 267-269: In the concluding sentence "On average, …. weathering rates derived for Iowa are about seven times higher than those in California…" This refers to the cases where wheat is the crop so please clarify this by adding "with wheat". Likewise it might be beneficial to repeat once again in the conclusions of the previous paragraph, lines 259-261, that these are model observations with corn.
Also: Where is this 7X higher weathering rate for Iowa compared to California derived from? The average daily ratio of Wr in Figure 5? Please clarify where this number comes from.

- lines 270-276 where the role of rainfall seasonality on EW dynamics is discussed for the Italian sites: It is unclear why the time-series heat map for the Italian sites is put as Supplementary material as despite the similarities with the US sites with/without irrigation, these maps are sufficiently different. Supplementary material is often a separate document from the main paper containing raw data, so it would be better if this Figure S1 would become the second figure in the subsection 3.1 after the time-series heat maps for the US sites. The explanation written in the Supplementary material along with Figure S1 is the exact same text as what is described here in this section, showing that text and figure best go together (in the main paper).

- lines 270-271: Please rephrase this sentence as it is awkward to read and not very clear.

- lines 275-276: Please rephrase this sentence as it is awkward to read and not very clear.

- lines 278-279: Please rephrase/rewrite these important conclusions regarding the modelled effect of rainfall seasonality and irrigation on EW dissolution rates as the text is difficult to read and unclear. Thereby keep in mind to replace 'significantly' with 'distinctly' (significantly usually refers to statistically verified differences between values).

- line 281-283: "Larger differences in mean annual precipitation would likely result in bigger changes of EW dynamics (Cipolla et al., 2021b), emphasizing the important effect of rainfall seasonality and climatic conditions on olivine dissolution and EW."

**3.2 The role of soil type on EW dynamics**

- line 286: … and silty clay loam soil,…

- lines 288-290: Add the parameter symbols please: …soil moisture (S), pH and weathering rate (Wr) … clay loam soil (CL) … silty clay loam soil (SCL).

- lines 294-295: …weathering rates obtained with the clay loam soil tends tend to be about twice as high as those obtained with the silty clay loam soil… Where is this 2X higher weathering rate for CL compared to SCL derived from? The average daily ratio of Wr in Figure 7? Please clarify where this number comes from.

**3.3 The role of vegetation on EW dynamics**

- line 300: … of $H^+$ to balance …

- lines 301-302: "Brady, 2017). Vegetation furthermore provides the organic matter that, once decomposed, is one of the $CO_2$ sources in the soil system…"

- line 305: … about four times higher than … for wheat… Where is this 4X higher weathering rate for CL compared to SCL derived from? The average daily ratio of Wr in Figure 8? Please clarify where this number comes from.

- line 306-307: " and fourth row of the figure). When both crops are planted in a silty clay loam soil in the Padan plain and Iowa (second and third row of the figure), the olivine dissolution dynamics are very similar.  An annual average weathering rate daily ratio equal to about 1.5 might reflect slightly higher weathering rates for corn."

- lines 311-312: ' when any of the two crops is in the rest phase' please specify which exact periods these are to make it easier to spot them in Figure 9. For example by: '… in the rest phase (from about day aa to day bb for wheat and from about day xx to day yy for corn)'

**3.4 EW case studies**

- lines 320-321: 'The time dynamics of soil moisture, pH and weathering rate across the four locations in Italy and the USA are shown in Figure 10. In all scenarios…'

The time series heat maps for the Italian sites now in Supplementary material Figure S2 should be brought to this section of the main text to illustrate it. As Figures 5, 7, 8 each have 4 rows of three heat maps and an extra bottom row with the average daily ratio, it should be possible to add the Italian heat maps in Figure S2 to those of the US sites in Figure 10.

- line 322: The information regarding the olivine application rate should already have been given in a subsection of section 2.2 on olivine amendment. Why was this rather high application rate of 10kg/m² chosen? Practically, farmers apply lime and other rock dusts annually at a rate of 1-4 tons/ha.

- line 324: …(i.e., before day 100) … (i.e., from day 300 onwards)…

- line 326: values from day 100 to about day 250 mainly …

- lines 333-336: Where can the annual average values for soil moisture, pH and weathering rate for Iowa and California be found? The start of the sentence with 'Comparing the annual average values…, one can observe…' suggests that this can be seen in a figure or table? If these data are only presented here within this paragraph, then please rephrase. "Annual average values of the three variables calculated for California and Iowa suggest that faster olivine dissolution occurs at the latter site ($2.13 \times 10^{-12}$mol/m²s) than at the former ($1.61 \times 10^{-12}$mol/m²s). This is in accordance with a lower annual average pH (6.61 in Iowa and 7.03 in California) and higher mean annual soil moisture (0.62 in Iowa and 0.57 in California). "

Whereas pH seems indeed different between the two US sites, soil moisture shows a smaller difference. How meaningful is the difference between the Iowa and California olivine dissolution rates? Any estimation of the uncertainty on these calculated values?

- line 337: Please add the heat maps of Figure S2 to Figure 10 and add the description of them which is currently in S2 here in the main text of the manuscript. "A similar situation is observed from the comparison of the two Italian sites as Sicily and Padan plain present only small differences in terms of the seasonality of soil moisture, pH, and , in turn, weathering rate. … (i.e., before day 110) and the last (i.e., from day 300 onwards) …with respect to the two sites in the USA."

- line 338: 'Because of the similar rainfall seasonality…' seems to be the start of a new paragraph where now Italian sites are being compared to US ones.

- lines 341-343: No need to repeat the pH and dissolution rates calculated for the Italian sites here if it is already mentioned in the previous paragraph which used to be the text of S2.

- lines 346-347: 'the achieved order of magnitude of weathering rate reflects the values presented in the mesocosm experiment of Amann et al. (2020), which present a condition very similar to the field.' What exactly are the weathering rates presented in Amann et al. (2020)? How do the conditions of their mesocosm experiment compare to those of the models discussed in this paper? What ´olivine´ type used, application rates, which crop in the mesocosm, irrigation scheme, soil type and composition? A comparison of the results of the current study with those of a published paper benefits from some info on the published study.

- lines 348-349: ´suitable calibration´ seems odd in this sentence, perhaps rewrite as "the importance of site representative model input data for the background flux, …"

Rainfall seasonality, irrigation scheme, CEC, soil type, main soil properties and crop phenology have indeed been determined as representative as possible for the four respective sites. In comparison, the soil mineralogy, another very important parameter influencing the olivine dissolution dynamics, chosen for the models is much less site specific or realistic.

- lines 352-353: "The overall rather low monthly values of sequestered $CO_2$ for all case studies are due to the generally low leaching rate, which reflects the low MAP values for all considered sites."

- line 354: "The annual average sequestered $CO_2$ equals 0.62 kg/ha for Sicily, …"

- lines 355-358: The difference between the Amann et al. (2020) $CO_2$ sequestration values and the ones of this study are on a scale of 1 to 2 orders of magnitude – yet considered comparable to one another. The weathering ratio values obtained for the US and Italian sites differ only 1 order of magnitude from one another – yet deemed different (and this difference explained by the least site representative/realistic model input parameter of soil mineralogy). Please be consistent with interpreting the difference between values. It is true that Amann et al. (2020) added 22kg/m² whereas in these models 10kg/m² was applied, but the rock dust of the former only contains about 90% olivine. How does the soil moisture throughout the year compare between both these studies? How was the $CO_2$ sequestration value calculated in Amann et al.

(2020)? What other factors might play a role in the difference of $CO_2$ sequestration rates obtained for these two different study approaches (mesocosm experiment and model)?

- line 362: … with a ==corresponding== increase of $HCO_3^-$…

**4 Discussion and conclusions**

- line 372: Analyzing the interactions ==between rainfall== and crop properties…

- line 378: … with a ==corresponding== increase of $HCO_3$…

- lines 379-380: "… by olivine reaction with $CO_2$. Higher soil water contents also mean higher leaching rates and hence better transport of the (bi)carbonate anions away from the active olivine dissolution zone."

- line 395: … the one we called $CO_{2,sw}$ … Please shortly define/explain this parameter instead of just giving the symbol and referring to a previous publication.
…In effect, even in this  previous work we obtained…

Although I understand the reasoning that (bi)carbonates formed by olivine dissolution but which stay in the ´EW zone´ are seen as a risk to recombine to carbonic acid releasing $CO_2$ back to the atmosphere, and that hence (bi)carbonates leached out from EW zone are interpreted as more reliable measure for $CO_2$ sequestration, it is not so straightforward. Some of the olivine dissolution sourced carbonate anions might precipitate in solid carbonate minerals within the soil (calcite) which is then stable carbon sequestration that can not be traced back in the leached groundwater below. On the other hand, (bi)carbonates dissolved in leached groundwater and hence taken into account for $CO_2$ sequestration calculations, might recombine to carbonic acid and degas $CO_2$ when they resurface or mix with water of different composition, temperature,… The permanence of $CO_2$ sequestered as (bi)carbonates in groundwater through olivine dissolution is difficult to estimate and probably varies from one context to the next. Maybe this suggests that (bi)carbonate anions and DIC are not the best parameters to estimate the amount of captured $CO_2$. Another product from olivine dissolution is $Mg^{2+}$ cations. In general, weathering of silicate rocks will release base cations into the soil water as well as (bi)carbonates. Please see what is written about this in literature and assess the pros and cons of using (bi)carbonates or cations to estimate sequestered carbon. Is there a possibility within your model to obtain values for cations resulting from olivine dissolution, and to use these data for an alternative calculation of $CO_2$ sequestration?

- line 414: Good to come back to possible more complexity in future models regarding the silicate rock dust that can be used for EW.

- line 415: *basalt is NOT a mineral, it is a rock and hence an assemblage of minerals. See comments for lines 22-25. The reason that basalt has lower Ni and Cr contents compared to olivine is because basalt only partially consists of olivine. Since the topic of potential Ni and Cr contamination resulting of EW of olivine rich rocks is touched upon here, please add a sentence explaining that both these heavy metals occur in olivine crystals and are thus released when the latter are dissolved.*

*Another reason to use basalt is that the other minerals it contains release plant nutrient cations upon dissolution, effectively being a natural fertilizer.*

All in all, using silicate rock powder consisting of different minerals, instead of just one mineral (olivine, wollastonite) would greatly improve the model´s representation of realistic field situations.

- line 417: The wollastonite EW field trials of Haque et al (2020) are NOT across the world but at three different locations in Canada.

- lines 417-718: *Wollastonite is a calcium silicate – $CaSiO_3$ – that upon reaction with water and $CO_2$ dissolves and forms, among other products, $Ca^{2+}$ and $CO_3^{2-}$. This cation and anion can combine within the EW zone to form secondary, pedogenic calcite which is then an easy measure to assess how much wollastonite dissolved, and hence $CO_2$ was sequestrated into this new calcite. In case of olivine dissolution, the released cations are less likely to form new carbonate minerals within the EW zone, only under certain chemical conditions they might. This is one of the reasons why $CO_2$ sequestration from olivine and other silicate rock dusts dissolution is more difficult to measure. (see comments line 395).*

- line 421: Indeed, most lab, pot and mesocosm experiments are carried out under continuous (near) saturation of soil moisture which is not representative of the real life situation. The detailed incorporation of rainfall seasonality and irrigation in the here presented model is therefore one if its greatest merits and strengths towards more realistic EW potential predictions.

- line 424-426: Precipitation of secondary minerals as pedogenic carbonates from products of silicate rock powder dissolution in the field is far from well understood and likely not the most common scenario. A more relevant improvement of the here presented excellent EW model would therefore be to go from single mineral olivine (which in reality is never applied as it is not available) to a realistic assemblage of minerals (for example resembling that of a mantle dunite, or a basalt) that takes into account the dissolution rates of the individual minerals and their relative presence in the silicate rock dust.

**Figures**

Figure 3: To allow easier comparison between the average rainfall depth α and rainfall frequency λ (please label both fully on the vertical axes) between the 4 different areas, maintain the same scale for all four diagrams (i.e. λ up to 0.47 and α up to 13.5). Putting the location names in each of the 4 plots would also make it easier to interpret this figure at a glance. In a black and white print out it is not clear which of the two lines is which, so perhaps make one a dotted line and either put in a small legend, or describe in the figure caption which parameter is represented by  the full, and which one by the dotted, line. Also fully describe what the α and λ "rainfall parameters" exactly represent.

Figure 4: Please write full name and symbol for both parameters (crop coefficient Kc and added carbon ADD) in both figure caption and alongside the vertical axes. To interpret more easily the graph, perhaps put a) b) c) vertically below one another in the first column, writing ´wheat´ above it and the site name in each of the 3 graphs. And then have d) and e) vertically below one

another in the second column, writing ´corn´ above this column and the names of the sites in the respective graphs.

Figure 5: Please add the respective symbols for the different variables to the figure caption: soil moisture (S), weathering rate (Wr), Iowa (IA), California (CA), corn in clay loam soil (C-CL), etc. To make this figure easier to read it would be good to have the colour legend just once, write the full parameter name above each column, and the full crop/soil type combination in front of every row.

Figure 6: Please add the symbol to the parameter name in the figure caption, and the full name to the symbol along the Y-axis of the specific graph (crop coefficient Kc, soil moisture S). Letters a), b). c) and d) are missing in the respective panels.

Figure S1: Please make a main manuscript figure occurring after the time-series heat maps for the US sites, and make the same corrections/changes as detailed above for Figure 5. The resolution of the current S1 figure needs to be improved as the Y-axis labels are poorly readable both in print and in the pdf on screen.

Figure 7: Please adjust in the same way as suggested for Figure 5.

Figure 8: Please adjust in the same way as suggested for Figure 5.

Figure 9: Please add the symbol to the parameter name in the figure caption, and the full name to the symbol along the Y-axis of the specific graph (bare soil evaporation (E), crop transpiration (T)). To allow easier comparison of these parameters between a) wheat sand b) corn, please have both Y-axis the same length (4.25 mm).

Figure 10: As before, please write the parameter codes in the figure caption and the full parameter name along with its code above the respective column of heat maps. (soil moisture (S), weathering rate (Wr)). To the left of each of the rows, add the codes of the case studies, and in the figure caption write the case study code with the full description (wheat in clay loam soil in California (W-CL-CA), corn in silty clay loam soil in Iowa (C-SCL-IA).

Figure S2: Include these two rows of each 3 heat maps for the Italian sites in Figure 10, with both full reference and code (wheat in clay loam soil in Sicily (W-CL-SI), corn in silty clay loam in Padan plain (C-SCL-PP)).

Figure 11: Please put the site name in each of the plots to make this figure easier to read. Do these plots reflect the model outcomes from the input parameters used in section 3.4 (last 4 rows in table 1)? Please write again in the figure caption which soil type and crop, with or without irrigation, is presented here for each of the 4 sites.

**Tables**

Table 2: Please have the same number of digits after the separation point for values of the same parameter (they are different for soil moisture at field capacity and saturation hydraulic conductivity). Please also write which of the four study sites are represented by which soil type.

Table 3: According to the text (lines 153-155), biomass pool $C_b$ is defined as 1% of the above-defined carbon input ($C_0$). Yet in the table the values for $C_b$ are 10 times higher than those for

$C_0$? Maybe $C_b$ is here expressed as g/m³ instead of kg/m³ as $C_0$ above?

Please also add to this table the model input values for the different study sites of CEC, derived soil mineralogy, soil pH and calculated dissolution rate constants.

Perhaps it is possible to combine tables 2 and 3 into one table with all soil type and composition data used in the models?

**References**

All references in text are in the references list and vice versa. The only irregularity is the first entry in the references list ´Critical Review …, 1979´ which seems to lack authors but might coincide with the incomplete reference in the text in line 170 ´44 (1979)´.

**Supplementary material**

Please include this in the main manuscript text as suggested in comments above

---

## Author Comment (AC1)

Replies to the comments on "**Effects of precipitation seasonality, vegetation cycle, and irrigation on enhanced weathering**".

G. Cipolla, S. Calabrese, A. Porporato, L.V. Noto

**Manuscript No.: egusphere-2022-196**

**COMMENTS FROM THE REVIEWERS: REVIEWER #1**
* * *
**General comments:**

**Referee:** *Enhanced weathering (EW) is a biogeochemical carbon sequestration strategy which is currently gaining interest in the light of climate change. This paper considers several case studies of EW in the USA and Italy, and considers important aspects of soil moisture and rainfall effects which have been largely neglected in previous studies. The authors have used an existing EW model (Cipollo et al. 2021, Adv. Water Res. **154**:103934) to determine the seasonality of leached bicarbonate and carbonate as the metric of carbon sequestration. Their weathering rates and carbon sequestration rates are within the range of other studies, but water fluxes (precipitation, evapotranspiration) reducing soil moisture greatly reduced leached carbon during the growing season.*

**Response**: We thank the Reviewer for his/her in-depth analysis of this work. Please find below our responses to the comments.
* * *
**Specific comments:**

**Referee:** *Generally, this is a good study but there are a number of omissions in the manuscript that need to be rectified. For example, no details are provided about the olivine application rate or whether the application was repeated. More detail about the tuning of the model, e.g. the background weathering rates, is required; it is unclear whether cation exchange capacity or other parameters were also tuned. The chemistry of rainfall and irrigation affecting key aspects of the model, such as the effect of soil moisture on pH and weathering, needs to be clarified. Comparison of the grain-scale weathering rates to those of Amann et al (2020) should be revisited as it seems the modelled rates are actually higher rather than similar. Some of the figures do not include enough information or could be improved by better labeling. These weaknesses should be easy for the authors to address; places in the text where these specific aspects should be addressed in the manuscript are detailed below amongst the English language and other technical corrections.*

**Response**: We thank the Reviewer for his/her very useful comments. Based on the Reviewer's suggestions, we provided more details in the manuscript, which can also be found in the responses to the comments below.
* * *
**Technical corrections:**

**Figures**

**Referee:** *As many readers may look at the figures without reading the text beforehand, it would be helpful to make it easier to understand the main elements of the story at a glance, with an indication of where more information can be found in the text. As they stand, some of the figures (especially the heatmaps) currently require careful study and references to the text to understand them.*

**Response**: Good point. We tried to make the figures clearer. Please find below our new arrangements.

**Referee:** *Figure 3. Rainfall frequency is noted as a reason why Iowa has higher weathering rates than California, and it would have been useful to see λ for the states superimposed. The top row could show λ for Padan, Sicily and IA, CA and the bottom row could show α for the same cases. This arrangement would still allow readers to compare α and λ seasonality for individual sites. In any case, please make it clear what α and λ are. Y axis labels should read "Mean rainfall α [mm]" and "Rain frequency λ" or something similar. In the caption, refer readers to section 2.2.1 for more details.*

**Response:** This is a very good suggestion. We modified the figure as follows.

[Figure]

Furthermore, we modified the caption:

*Values of the average rainfall depth (α) and frequency (λ) from January (i.e., month indicated with 1) to December (i.e., month indicated with 12). The reader is referred to Section 2.2.1 for more details.*

**Referee:** *Figure 4. Y axes labels for Kc should read "Crop coefficient Kc". In the caption, clarify that ADD is added organic carbon. Refer readers to Section 2.2.3 for more details.*

**Response:** Following this suggestion, we modified the figure:

[Figure]

Furthermore, we modified the caption:

*Seasonal variability of the crop coefficient (Kc) and the added carbon from vegetation (ADD) for a) wheat in Sicily, b) wheat in California, c) wheat in the Padan plain and Iowa, d) corn in Padan plain and Iowa and e) corn in Sicily and California. The reader is referred to Section 2.2.3 for more details.*

**Referee:** *Figure 5. It would be helpful if the labels "Corn Clay Loam" etc could appear on the left of each row of this figure, the individual panel colorbars were removed, and a single colorbar with legible label e.g. "soil moisture ratio SIA/SCA" appeared at the top of each row. Not all software makes it easy to do this but it would greatly improve the readability of the whole figure. The bottom row lefthand label would be "Average daily ratio" and the individual Y-axes labels would be e.g. "SIA/SCA". The label for column 3 should make it clear what scale is being considered, e.g. "Grain scale weathering rate ratio".*

**Response:** Figure modified as follows:

[Figure]

**Referee:** *Figure 6. Please add letter designations to the panels (a, b, c, d). The Kc panel Y axis label should say "Crop coefficient Kc" and that for the bottom left panel should say "Specific soil moisture s".*

**Response:** Thanks for highlighting this. Please find below the new version of the figure:

[Figure]

**Referee:** *Figures 7, 8, 10: Similar remarks as for Figure 5. Please add column and row labels. Room for these labels will be available if the individual colorbars are removed and master colorbars for each column appear at the top with the column labels. Also, make it clear that the weathering rates are per mineral surface area rather than per land area, e.g., mol/(m2 olivine s) in Figure 10.*

**Response:** We improved Figures 7, 8, and 10.

Figure 7:

[Figure]

Figure 8:

[Figure]

Figure 10: In this figure, we included the heatmaps regarding the Italian case studies that were previously in the supplementary materials, in order to provide a better comparison between the four selected sites.

[Figure]

**Referee:** *Figure 11. Are the samples contributing to the boxplots monthly values from the ten years of a single run, or from several runs where some parameter(s) varied? Which crops and soils are involved in these simulations; which simulations from Table 1 are included? Please clarify. Consider adding the place names to the figures so that readers can see this information at a glance.*

**Response:** Thank you for pointing this out. The boxplots of sequestered $CO_2$ are related to single runs of the four case studies, considering the single climatic conditions, along with the most frequent crop and soil type. Basically, these simulations are those referred to as "Most frequent scenario (Section 3.4)" shown in table 1. We better clarified this aspect at the beginning of Section 3.4 with this sentence:

*The results here analyzed are those related to the simulations referred to as "Most frequent scenario (Section 3.4)" as shown in table 1, where the considered climatic condition is analyzed along with the most frequent crop and soil type.*

We also clarified this aspect in the caption:

*Box plots representing the seasonality of the $CO_2$ sequestered by leaching of extra $HCO_3^-$ and $CO_3^{2-}$ produced by olivine dissolution, computed over the 10 years subsequent to olivine amendment. The plots are related to a) Wheat in clay loam soil for Sicily (W - CL - SI), b) Corn in silty clay loam soil for Padan plain (C - SCL - PP), c) Wheat in clay loam soil for California (W - CL - CA) and d) Corn in silty clay loam soil for Iowa (C - SCL – IA).*

Finally, the names of the locations were added to the panels as suggested. The new figure is the following:

[Figure]

**Introduction**

**Referee:** *line 20: allows*

**Response:** Thank you for highlighting this mistake. We corrected it.

**Referee:** *lines 20 and 21: Yes, there are many studies about olivine because it is widely distributed and has relatively fast dissolution rates. However, the sentence is a little awkward and I had to read it several times. Better to say that "many studies discuss using olivine (often modelled as the endmember forsterite Mg2SiO4) ..." Olivine is a solid solution series between forsterite (Mg2SiO4) and fayalite (Fe2SiO4), but the common ones tend to be more Mg-rich and rate laws for forsterite dissolution are freely available.*

**Response:** Good insight! We modified the sentence as follows:

*Many studies discuss using olivine (often modeled as the end-member forsterite Mg2SiO4) in EW applications...*

**Referee:** *line 31: "...(i.e., fungi and bacteria) that, ..."*

**Response:** Thank you for highlighting this mistake. We corrected it.

**Referee:** *line 42: please make it clear that the weathering rate is per square meter of mineral, not per square meter of land.*

**Response:** Thank you for this suggestion. We now clarified this aspect. The modified sentence is the following:

*The achieved weathering rates, expressed in moles of dissolved olivine per unit of specific surface of the mineral and time, were on the order of…*

**Referee:** *lines 50–52: The sentence about the models summarized by Taylor et al is a little bit difficult to understand. The models do indeed vary in their degree of complexity and plant processes may well be absent or oversimplified. A better wording might be: "The reactive transport models summarized by Taylor et al. (2017) vary in their degree of complexity and plant processes may be absent or oversimplified."*

**Response:** Thank you for your suggestion. We modified the sentence as suggested.

**2.1 Methodology**

**Referee:** *Please describe the olivine applications: one-time or annual, mass per unit area applied, specific surface area modelled, depth of soil into which the olivine is mixed. This information deserves either a subsection or, if journal guidelines and space permits, a table. The source of the weathering rate law for olivine used in the model should also be cited.*

**Response:** This is a worthwhile comment, thank you! We modeled a one-time olivine application with a rate of 10 kg m$^{-2}$ of cropland area. The olivine is assumed to be mixed for the whole active root zone depth of the considered crops. All the particles dissolve according to the same rate since the presence of preferential flow paths is not here considered. The dissolution of olivine particles is modeled according to the shrinking core model of Lasaga (1984) (https://doi.org/10.1029/JB089iB06p04009), considering particles as perfect spheres having an initial diameter of 200 μm, since the model considers a single effective diameter, defined as the mean diameter of a particle size distribution, in the name of simplicity. In fact, to consider the actual particle size distribution, the model would need to include partial differential equations, which would greatly increase the computational costs. This effective diameter is meant to be representative of the whole particle size distribution, in that its weathering rate represents the average weathering rate found by integrating over the particle size distribution.
The dissolution rate law is the one presented in Cipolla et al., (2021a), (https://doi.org/10.1016/j.advwatres.2021.103934), where the weathering rate, expressed in number of moles of dissolved olivine per unit of reactive surface of the mineral and per unit of time, is a function of soil moisture, pH and the ion activity product, which expresses the products of olivine dissolution reaction (i.e., magnesium and silicates) with respect to soil water pH. We better clarified all these aspects at the end of Section 2.1 of the manuscript. The added part is the following:

*For all the analyzed scenarios, a one-time olivine amendment with a rate of 10 kg m$^{-2}$ (i.e., 100 t ha$^{-2}$) was considered. The olivine is assumed to be mixed for the whole active root zone depth of the*

*considered crops. All the particles dissolve according to the same rate since the presence of preferential flow paths is not considered. The dissolution of olivine particles is modeled according to the shrinking core model of Lasaga (1984), considering particles as perfect spheres having an initial diameter of 200 µm, since the model considers a single effective diameter, defined as the mean diameter of a particle size distribution, in the name of simplicity. The dissolution rate law is the one presented in Cipolla et al. (2021a), where the weathering rate, expressed in number of moles of dissolved olivine per unit of reactive surface of the mineral and per unit of time, is a function of soil moisture, pH and the ion activity product, which expresses the products of olivine dissolution reaction (i.e., magnesium and silicates) with respect to soil water pH.*

**2.1.1 Soil type and composition**

**Referee:** *How was the Hartmann and Moosdorf (2012) lithological map used? Were minerals assigned to the different lithological classes and then weathered individually in the soil, or was a generic rock defined for which the rate constant (rather than the apparent surface area of the rock/mineral) was tuned to the reported pH for the soils? What stoichiometry (base cations, Al, C, Si) was assigned to the native minerals/rocks?*
*Soil properties which differ for the four sites should be tabulated, perhaps extending Table 3: CEC, mean initial pH, bedrock type from Hartmann and Moosdorf (2012).*

**Response:** Thank you for the very in-depth analysis of the work. We actually extracted the most frequent lithological class for the sites under study and look at its basic dissolution rate constant. Then, we calibrated these constants to achieve the reported steady-state pH for the analyzed soils, as explained in lines 176-181. We actually did not consider a proper stoichiometry of the bedrock since our main idea is to incorporate different possible consumptions of $H^+$ in a single term, without considering the dissolution of individual minerals (too cumbersome computationally). This is a simplified way to look at these aspects but is effective since it easily allows obtaining the typical soil pH values by calibrating a single constant.

We also extended table 3 adding the typical pH, CEC, and the achieved dissolution rate constants for the soils under study, which better represent the background weathering component. The table in its new form is reported as follows:

*Table 3 - Initial organic carbon content in the litter and humus pools ($C_0$) and the biomass pool ($C_b$) for the four sites under study. Typical values of pH, CEC and background dissolution rate constants are also reported.*

| | Sicily | Padan plain | California | Iowa |
|---|---|---|---|---|
| $C_0 [kg\ m^{-3}]$ | 13.19 | 18.19 | 19.74 | 11.58 |
| $C_b [g\ m^{-3}]$ | 131.9 | 181.9 | 197.4 | 115.8 |
| pH [-] | 7.2-8.8 | 7.2-8.8 | 7 | 6 |
| CEC [cmol $kg^{-1}$] | 50 | 50 | 35 | 25 |
| $k_{bg}$ [mol $m^{-2}\ s^{-1}$] | $10^{-5}$ | $10^{-5}$ | $10^{-6}$ | $10^{-8}$ |

**2.2.3 Crop cycle**

**Referee:** *Please clarify what the "crop coefficient" represents as soon as it is introduced, i.e., it is a*

*proportionality constant relating actual evapotranspiration to potential evapotranspiration and depends on the crop and stage of growth. Are you using the Kc and/or crop stage length values tabulated in Tables 11 and 12 of the FAO website (https://www.fao.org/3/X0490E/x0490e0b.htm), or following the procedures outlined on that website? In either case the FAO guidelines and tables deserve a proper citation.*

**Response:** Thank you for highlighting this aspect. We clarified the meaning of the crop coefficient and its use after mentioning it in the manuscript for the first time. We also cited the FAO guidelines and tables, from which we extracted the Kc and crop stage length values. The sentence is the following:

*To investigate the role of the crop cycle on the EW dynamics, the monthly variation of the crop coefficient (Kc), commonly defined as the ratio of the crop evapotranspiration over the reference evapotranspiration, was considered. It is here used as a proportionality constant relating actual evapotranspiration to potential evapotranspiration, depending on the crop and stage of growth. For each development stage, a single crop coefficient per crop type and the climatic area was obtained following FAO guidelines (tables 11 and 12 in Allen et al. 1998) (solid lines in Figure 5).*

**Referee:** *Lines 204–215: This sentence is very long and the beginning of it is awkward. It is obvious that root exudation products are connected to the vegetation cycle so this does not need to be stated. Reword as follows: "Root exudation products consist of carbon-based compounds ... (Shen et al.,2020). Their contribution ... during the initial growing stage (... all four locations). During the development phase ..."*

**Response:** Thank you for the useful suggestion. We rephrased the sentence as follows:

*Root exudation products consist of carbon-based compounds deriving from plant metabolic activity that are released from living roots (Shen et al., 2020). Their contribution to the carbon input to the soil can be modeled as a slight linear increase from the background ADD (i.e., the starting point of the ADD axis in Figure 5) to a minimum ADD value during the initial growing stage…During the development phase, it can be modeled with a more relevant growth…*

**Results**

**3.1 The role of rainfall seasonality on EW dynamics**

**Referee:** *line 239: The comma after "... the figure" is unnecessary.*

**Response:** Thank you. We removed the comma as suggested.

**Referee:** *Most of the paragraph about irrigation does not belong here; it deserves its own subsection. A bit more information would help explain why irrigation lowers the pH, and whether rainfall does the same. Are any ions being included in the irrigation water and does this differ from rainfall? Are rainwater and/or irrigation water in equilibrium with atmospheric pCO2? Is the saturation state of the olivine playing a role where soil moisture is low? Discussion of the heatmaps and the influence of the irrigation shown there can remain in Results.*

**Response:** Thank you for your useful comment. We moved the part where we explain the application of irrigation contributions (lines 242 – 257 of the old submitted version of the manuscript) to a specific subsection of the methodology section, named *2.2.2 Irrigation*. Basically, we did not consider a

specific chemistry of the irrigation water; green water (irrigation) and blue water (rainfall) lowers the pH since it increases soil moisture and wet environments are characterized by a more acidic soil water. Then, the equilibrium between soil water (which derives from rain and irrigation, when applied) and atmospheric $pCO_2$ is set in the formation of the carbonic acid ($H_2CO_3$), which is produced by the part of the dissolved $CO_2$ that reacts with water (refer to Section 2.3 of Cipolla et al., 2021a, https://doi.org/10.1016/j.advwatres.2021.103934, for more details). Answering your last question, when soil moisture is too low (i.e., close to the hygroscopic point) the olivine weathering rate is basically low because it linearly depends on soil moisture (see equation 3 in Cipolla et al., 2021a, https://doi.org/10.1016/j.advwatres.2021.103934). At that point, the saturation state of olivine is low as well given the slow dissolution of the mineral and of the consequent release of reaction products (i.e., magnesium and silicates).

**Referee:** *lines 259–260: Awkward sentence. Reword: "These considerations about soil moisture and pH affect weathering rates, which are higher in California in summer due to irrigation."*

**Response:** Thank you for the useful suggestion. We rephrased the sentence as follows:

*Weathering rate dynamics are affected by soil moisture and pH. Due to irrigation, which leads to higher soil moisture and lower pH, weathering rate is higher in California with respect to Iowa during summer.*

**Referee:** *Lines 265–266: Reword: "... Julian day 300). Combined with low transpiration during the initial growing stage, this leads to higher soil water content."*

**Response:** Thank you for the useful suggestion. We rephrased the sentence as suggested.

**Referee:** *Line 268: Edit: "... resulting from slightly higher average soil moisture and slightly lower pH."*

**Response:** Thank you for the useful suggestion. We rephrased the sentence as suggested.

**Referee:** *Lines 270–271: Reword the whole sentence: "Similar considerations apply to corn grown in Italy (Figure S1). In summer, corn requires irrigation in Sicily but not on the Padan plains. During the rest of the year, the Sicilian and Padan plains soil have similar soil moisture and the soil moisture ratio is near 1."*

**Response:** Thank you for the useful suggestion. We rephrased the sentence as suggested.

**Referee:** *Line 276: Reword: " Therefore, the Pandan soil tends to be more acidic and weathering rates tend to be higher ...).*

**Response:** Thank you for the useful suggestion. We rephrased the sentence as suggested.

**Referee:** *Line 277–279: The word "significantly" implies that a statistical test has been done but this does not seem to be the case. The sentence would also benefit from rewriting, e.g., "For the same soil and vegetation, higher rainfall leads to considerably faster olivine dissolution in Iowa than in*

*California due to higher soil moisture driven by higher seasonal rainfall frequency (λ). For the Italian case studies ...".*

**Response:** Thank you for the suggestion. The rephrased sentence is:

*For the same soil and vegetation, higher rainfall leads to an olivine dissolution considerably faster in Iowa than in California due to higher soil moisture driven by higher seasonal rainfall frequency (λ). For the Italian case studies…*

**Referee:** *Line 281: What is meant by "More relevant differences..."? Were your selected case studies not relevant in terms of MAP? Do you mean larger differences?*

**Response:** We actually meant "larger differences" in terms of MAP. We provided to replace "More relevant differences" with "larger differences" to better clarify the sentence.

**Referee:** *Line 282: Too many instances of "emphasizing"; reword as "2021b), emphasizing the effect of rainfall seaonality and climatic conditions on olivine dissolution and EW."*

**Response:** Thank you for the useful suggestion. We rephrased the sentence as suggested.

**3.2 The role of soil type on EW dynamics**

**Referee:** *As the range of soil textures for the case studies was somewhat limited, it would have been interesting to see what the model predicts with more extreme textures, such as clay and sandy soils.*

**Response:** We totally agree with the Reviewer in this statement. However, we took only these two soil types into account since they are the most frequent textures in the areas under study. We are aware that considering more extreme textures would emphasize the differences in EW dynamics due to soil types, but since we are referring all the analyses to four specific locations, we would prefer to maintain the presented structure in this manuscript. However, what is suggested by the Reviewer is certainly a useful aspect to be analyzed in a future work.

**3.3 The role of vegetation on EW dynamics**

**Referee:** *Line 311–313: Too many commas and repeated words here. Please edit: "... when either of the two crops is in the rest phase, water losses due to bare soil evaporation are similar in magnitude to transpiration for the other crop. The fact that wheat and corn cycles are not in phase ..."*

**Response:** Thank you for the insight. We rephrased the entire period as follows:

*As visible in Figure 10, when either of the two crops is in the rest phase (for example, in DOY 180-300 for wheat in Sicily and in DOY 0-100 and 250-365 for corn in Padan plain), water losses due to bare soil evaporation are similar in magnitude to transpiration for the other crop. It happens that, when corn is in the rest phase and at the same time wheat is in its initial or mid-season stage, in the first case water losses are mainly governed by bare soil evaporation, while in the other one by crop transpiration. The fact that wheat and corn cycles are not in phase does not affect much water balance and, in turn, pH and weathering rate dynamics.*

**3.4 EW case studies**

**Referee:** *Line 322: Is this a one-time application of 10kg/m2 or is it repeated annually as in some other studies? As stated above, information about the olivine treatments needs to be either tabulated or presented in Section 2.*

**Response:** Thank you for the insight. We actually moved this information to the end of Section 2.1, where an outline of the various simulation scenarios is presented. You can refer to the response to the comment related to section *2.1 Methodology* for more details about the olivine amendment.

**Referee:** *Lines 330–332: Awkward sentence. Reword: "... spring months, but in summer soil moisture is low due to high transpiration losses associated with a peak of the corn crop coefficient."*

**Response:** Thank you for the useful suggestion. We rephrased the sentence as suggested.

**Referee:** *Line 333: "... one can observe that Iowa ..."*

**Response:** Thanks for noticing the mistake. We corrected it.

**Referee:** *Line 337: The first sentence ("A similar situation ... ") looks like it belongs at the bottom of the previous paragraph as it compares the two Italian sites with similar conclusions to the two American sites. This paragraph is about comparison of Sicily with California, and Padan with Iowa.*

**Response:** Thank you for this suggestion. We split the paragraphs in a better way. Indeed, the sentence "*A similar situation can be observed from the comparison between Sicily and the Padan plain (Figure S2 of the supplementary material).*" belongs to the previous paragraph, where a comparison between Iowa and California is presented. The new paragraph, where a comparative analysis between Sicily and California and Padan plain and Iowa is reported, now starts with the sentence "*Because of the similar rainfall seasonality....*".

**Referee:** *Line 338: Remove "thus"*

**Response:** Thanks. Removed!

**Referee:** *Lines 340–341: Please reword as "... can be observed in soil pH and the order of magnitude of the olivine weathering rate."*

**Response:** Thank you for the useful suggestion. We rephrased the sentence as suggested.

**Referee:** *Line 347: Amann et al. (2020) used a Belgian soil with pH~6.6 (their table S1) which is very similar to the average annual Iowa soil pH of 6.61 given in line 335. However, the weathering rates (mol Olivine m-2 s-1) modelled for Iowa (2.13e-12 = 10-11.67), California (1.61e-12 = 10 11.79), Padan (3.17e-13 = 10-12.50), and Sicily (4.78e-13 = 10-12.32) are all at least an order of magnitude faster than Amann et al's rates (10-13.12 and 10-13.75 for coarse and fine dunite respectively). This is not necessarily a problem but the model rates do not really "reflect" those of the*

*Amann et al study. Their specific surface areas (their Table 2, m2 g-1) were measured with gas adsorption which likely overestimates the actual reactive surface area of the dunite, unfortunately they do not also present more conservative geometric surface areas. What specific surface area was used in the model? As stated above, basic information about the olivine treatments should be tabulated or described.*

**Response:** Thank you for your valuable comment! The dissolution rates we achieved here are faster than those of Amann et al., (2020), despite the similar annual rainfall rate and soil pH, although this could depend on many other factors, such as CEC and rainfall seasonality that affects the soil moisture signal. We tried to rephrase our sentence, specifying that our dissolution rates are more typical of a field environment rather than those obtained in laboratory conditions, also indicating some characteristics of the experiment presented in Amann et al., (2020), such as soil type, pH and average annual rainfall. Regarding the information about the olivine application, please refer to the response to the comment related to section *2.1 Methodology*. The modified sentence in Section 3.4 is reported below:

*The order of magnitude of weathering rates provided by our model is more typical of the field environment with respect to those achieved in laboratory conditions. Indeed, we achieved weathering rate values similar to those presented in the mesocosm experiment of Amann et al., (2020), which used a loamy sandy soil with pH equal to about 6.6 and a total amount of annual rain of 800 mm y$^{-1}$, similar to the annual average soil pH for Iowa.*

**Referee:** *Line 348: Section 2.2.2 discusses the calibration of the background weathering, but says that CEC was set based on existing CEC data (Ballabio et al. 2019 and USDA) which does not necessarily imply that CEC was calibrated; it seemed reasonable to assume that the CEC values used were simply means of CEC measurements from the cropland areas in the four regions (pink areas of the maps in Figure 2). If CEC or any other parameter of the weathering model was calibrated, please give details in Section 2.2.2. Then explain how these parameters affect the weathering fluxes and link to the rest of this paragraph comparing the Italian and US case studies.*

**Response:** Thank you for the in-depth analysis of the manuscript. As we stated in Section 2.2.2, the CEC was not calibrated, but just extracted from measurements related to the cropland areas under study. We understand this may be a bit confusing in line 348, where we stated to have calibrated both the CEC and the background weathering flux. We thus rephrased the sentence as follows, also explaining how these components affect EW dynamics:

*This aspect stresses the importance of using measurements of soil properties (e.g., CEC, pH) for calibrating the background weathering flux, allowing to obtain more realistic estimates of olivine dissolution dynamics. Indeed, CEC and background weathering strongly affect pH levels before olivine amendment and, in turn, dissolution rates which are faster under more acidic conditions (e.g., the case of Iowa that has the slowest background weathering flux).*

**Referee:** *Line 357: This sentence is ambiguous; it is not clear which study the 22 kg m-2 applies to. Amann et al. (2020) seem to have applied 22 kg dunite m-2 to their mesocosms; they said the dunite was about 90% olivine of which 92% was forsterite, so they applied 19.8 kg olivine m-2 and 18.216 kg forsterite m-2. What was the application rate here? it is never stated anywhere in this manuscript as far as I can see.*

**Response:** The experiment we cited is that of Amann et al., (2020) since about the double rate of olivine is added to the soil, with respect to our setup. We corrected the amount of forsterite added to

their mesocosm experiment to make a more reliable comparison with our results. The forsterite application rate in our setup (i.e., 10 kg m$^{-2}$) is here specified at the end of Section 2.1, along with other details of olivine application (refer to the response to the comment related to section *2.1 Methodology*). The modified sentence is the following:

*The values achieved for the US locations are lower but still comparable with those of Amann et al. (2020), that derived a sequestered CO$_2$ within the range 23 - 49 kg ha$^{-1}$ y$^{-1}$, amending more than the double olivine with respect to our study (i.e., 22 kg m$^{-2}$ of dunite corresponding to about 18 kg m$^{-2}$ of olivine) in his mesocosm experiment with conditions similar to the field environment.*

**Referee:** *line 362: Replace "correspondence" with "corresponding".*

**Response:** Thank you. We replaced the word as suggested.

**Referee:** *Line 395: Here olivine-derived CO2 leached and CO2 in soil water are distinguished, with leached CO2 being the preferred metric of carbon sequestration. If EW were rolled out on a large scale and CO2 consumption then calculated based on river water samples, then the leached DIC (dissolved inorganic carbon), alkalinity or HCO3 based on major cations, is indeed relevant rather than the chemistry of the soils. It could also be argued that DIC stored for centuries to millennia in groundwater is actually sequestered at least on those timescales. If those long flow paths comprise closed systems, the total carbon will be constant and speciation of the carbonate system will not lead to degassing. when that water eventually enters streams or rivers the likelihood of degassing due to turbulence or seasonal mixing in estuaries should be similar to waters entering at similar points on shorter flow paths. Following the carbon all the way to the ocean where it is believed to be sequestered on 105-year timescales (e.g., Renforth and Henderson 2017, Rev. Geophys. 55:636–674) is not really straightforward! In any case, it is not clear how the Cipolla et al. (2021) model calculates loss to groundwater. Please clarify.*

**Response:** Thank you for the very important comment. We totally agree with the Reviewer that assessing EW carbon sequestration is not an easy task. There is, indeed, much uncertainty in the scientific community about the path that carbonates and bicarbonates, produced by the weathering reaction, may follow on the way to the oceans where, as you stated, they are assumed to be sequestered on long timescales. We here distinguished the HCO$_3^-$ and CO$_3^{2-}$ produced by the reaction of olivine with CO$_2$ and remain dissolved in soil water from those that are leached away from the reference domain. Indeed, the former can easily react with H$^+$, making the carbonic acid (i.e., H$_2$CO$_3$) which, since it is in equilibrium between the liquid and the gas phase, can thus lead to a CO$_2$ release to soil pores and, in turn, to the atmosphere (refer to Cipolla et al., 2021a, https://doi.org/10.1016/j.advwatres.2021.103934, and Cipolla et al., 2021b, https://doi.org/10.1016/j.advwatres.2021.103949, for more details). The mass of HCO$_3^-$ and CO$_3^{2-}$ that goes to groundwater is therefore computed as the product between the leaching rate and the concentration of these ions in soil water at each time step, taking into account only the extra amount produced by weathering reaction. The boxplots reported in Figure 11 are representative of the seasonal variability of the sequestered CO$_2$ and are derived by means of 10 values (one per year in the considered time window) of leached mass of CO$_2$ per month. We better clarified this aspect in Section 3.4 of the manuscript. The added part is the following:

*In particular, for each time-step, the loss of HCO$_3^-$ and CO$_3^{2-}$ to groundwater is derived by the product of the leaching rate and the concentration of these ions in soil water, taking into account only the extra amount produced by weathering reaction.*

**Referee:** *Lines 410–413: Nitric acid weathering may well weather both olivine and carbonate rocks, resulting in loss of N2O (a potent greenhouse gas) either on site or downstream, as well as CO2 degassing in the latter case . The beginning of this sentence suggests that nitric acid is beneficial for EW. Reading the whole sentence several times, it seems this is not the intention, but rather that nitric acid is not beneficial even though it may increase olivine weathering rates. Please reword, e.g. "Even though acidification may increase olivine weathering rates, nitric acid (i.e., NHO3) from nitrogen fertilizer would react with carbonate rocks such as those comprising the bedrock in Sicily and the Padan plain, releasing CO2 to the atmosphere (reference) and reducing carbon sequestration potential." Please find another reference because Hartmann and Moosdorf (2012) do not mention nitrogen, although several other Hartmann papers do.*

**Response:** We rephrased the sentence as suggested. It was a mistake to cite Hartmann and Moosdorf (2012) here since, as you correctly state, the role of nitrogen on weathering is not mentioned. We meant to cite Hartmann et al., (2013) (https://doi.org/10.1002/rog.20004), where this aspect is clearly defined (i.e., see Figure 1).

**Referee:** *Line 414: "Despite we here considered olivine application for EW" is a bit awkward; please consider rewording, e.g.: "Even though we considered olivine application for EW here"*

**Response:** Thank you for the useful suggestion. We rephrased the sentence as suggested.

---

## Author Comment (AC2)

Replies to the comments on "**Effects of precipitation seasonality, vegetation cycle, and irrigation on enhanced weathering**".

G. Cipolla, S. Calabrese, A. Porporato, L.V. Noto

**Manuscript No.: egusphere-2022-196**

**COMMENTS FROM THE REVIEWERS: REVIEWER #2**

**General comments:**

**Referee:** *Due to the climate crisis emergency, research into enhanced weathering (EW) as a potential method for carbon dioxide removal (CDR) has increased exponentially over the past decade. Land based application of EW is thereby tested from lab over mesocosm to field scale experiments, representing increasingly more realistic conditions which are however also increasingly more complex - and require increasingly more time (from weeks-months in lab experiments to up to 10 years in field experiments). As time is of the essence when it comes to climate change mitigation, model computations of EW scenarios can play an important role in assessing the potential for CO2 sequestration under specific climate, soil and crop conditions. To achieve this, close collaboration between EW ´lab/field´ and ´computer´ based researchers is necessary to coordinate their research and continuously use insights gained from one field as new input for the research in the other field.*

*This manuscript thus represents a very relevant study on modelling the effects of rainfall seasonality, irrigation, crop growth cycle and soil type at 4 different cropland sites across the world. The complexity of the authors´ EW model and the as realistic as possible input data for most of their model´s variables make it stand out and represent an important contribution to research into CDR potential of terrestrial EW.*

*The main weakness of this manuscript is the lack of some relevant background knowledge regarding mineralogy, petrology and soil formation. This is reflected in a rather poor and unrealistic modelling of the mineralogy of the soil, and the absence of necessary information on the ´olivine´ material used as soil amendment for EW. This can, however, certainly be addressed in a revised version of the manuscript.*

*Comparison of the dissolution and CO2 sequestration rates obtained from the current model with those from the (few) published lab and field experiments could also use further discussion. It would be more valuable when the conditions (crop, olivine amendment, soil type, water availability, …), and the methods to calculate these rates, are also compared for the lab/field and model studies. Further exploration of the plausible reasons for any observed differences between lab/field and model results would also be interesting.*

*Future further improvements of this excellent EW model could be to introduce a combination of different minerals as EW source material, reflecting the reality of Ca-Mg silicate rock powders proposed for EW. A multi-mineral design of the soil´s mineral composition would also greatly benefit the computed background weathering ratio prior to EW. Using model parameter data from ongoing field experiments could be a next step to overall improve the EW model, which then in turn will provide relevant insights into real life EW experiments.*

**Response**: We thank the Reviewer for the in-depth analysis of this work. The comments provided many valuable suggestions about soil mineralogy, highlighted some aspects that needed to be explained more in detail (i.e., the characteristics of the olivine amendment), and pointed out possible future developments of the model. The multi-mineral EW represents certainly an aspect worth to be analyzed, either from the perspective of putting it into practice, or understanding which of the various minerals can provide the best carbon sequestration rates.
* * *
**Specific comments:**

**Referee:** *Below follows a list of all my comments, ordered according to the manuscript´s structure. Besides language corrections (yellow) there are "requests and suggestions to rephrase" as well more explanatory paragraphs to clarify (geological-mineralogical) concepts relevant to EW and this manuscript.*

*The changes suggested above to improve the manuscript are presented in more detail within these comments.*

**Response**: We thank again the Reviewer for the in-depth analysis of this work. Please find below our responses to the comments.

**Title:**

**Referee:** *As the title is now, it suggests that mainly rainfall seasonality, vegetation cycle and irrigation have been studied in detail to assess their effect on EW. The manuscript however also investigates the effect of having two (not too) different soil types. So perhaps include this as a fourth variable, and also point out that this is a model.*

*For example "Effects of precipitation seasonality, irrigation, vegetation cycle and soil type on enhanced weathering – Modelling of cropland case studies across four sites."*

*As pointed out above and further discussed below, the mineralogical composition of the soil used in these models is significantly less representative or realistic than the other four model parameters mentioned in the title.*

*If this is corrected in a revised version of the manuscript, 'soil type' in the above suggested title can be replaced by 'soil composition'.*

**Response**: Good point. We revised the title as: "Effects of precipitation seasonality, irrigation, vegetation cycle and soil type on enhanced weathering – Modelling of cropland case studies across four sites".

**Abstract:**

**Referee:** *- lines 9-10: ... strongly affected also by the pre-EW soil pH, which is one of the main factors controlling soil pH before olivine amendment. The same parameter is referred to her: pre-EW soil pH = soil pH before olivine amendment. After having read the rest of the manuscript, it seems that ´pre-EW soil pH´ should be replaced by ´background weathering flux´ or the ´mineral composition of the soil´, which largely determines the background weathering flux.*

**Response**: Thank you for the recommendation. We replaced "*pre-EW soil pH*" with "*background weathering flux*", which is the component of the model that mostly determines soil pH before olivine amendment.

**Referee:** *- lines 10-11: Looking at the numbers presented here for sequestered CO2, and without further explanation on the modelled rainfall seasonality, crop cycle or soil type here in the abstract, this sentence does not make so much sense. How are 4.20 and 0.62 the largest compared to 2.21 and 0.39? Do you mean to compare the two US sites with one another, and the two Italian sites with one another? After reading the manuscript I understand what is meant here, but the abstract should make sense on its own. Please rephrase to make the ´take home´ message more clear.*

**Response**: Good point! We now better clarified the sentence as suggested. You can find below the modified part of the manuscript.

*Regarding the US case studies, Iowa sequesters the greatest amount of $CO_2$ if compared to California (4.20 and 2.21 kg ha$^{-1}$ y$^{-1}$, respectively), and the same happens for Sicily with respect to the Padan plain (0.62 and 0.39 kg ha$^{-1}$ y$^{-1}$, respectively).*

**1 Introduction:**

**Referee:** *- line 20: bicarbonate*s *(as on line 21 there is also the plural carbonates)*

**Response**: Thank you for highlighting this mistake. We corrected it.

**Referee:** *- lines 21-22: please consider rephrasing "which are then leached out of the soil, transported by groundwater, and eventually reach the oceans or precipitate as carbonates" to clarify that carbonate precipitation may happen at any stage from (bi)carbonate formation in the soil to transportation into groundwater and transfers via rivers into the ocean.*

**Response**:. We rephrased the sentence as follows:

*...which can precipitate as carbonates in the soil or at any stage during their transport from land to ocean.*

**Referee:** *- lines 22-25: please rewrite/revise the sentence "Many studies... ...(Hartmann et al.,2013).'' To clarify/correct the following:*
*    - Olivine is the general name for the solid solution series between the ideal end member minerals forsterite (Mg2SiO4) and fayalite (Fe2SiO4), where the Mg richer varieties are more common and also more reactive with CO2 and H2O. (Generally, it is the Mg-Ca-silicates that have the most potential for CDR - forsterite and wollastonite CaSiO3.) So for ease of representation/calculation Mg2SiO4 is often used to represent an olivine mineral with real formula (Mg$_{1-x}$,Fe$_x$)$_2$SiO$_4$.*
*    - The mineral olivine is found in igneous rocks: whereas (1) volcanic rocks such as basalt and (2) plutonic rocks such as gabbro typically have up to ca 10-20vol% of olivine; (3) ultramafic rocks such as parts of the earth's mantle exposed on the surface in ophiolite assemblages can have much higher olivine contents up to 95%. So most 'olivine' mines across the world are quarrying ultramafic mantle rocks (for example dunite, peridotite) as they have higher olivine contents, but 'basalt' is also*

*quarried and used for EW as despite its somewhat lower olivine contents it consists of other silicates that provide plant nutrition upon dissolution.*

*(Gabbro is NOT a volcanic rock)*

**Response**: Thank you for your clarifications. The rephrased sentence is as follows:

*Many studies discuss using olivine (often modeled as the end-member forsterite $Mg_2SiO_4$ or fayalite $Fe_2SiO_4$, although the former is the most common mineral that dissolves and reacts faster with $CO_2$) in EW applications (Köhler et al., 2010; ten Berge et al., 2012). This mineral can be extracted from igneous rocks, such as volcanic (i.e., basalt), plutonic (i.e., gabbro) and mostly from ultramafic rocks, which can have up to 95 % of olivine and are widely distributed across the globe. Additionally, olivine is characterized by relatively fast dissolution rates if compared to other silicate minerals, such as albite and orthoclase (Hartmann et al., 2013).*

**Referee:** *- lines 28-29: There is indeed still a discrepancy between silicate dissolution rates observed in labs (where they are more easily measurable) and in the field (where the main challenge is to differentiate the EW signal from the other biogeochemical processes going on). However a lot of lab, mesocosm and field experiments have been carried out since the reference to this issue in White & Barley (2003). As the research field for EW as CDR method has exponentially grown in the last 1-2 decades, it seems better to provide a more recent reference on this issue.*

**Response**: We agree with the reviewer. Therefore, we replaced the "*White & Barley (2003)*" reference with a more recent one ([https://iopscience.iop.org/article/10.1088/1748-9326/aaa9c4/meta](https://iopscience.iop.org/article/10.1088/1748-9326/aaa9c4/meta)) that mentions the uncertainties of field weathering rates.

**Referee:** *- line 34: "any other Ca-Mg-silicate mineral" (see comment lines 22-25); basalt is NOT a mineral, it is a rock containing different minerals one of which can be olivine -> "such as basalt or wollastonite"*

**Response**: Good suggestion. We specified that silicate minerals containing calcium and magnesium are usually used for EW. The modified sentence is the following:

*To begin to address these uncertainties, several experimental approaches have been carried out to characterize olivine or any other Ca-Mg-silicate mineral (such as wollastonite) used for EW dissolution dynamics.*

**Referee:** *- line 35: the single-mineral particle lab experiments of dissolution you refer to here are not on Ca-Mg-silicates most often considered for EW, but instead on other silicate minerals that are relevant to natural weathering and soil formation (albite feldspar in Hellman and Tisserand, 2006; illite clay mineral in Koehler et al., 2003). Perhaps you can replace these with references to olivine, wollastonite, ... dissolution rate experiment studies which are more relevant to this study? (for example: Pokrovsky & Schott, 2000 [https://doi.org/10.1016/S0016-7037(00)00434-8](https://doi.org/10.1016/S0016-7037(00)00434-8) ; Oelkers et al 2018 https://doi.org/10.1016/j.chemgeo.2018.10.008 ...)*

**Response**: Thank you for highlighting this aspect. We replaced the references as suggested, also adding a work about an experimental setup used to extract wollastonite dissolution rates (10.1023/B:BIOG.0000015787.44175.3f).

**Referee:** - *lines 35-36: Please rewrite to clarify and correctly group the different types of EW experiments. Besides single mineral grain dissolution experiments (see above), terrestrial EW experiments can be classified in the following 3 categories: (1) Laboratory experiments involving soil cores/columns to which silicate rock powder (SRP) is added, under controlled T and irrigation conditions, without biological processes (Renforth et al. 2015, Dietzen et al., 2018). (2) Mesocosm or pot experiments where plants and/or soil organisms are added to larger containers of soil with SRP, representing more closely 'real life' conditions but still closed and controlled system (ten Berge et al. 2012, Amann et al. 2020, Kelland et al. 2020). (3) Field trials where SRP is added outdoors to a field, grassland, forest soil representing complex open system of real life conditions (published study results so far only with wollastonite: Haque et al. 2020, Taylor et al, 2021 https://doi.org/10.5194/bg-18-169-2021 ).*

**Response**: Thank you. We better distinguished the state-of-the-art of EW experiment. The adjusted part of the manuscript is the following:

*These are mainly based on laboratory experiments conducted on single mineral particles (Oelkers et al., 2018; Pokrovsky and Schott, 2000; Peters et al., 2004), laboratory experiments involving soil cores/columns amended with silicate rock powder (SRP), under controlled temperature and irrigation conditions, without biological processes (Renforth et al., 2015; Dietzen et al., 2018), mesocosm or pot experiments where plants and/or soil organisms are added to larger containers of soil with SRP, representing more closely 'real life' conditions but still closed and controlled system (ten Berge et al., 2012; Amann et al., 2020; Kelland et al., 2020) and field trials where SRP is added outdoors to a field, grassland or forest soil representing complex open system of real life conditions (Taylor et al. (2021); Haque et al. (2020) using wollastonite).*

**Referee:** - *line 41: "magnesium and* silica *concentrations"*

**Response**: Thank you for highlighting this mistake. We corrected it.

**Referee:** - *lines 42-43: Please clarify that weathering rates of 10-13 mol/(m2.s) refers to the surface of the mineral grain in contrast to sequestration rates in kgCO2/(ha.year) which refers to land surface on which mineral dust is spread on.*

**Response**: Good point. We modified this part of the manuscript as follows:

*The achieved weathering rates, expressed in moles of dissolved olivine per unit of specific surface of the mineral and time, were on the order of $10^{-13}$ mol m$^{-2}$ s$^{-1}$ corresponding to carbon sequestration rates of 23 and 49 $kg_{CO2}$ ha$^{-1}$ y$^{-1}$, where ha refers to land surface over which mineral dust is spread.*

**Referee:** - *line 46: Although White and Brantley (2003) indeed compare field and laboratory observed dissolution rates, the subject of this study is natural weathering of plagioclase and other non Ca- Mg-silicates present in a granite. Could you perhaps find more recent references pointing out the discrepancy of lab, mesocosm and field derived dissolution rates of Ca-Mg silicate minerals relevant for enhanced weathering?*

**Response**: Thank you. We added a reference ([https://doi.org/10.1016/j.gca.2014.10.013](https://doi.org/10.1016/j.gca.2014.10.013)) describing the gap between laboratory and field weathering rates for albite, which is a silicate mineral containing calcium, also used for EW applications. However, we prefer to cite "*White and Brantley (2003)*" in this part of the manuscript since it makes a very clear distinction between intrinsic (e.g., shape and roughness of mineral surface particles) and extrinsic (e.g., pH, temperature and soil water content) factors responsible for the differences between lab and field weathering rates.

**Referee:** *- line 60: "suggesting that the model estimates approach a condition that is more similar to what happens in the field" (mesocosm experiments still do not represent the full complexity of field trials)*

**Response**: Thank you for highlighting this aspect. We modified the sentence in the manuscript as follows:

*By introducing stochasticity in rainfall and connecting ecohydrological with biogeochemical processes, the model presented in Cipolla et al. (2021a) leads to carbon sequestration rates of the same order of magnitude as those in the mesocosm experiment of Amann et al. (2020), that represents conditions similar to those in the field, despite not being in the full extent of their complexities.*

**Referee:** *- lines 63-64: "Many of the model components are characterized on the base of measurements (i.e. pH and cation exchange)" perhaps better formulated as "Many of the model parameters are obtained from measurements"?*

**Response**: Good suggestion. We modified the sentence as indicated.

**Referee:** *- line 66: The acronym "MAPs" is used here without introducing/explaining it.*

**Response**: Thank you for raising this aspect. "MAPs" was a mistake. We replaced it with "MAP", which stands for "Mean Annual Precipitation".

**2.1 Methodology:**

**Referee:** *- line 88: Long sentence which might be more easily readable by splitting as "… to which we refer for details. It links ecohydrological and …"*

**Response**: We split the sentence as suggested.

**Referee:** *- line 90: "The model is composed of four closely related components." After reading this paragraph a number of times it is not clear to me which one are these four. Could you please sum them up here, or number them in the following description?*

**Response**: The four components are presented as i) "Organic matter", ii) "DIC system", iii) "CEC" and iv) "Dissolved minerals" in Cipolla et al., (2021a) (https://doi.org/10.1016/j.advwatres.2021.103934). Since here only a short description of the model is provided, we referred the reader to this manuscript for more details.

**Referee:** *- line 93: "... of soil water* ==ions== *released by olivine dissolution..." as you refer both to silicates which are anions and magnesium which forms cations*

**Response**: Thank you for highlighting this mistake. We corrected it.

**Referee:** *- line 94: Mg2+ can be removed here as base cation as it is already referred to as one of the main ions formed upon olivine dissolution*

**Response**: Good point! We removed it.

**Referee:** *- line 95: "the ... (CEC) accounts for the process between": which process?*

**Response**: Thank you for raising this unclear aspect. We were referred to the adsorption process, which is now clearly expressed in this sentence.

**2.2 Study areas and data:**

**Referee:** *- line 116: there seems to be a mistake with the web link: there is a ´c´ in subscript https://ipad.fas.usda.gov/rssiws/al/globalcropprod.aspx and I get an error message when trying this link*

**Response**: Thank you. It was a LaTeX typo. We corrected it.

**Referee:** *- line119: "active root zone depth of the involved crop." Could you please already here write these specific depths chosen for the corn and the wheat crops in the models?*

**Response**: We specified in this part of the manuscript the active root zone depths of the considered crops. The modified version is the following:

*As in Cipolla et al. (2021b), all simulations are related to a unit ground area of homogeneous soil, vegetation and rainfall characteristics, vertically delimited by the active root zone depth of the involved crop, i.e., 40 and 60 cm for the corn and wheat, respectively (Fan et al., 2016).*

**2.2.1 Rainfall seasonality:**

**Referee:** *- line 123: The acronym MAP is used again without writing it out in full before*

**Response**: Replying to the comment related to line 66, we introduced the explanation of the acronym MAP before this point of the manuscript. So, the reader at this point can understand its meaning.

**Referee:** *- line 126: Since acronyms SIAS and USGS in the previous and next line, respectively, are fully written out perhaps this might also be done here for the acronym ARPA.*

**Response**: That is correct. We extensively wrote what the acronym ARPA stands for.

**Referee:** *- lines 133-134: "… are two months out of phase, …" If ´out of phase´ refers to different trends for α and λ, one increasing and the other decreasing, it seems to me this happens in more than 2 months (from 2 through to 6, and from about 9 to 11).*

**Response**: We actually meant that the trend of α presents a time shift of about two months with respect to the trend of λ. We better clarified this aspect in this way:

*For the latter, the monthly time series of the two parameters are shifted by about two months…*

**2.2.2 Soil type and composition:**

**Referee:** *- line 151: SOC estimations are derived from the GSOCmap which represents "organic carbon content of the first 30 cm soil layer" – are the retrieved C0 and Cb values in the model applied only to the top 30 cm, or also further down to the root depths of 40 cm and 60 cm for corn and wheat, respectively?*

**Response**: This is a very valuable consideration! We actually derived $C_0$ and $C_b$ values from the GSOC map, so they are related to the first 30 cm soil layer. However, the application domain is characterized by a unit surface area and a depth equal to the active root zone, which is higher with respect to the carbon data availability, for both the considered crops. Despite we are fully aware that soil organic carbon is not homogeneously distributed, we assumed, because of limited data availability, that the carbon stock over the first 30 cm is distributed over the whole 40-60 cm rooting depth.

**Referee:** *- line 163: … consume H+ ions…*

**Response**: Thank you for highlighting this typo. We corrected it.

**Referee:** *- line 165: "existing bedrock. This last information was extracted from the lithological map presented in Hartmann and Moosdorf (2012)." Although a very valuable publication, it is too general to derive soil mineralogical input data for these 4 respective regions in comparison to the rainfall input data that are carefully derived from real meteorological measurements at these locations. The here used mineralogical/background lithological input data would compare to using the most common meteorological pattern in south Europe, west and central USA. So either acknowledge that the input data for the soil mineralogy of the four sites is much less representative for the real locations than the rainfall data. Or try to find more accurate data for the local geology of these four areas.*

*In case of the latter, please take into account that soils in plains retrieved a lot of their minerals from the weathering of surrounding mountains and might hence not only reflect the local bedrock of the plain but also the mineralogical composition of surrounding mountains. Furthermore, weathering of bedrock and surrounding rocks creates new minerals that end up in the soil. Eventually, the most accurate model input for the mineralogical composition of a soil is obtained from XRD measurements of that soil.*

**Response**: Thank you for this comment. Even if not well described in the original version of the manuscript, we are aware that soil mineralogical input data are not as representative as meteorological input data for the four analyzed locations. Indeed, our aim is not to describe an exact location (with specific coordinates), but is more devoted to describe a generic geographical area. For this reason, the background component of the model, which affects the baseline soil pH (i.e., before olivine amendment), is characterized by calibrating the background dissolution rate constants on the base of pH measurements. This has also been carried out to incorporate other factors contributing to the consume of H$^+$ (i.e., the action of microorganism or other less present minerals). As can be seen in Section 2.2.2, at the end of the calibration of the background weathering flux, we achieved dissolution rate constants values very different to those typical of calcite and quartz minerals, that mainly compose carbonate and siliciclastic sedimentary rocks, respectively. The availability of pH data and the possible calibration of the background weathering component are therefore the main reasons why we decided to use these "raw" data to describe the mineralogical characteristics of the soils under study. To make this clear in the manuscript, we modified this part in the following way:

*...This weathering flux can be estimated on the basis of the mineral composition of the soil and the type of the existing bedrock, but also depends on the action of various other factors that consume H$^+$. As a preliminary indication of the mineralogical composition of the soil, the lithological map presented in Hartmann and Moosdorf (2012) was used to extract the nature of bedrock at the cropland areas for the four considered sites. Sicily and the Padan plain are prevalently characterized by carbonate sedimentary rocks (e.g., limestone, dolostone mainly composed of carbonate minerals, such as calcite or dolomite), while the other two sites in the USA mainly present siliciclastic sedimentary rocks (e.g., sandstone, conglomerate mainly composed of silicate minerals, such as quartz or feldspars)....*

*...*

*However, soil pH depends on other factors that are not considered in the EW model (i.e., the presence of fertilizers, the action of microbes, fungi and bacteria or the action of other minerals that may release or take up H$^+$ ions).*

**Referee:** *- lines 168-170: Carbonate sedimentary rocks are NOT calcite which is a mineral – carbonate sedimentary rocks (e.g. limestone, dolostone, ...) are mainly composed of carbonate minerals (e.g. calcite, dolomite, ...). Siliciclastic sedimentary rocks are NOT quartz which is a mineral - siliciclastic sedimentary rocks (e.g. sandstone, conglomerate, siltstone, shales, breccia, ...) are mainly composed of silicate minerals (e.g. quartz, feldspars, micas, clay minerals, ...). Please correct this by rewriting the sentence.*

**Response**: Thank you for highlighting this aspect. We corrected the sentence of the manuscript as indicated:

*Sicily and the Padan plain are prevalently characterized by carbonate sedimentary rocks (e.g., limestone, dolostone mainly composed of carbonate minerals, such as calcite or dolomite), while the other two sites in the USA mainly present siliciclastic sedimentary rocks (e.g., sandstone, conglomerate mainly composed of silicate minerals, such as quartz or feldspars).*

**Referee:** *- line 170: "Lasaga (1984) and 44 (1979)" Please correct the later reference and perhaps also check more recent references on dissolution rate constants for carbonate and siliciclastic rocks.*

**Response**: Thank you for highlighting this typo. We corrected it in this way:

*...considering Lasaga (1984) and Plummer et al. (1979),...*

**Referee:** *- line 171: ...”calcite and quartz minerals…” It seem from the text that just these two minerals are used for the modelling of background soil weathering, calcite for the Italian sites and quartz for the US sites? If so, please do mention that this is a big simplification of the real soil´s mineralogy which is highly unlikely to exist only of calcite or only of quartz. In case a more accurate estimation of the soil mineralogy is used, a combination of mineral dissolution rate constants of the main occurring minerals should be taken into account.*

**Response**: As in the response to the comment related to *line 165*, we are aware that this is a simplistic view of the mineralogical characteristics of the soil under study. Indeed, we only initially looked at the type of bedrock to have an idea of the order of magnitude of the background dissolution rate constants; then, we calibrated these parameters based on soil pH data. Please refer to the response to the comment related to *line 165* for the modified part of the manuscript.

**2.2.3 Crop cycle:**

**Referee:** *- line 184: What does FAO stand for? Reference please?*

**Response**: We added the meaning of FAO and the reference related to the crop coefficient values and length of the corresponding stages. The modified part of the manuscript is the following:

*For each development stage, a single crop coefficient per crop type and the climatic area was obtained following the Food and Agriculture Organization (FAO) guidelines (tables 11 and 12 in Allen et al., 1998).*

**Referee:** *- lines 190-198: When introducing these important computation calculations (1) and (2), please clarify all the different variables in them, as was done for the next computation calculation (3). For example Crop transpiration loss T(s) where s refers to varying soil moisture - refer to Table 2 – and Bare soil evaporation E(s) where s represents…*

**Response**: We specified the meaning of all the variables in equations (1) and (2) as suggested. Below you can find the modified part of the manuscript:

*The effects of the seasonal pattern of the crop coefficient on transpiration losses, T(s), were computed as,*

*eq(1)*

*where s refers to varying soil moisture, $s_w$ is soil moisture at wilting point, while $s^*$ is soil moisture at the incipient stress…*

*The bare soil evaporation, E(s), is evaluated as,*

*eq(2)*

*where $s_h$ is soil moisture at the hygroscopic point and $s_{fc}$ is soil moisture at the field capacity.*

**Referee:** *- lines 208-218: The details on the crop cycle´s different stages and their length at each of the four sites and for each specific crop is better represented in a figure/graph than written out in detail here, introduced in the first paragraph (183-189). For example with an horizontal axis representing the year and a vertical axis which reflects different sites and crops, showing horizontal bars divided in blocks which represent the different stages, having number of length in days inside and a specific colour for each of the specific crop cycle stage.*

**Response**: Thank you for your comment. We believe that the information regarding the crop cycle´s different stages and lengths can be easily found in tables 11 and 12 in Allen et al. (1998). For this reason, dedicating an entire figure of the manuscript to this aspect may be redundant. We believe that how this aspect is presented in the current version, and also looking at Figure 4 of the manuscript, the reader can get a complete picture of the seasonal variability of the crop coefficient and the length of different stages.

**3 Results:**

**Referee:** *- lines 226-228: Rainfall seasonality, soil type, crop phenology and soil composition are correctly mentioned as some of the factors mostly affecting EW dynamics. And parameter input data of these variables for the model calculations are carefully determined based on real life data from the 4 sites. Except when it comes to the soil's mineral composition, where general, non-sitespecific and somewhat unrealistic mineral assemblages are used (quartz for the US sites and calcite for the Italian sites) to derive background weathering fluxes. Please either clearly state that these parameter input values are less site accurate than the other ones. Or better find more accurate mineral assemblages typical for each of the four sites and use these to calculate a background weathering flux based on each mineral's relative presence and dissolution rate constant.*

**Response**: As in the response to the comment related to *line 165*, we are aware that this is a simplistic view of the mineralogical characteristics of the soil under study, since we are not describing a specific location with its own coordinates but rather a generic geographical area. We stated in Section 2.2.2 that the considered mineralogical characteristics of the soils under study were derived in a simplistic way, but this was done given that the dissolution rate constants of the background weathering were then calibrated on the basis of pH data. Please refer to response to the comment related to *line 165* for more details.

**Referee:** *- line 229: Another major control factor of EW dynamics is the silicate rock dust powder (SRP) applied for EW. Its mineral composition greatly determines CO2 sequestration potential (for example whether it is mostly olivine in ultramafic matle rocks, or olivine along with feldspars andvolcanic glass in basalt). CO2 sequestration potential is also influenced by how much the SRP's mineralogy differs from the soil mineralogy (see Swoboda et al, 2022 - 10.1016/j.scitotenv.2021.150976 ).*

*In general, information seems to be missing on the 'olivine amendment' that is used in these EW models. It seems that the same imaginary 100% forsterite rock dust is used across the four sites, keeping this input parameter simple and the same everywhere to allow investigation of the effects of rainfall seasonality (with/without irrigation), soil type and composition, and crop cycle – which is*

*the main aim of this study? Or is real olivine rich rock dust modelled, for example the one used in the mesocosm experiments of Amann et al. (2020) to which results the outcome of these models are compared?*

*Besides the mineralogical composition of the applied silicate rock dust powder there is other information that is important to better compare the model results to insights from field scale experiments: what is the grainsize of the rock dust? How much of it is applied per m²? Is it left on top or worked into the soil? If the latter, to which depth is it mixed with the soil? Is this application repeated annually throughout the 10 years, or is it a one time application? These SRP parameters also have an important influence on EW dynamics (see Swoboda et al, 2022) and are therefore usually well defined in lab, pot/mesocosm or field experiments. So in order to allow better comparison of EW models and EW field trials, as well as better communication between the scientists carrying out these two kinds of studies, please also include this information as a separate subsection of 2.2 Study areas and data, for example "2.2.4 olivine amendment".*

**Response**: Thank you for the very in-depth comment. For the presented analysis, we considered the same 100% forsterite rock dust since, as you correctly affirm, the main scope of our work is to explore the role of different factors (i.e., rainfall seasonality and irrigation, soil type and composition and crop cycle) on EW dynamics.

In particular, we modeled a one-time olivine amendment with a rate of 10 kg m$^{-2}$ of cropland area. The olivine is assumed to be mixed for the whole active root zone depth of the considered crops. All the particles dissolve according to the same rate since the presence of preferential flow paths is not considered. The dissolution of olivine particles is modeled according to the shrinking core model of Lasaga (1984) (https://doi.org/10.1029/JB089iB06p04009), considering particles as perfect spheres having an initial diameter of 200 μm, since the model considers a single effective diameter, defined as the mean diameter of a particle size distribution, in the name of simplicity. In fact, to consider the actual particle size distribution, the model would need to include partial differential equations, which would greatly increase the computational costs. This effective diameter is meant to be representative of the whole particle size distribution, in that its weathering rate represents the average weathering rate found by integrating over the particle size distribution.

The dissolution rate law is the one presented in Cipolla et al., (2021a), (https://doi.org/10.1016/j.advwatres.2021.103934), where the weathering rate, expressed in number of moles of dissolved olivine per unit of reactive surface of the mineral and per unit of time, is a function of soil moisture, pH and the ion activity product, which expresses the products of olivine dissolution reaction (i.e., magnesium and silicates) with respect to soil water pH. We better clarified all these aspects at the end of Section 2.1 of the manuscript. The added part is the following:

*For all the analyzed scenarios, a one-time olivine amendment with a rate of 10 kg m$^{-2}$ (i.e., 100 t ha$^{-2}$) was considered. The olivine is assumed to be mixed for the whole active root zone depth of the considered crops. All the particles dissolve according to the same rate since the presence of preferential flow paths is not considered. The dissolution of olivine particles is modeled according to the shrinking core model of Lasaga (1984), considering particles as perfect spheres having an initial diameter of 200 μm, since the model considers a single effective diameter, defined as the mean diameter of a particle size distribution, in the name of simplicity. The dissolution rate law is the one presented in Cipolla et al. (2021a), where the weathering rate, expressed in number of moles of dissolved olivine per unit of reactive surface of the mineral and per unit of time, is a function of soil moisture, pH and the ion activity product, which expresses the products of olivine dissolution reaction (i.e., magnesium and silicates) with respect to soil water pH.*

**3.1 The role of rainfall seasonality ==and irrigation== on EW dynamics**

**Referee:** - *line 235: "…between soil moisture (S), pH and weathering rate (Wr) achieved…"*

**Response**: We modified the title of Section 3.1 and added the symbols of the considered variables as suggested.

**Referee:** - *line 236: Before describing the top 4 rows with heat panels, it would be helpful for scientists not familiar with such diagrams to shortly describe how to interpret them. For example, blue colours indicate higher values for a parameter (soil moisture, pH, weathering rate) in California than in Iowa at a specific time and under specific crop and soil conditions. Red colours indicate that at a given circumstances of soil, crop type and rainfall seasonality the soil moisture, pH or weathering rate is higher in Iowa than in California.*

**Response**: This is a good suggestion. We added what the blue and red colors represent in the heatmaps.

**Referee:** - *line 240: Is it necessary to use the computation term 'Julian day' here as the model output data are shown horizontally as a year from day 0 to day 365, so one could say "from day 150 through to about day 250" which is more easily understandable for non-modelling scientists? If 'Julian day' needs to be mentioned perhaps shortly explain what exactly this means?*

**Response**: That is correct. Given that results are always expressed at each year from day 0 to day 365, we do not need to specify "Julian" day. We replaced it with DOY (Day Of the Year) in the text.

**Referee:** - *line 241: As before, Julian day needed or is "some days around day 300" also ok?*

**Response**: The response to this comment follows the one to the comment related to *line 240*.

**Referee:** - *line 254: Soil moisture time-series in the figure 6 caption is referred to as panel b), not c)*

**Response**: Thank you for highlighting this mistake. We corrected it.

**Referee:** - *line 255: Please rewrite as "the field capacity in the days from about day 100 up to day 250".*

**Response**: Done, thank you.

**Referee:** - *lines 242-257 until "… is provided.": This paragraph introducing irrigation for the Mediterranean climates – the reason why it is necessary and how it is implemented in the model – should be moved to '2.2 Study area and data' as a new subsection right after 2.2.1 Rainfall seasonality. So 2.2.2 Irrigation, 2.2.3 Soil type and composition, 2.2.4 Crop cycle, 2.2.5 Olivine amendment. Figure 6 should then also be moved to this earlier section of the paper. The stress avoidance irrigation procedure for corn planted in Sicily should also be shown in 2.2.2 Irrigation for one of the two soil types, either added to Figure 6 or as a new Figure.*

**Response**: Thank you for your useful comment. We moved the part where we explain the application of irrigation contributions (lines 242 – 257 of the old submitted version of the manuscript), along with Figure 6 that became Figure 4, to a specific subsection of the methodology section, named *2.2.2 Irrigation*. Discussions on the effects of irrigation contributions on EW dynamics (i.e., heatmaps in Figure 5 of the old submitted version) remained in Section 3.1 instead.

**Referee:** *- line 240: When the irrigation paragraph is moved to an earlier section, you can then refer back to it here "soil moisture is higher in California than in Iowa* due to irrigation*".*

**Response**: Right! Thank you.

**Referee:** *- line 259: What is the reason that the soil pH becomes lower, more acid, with increased soil moisture, irrigation? Please briefly clarify.*

**Response**: Good point! In general, the presence of a high soil water content is certainly due to greater rainfall and/or irrigation contributions. If you look, for instance, at equation (21) in Cipolla et al., (2021a) (https://doi.org/10.1016/j.advwatres.2021.103934), you can notice that higher precipitation leads to a higher input of $H^+$ since rain is characterized by a slightly acidic pH (about 5.6). Furthermore, a greater soil water availability leads to a greater transpiration rate, which reflects into a higher nutrient cations uptake by plants (i.e., $Mg^{2+}$, $Ca^{2+}$ and $K^+$). This is translated in a higher input of $H^+$ by plants, given that they tend to maintain a neutral charge.

**Referee:** *- line 261: Please be consistent, in line 255 'Julian' was omitted when describing the period from day 100 to day 250. So perhaps generally remove the word 'Julian' throughout this document.*

**Response**: We replaced the word "Julian" with DOY, as indicated before.

**Referee:** *- line 263: Please replace "the Julian day 300" with "the 300th day" or "around day 300".*

**Response**: We replaced the word "Julian" with DOY, as indicated before.

**Referee:** *- lines 267-269: In the concluding sentence "On average, .... weathering rates derived for Iowa are about seven times higher than those in California…" This refers to the cases where wheat is the crop so please clarify this by adding "with wheat". Likewise it might be beneficial to repeat once again in the conclusions of the previous paragraph, lines 259-261, that these are model observations with corn.*
*Also: Where is this 7X higher weathering rate for Iowa compared to California derived from?*
*The average daily ratio of Wr in Figure 5? Please clarify where this number comes from.*

**Response**: We better clarified that, in lines 267-269, we were referring to wheat and that the seven times higher weathering rate for Iowa with respect to California comes from averaging the grain scale weathering rate ratio. At the same time, we specified in lines 259-261 that the results described here are related to corn. The modified part is the following:

Lines 267-269:

*Averaging the grain scale weathering rate ratio achieved considering wheat, over the considered 10 years, weathering rates derived for Iowa are about seven times higher than those in California under the two considered soil types, resulting from slightly higher average soil moisture and slightly lower pH.*

Lines 259-261:

*Due to irrigation, which leads to higher soil moisture and lower pH, weathering rate is higher in California during summer. The average daily weathering rate ratio with corn assumes values higher than one…*

**Referee:** *- lines 270-276 where the role of rainfall seasonality on EW dynamics is discussed for the Italian sites: It is unclear why the time-series heat map for the Italian sites is put as Supplementary material as despite the similarities with the US sites with/without irrigation, these maps are sufficiently different. Supplementary material is often a separate document from the main paper containing raw data, so it would be better if this Figure S1 would become the second figure in the subsection 3.1 after the time-series heat maps for the US sites. The explanation written in the Supplementary material along with Figure S1 is the exact same text as what is described here in this section, showing that text and figure best go together (in the main paper).*

**Response**: We originally put this figure in the supplementary material for the sake of length of the manuscript. However, as you affirm, it makes sense to add it within the main paper since it describes different results with respect to the US case studies. We therefore provided to add it to the main paper, along with the related explanations of results.

**Referee:** *- lines 270-271: Please rephrase this sentence as it is awkward to read and not very clear.*

**Response**: We rephrased the sentence as follows:

*Similar considerations apply to corn grown in Italy (Figure S1 of the supplementary material). In summer, corn requires irrigation in Sicily, given the scarcity of precipitation, but not on the Padan plain.*

**Referee:** *- lines 275-276: Please rephrase this sentence as it is awkward to read and not very clear.*

**Response**: We rephrased the sentence as follows:

*For the rest of the year, the weathering rate ratio between the Padan plain and Sicily, tends to be slightly less than 1, translating to slightly more favorable olivine dissolution dynamics in Sicily.*

**Referee:** *- lines 278-279: Please rephrase/rewrite these important conclusions regarding the modelled effect of rainfall seasonality and irrigation on EW dissolution rates as the text is difficult to read and unclear. Thereby keep in mind to replace 'significantly' with 'distinctly' (significantly usually refers to statistically verified differences between values).*

**Response**: We rephrased the sentence as follows and replaced "significantly" with "considerably":

*For the same soil and vegetation, higher rainfall leads to an olivine dissolution considerably faster in Iowa than in California due to higher soil moisture driven by higher seasonal rainfall frequency (λ). For the Italian case studies, rainfall seasonality leads to small differences in EW dynamics, given the similar distribution of precipitation across the year.*

**Referee:** *- line 281-283: "Larger differences in mean annual precipitation would likely result in bigger changes of EW dynamics (Cipolla et al., 2021b), emphasizing the important effect of rainfall seasonality and climatic conditions on olivine dissolution and EW."*

**Response**: We rephrased the sentence as indicated.

**3.2 The role of soil type on EW dynamics**

**Referee:** *- line 286: ... and silty clay loam soil,...*

**Response**: Word "soil" added!

**Referee:** *- lines 288-290: Add the parameter symbols please: ...soil moisture (S), pH and weathering rate (Wr) ... clay loam soil (CL) ... silty clay loam soil (SCL).*

**Response**: We added the parameters symbols as suggested (lower case *s* letter for soil moisture).

**Referee:** *- lines 294-295: ...weathering rates obtained with the clay loam soil tends tend to be about twice as high as those obtained with the silty clay loam soil... Where is this 2X higher weathering rate for CL compared to SCL derived from? The average daily ratio of Wr in Figure 7? Please clarify where this number comes from.*

**Response**: We better clarified that, in lines 294-295, the two times higher weathering rate for CL with respect to SCL soil comes from averaging the grain scale weathering rate ratio. The modified part is the following:

Lines 294-295:

*Apart from some spikes, occurring on some specific days, averaging the grain scale weathering rate ratio, we achieved that the clay loam soil results in a weathering rate about twice as high as what is obtained with the silty clay loam soil, at all four locations.*

**3.3 The role of vegetation on EW dynamics**

**Referee:** *- line 300: ... of $H^+$ to balance ...*

**Response**: Corrected!

**Referee:** *- lines 301-302: "Brady, 2017). Vegetation furthermore provides the organic matter that, once decomposed, is one of the CO2 sources in the soil system..."*

**Response**: Sentence rephrased.

**Referee:** *- line 305: … about four times higher than … for wheat… Where is this 4X higher weathering rate for CL compared to SCL derived from? The average daily ratio of Wr in Figure 8? Please clarify where this number comes from.*

**Response**: We better clarified that, in line 305, the four times higher weathering rate for corn with respect to wheat, if planted in a clay loam soil in Sicily and California, comes from averaging the grain scale weathering rate ratio. The new part of the manuscript is:

*Looking at the panels in Figure 9, averaging the grain scale weathering rate ratio, it is evident that corn leads to a weathering rate, on average, about four times higher than the one achieved for wheat when planted on clay loam soil in Sicily and California…*

**Referee:** *- line 306-307: " and fourth row of the figure). When both crops are planted in a silty clay loam soil in the Padan plain and Iowa (second and third row of the figure), the olivine dissolution dynamics are very similar. An annual average weathering rate daily ratio equal to about 1.5 might reflect slightly higher weathering rates for corn."*

**Response**: Sentence rephrased as suggested.

**Referee:** *- lines 311-312: ' when any of the two crops is in the rest phase' please specify which exact periods these are to make it easier to spot them in Figure 9. For example by: '… in the rest phase (from about day aa to day bb for wheat and from about day xx to day yy for corn)'*

**Response**: Good suggestion. We rephrased the sentence adding, for example, the rest days for wheat and corn crop in California and Iowa.

*As visible in Figure 10, when either of the two crops is in the rest phase (for example, in DOY 180-300 for wheat in Sicily and in DOY 0-100 and 250-365 for corn in Padan plain), water losses due to bare soil evaporation are similar in magnitude to transpiration for the other crop.*

**3.4 EW case studies**

**Referee:** *- lines 320-321: 'The time dynamics of soil moisture, pH and weathering rate across the four locations in Italy and the USA are shown in Figure 10. In all scenarios…'*

**Response**: Sentence rephrased.

**Referee:** *The time series heat maps for the Italian sites now in Supplementary material Figure S2 should be brought to this section of the main text to illustrate it. As Figures 5, 7, 8 each have 4 rows of three heat maps and an extra bottom row with the average daily ratio, it should be possible to add the Italian heat maps in Figure S2 to those of the US sites in Figure 10.*

**Response**:. We modified Figure 10 as suggested. You can find the new version of the figure below in your comment related to *Figure 10.*

**Referee:** *- line 322: The information regarding the olivine application rate should already have been given in a subsection of section 2.2 on olivine amendment. Why was this rather high application rate of 10kg/m² chosen? Practically, farmers apply lime and other rock dusts annually at a rate of 1-4 tons/ha.*

**Response**: We moved the olivine amendment information at the end of Section 2.1 of the manuscript, as already suggested by the Reviewer. We have chosen this slightly higher application rate to get a relevant signal of olivine application. We also considered the application rate in the experiment of Renforth et al., (2015) (https://doi.org/10.1016/j.apgeochem.2015.05.016), in which 100 g of olivine are added to a soil sample within a cylindrical pot characterized by a diameter of 10 cm. This amount of olivine corresponds to about 13 kg/m$^2$, higher than our amendment rate.

**Referee:** *- line 324: ...(i.e., before day 100) ... (i.e., from day 300 onwards)...*

**Response**: "Julian day" replaced with "DOY".

**Referee:** *- line 326: values from day 100 to about day 250 mainly ...*

**Response**: "Julian day" replaced with "DOY".

**Referee:** *- lines 333-336: Where can the annual average values for soil moisture, pH and weathering rate for Iowa and California be found? The start of the sentence with 'Comparing the annual average values…, one can observe…' suggests that this can be seen in a figure or table? If these data are only presented here within this paragraph, then please rephrase. "Annual average values of the three variables calculated for California and Iowa suggest that faster olivine dissolution occurs at the latter site (2.13X10-12mol/m²s) than at the former (1.61X10-12mol/m²s). This is in accordance with a lower annual average pH (6.61 in Iowa and 7.03 in California) and higher mean annual soil moisture (0.62 in Iowa and 0.57 in California). "*

*Whereas pH seems indeed different between the two US sites, soil moisture shows a smaller difference. How meaningful is the difference between the Iowa and California olivine dissolution rates? Any estimation of the uncertainty on these calculated values?*

**Response**: Thank you for suggesting a way to rephrase the sentence.

Regarding your second point, the main difference in weathering rate between Iowa and California is due to the background weathering flux, which is greater in California with respect to Iowa. For this reason, olivine is amended in a more acidic soil in Iowa and its dissolution is faster.

**Referee:** *- line 337: Please add the heat maps of Figure S2 to Figure 10 and add the description of them which is currently in S2 here in the main text of the manuscript. "A similar situation is observed from the comparison of the two Italian sites as Sicily and Padan plain present only small differences*

*in terms of the seasonality of soil moisture, pH, and , in turn, weathering rate. ... (i.e., before day 110) and the last (i.e., from day 300 onwards) ...==with== respect to the two sites in the USA."*

**Response**: We added the heatmaps related to the Italian case studies to Figure 10 and their description was moved to the main text. The added part to the manuscript, that in the previous version was in the supplementary material, is the following:

*A similar situation can be observed by comparing the two Italian sites as Sicily and Padan plain, which present only small differences in terms of the seasonality of soil moisture, pH, and, in turn, weathering rate. The highest soil moisture values for Sicily occur in the first (i.e., before DOY 100) and the last (i.e., from DOY 300 onwards) part of the year since, during those days, the greatest part of the total annual rainfall occurs. Low soil moisture values from the DOY 100 to about 250 are due, as in California, to the scarcity of precipitation in this period.*

**Referee:** *- line 338: 'Because of the similar rainfall seasonality...' seems to be the start of a new paragraph where now Italian sites are being compared to US ones.*

**Response**: That is correct. We moved this part as the starting point of a new paragraph.

**Referee:** *- lines 341-343: No need to repeat the pH and dissolution rates calculated for the Italian sites here if it is already mentioned in the previous paragraph which used to be the text of S2.*

**Response**: Corrected, thank you.

**Referee:** *- lines 346-347: 'the achieved order of magnitude of weathering rate reflects the values presented in the mesocosm experiment of Amann et al. (2020), which present a condition very similar to the field.' What exactly are the weathering rates presented in Amann et al. (2020)? How do the conditions of their mesocosm experiment compare to those of the models discussed in this paper? What ´olivine´ type used, application rates, which crop in the mesocosm, irrigation scheme, soil type and composition? A comparison of the results of the current study with those of a published paper benefits from some info on the published study.*

**Response**: By using a loamy sandy soil with pH equal to about 6.6, which is similar to the annual average soil pH for Iowa, and a total amount of annual rain of 800 mm $y^{-1}$, Amann et al., (2020) achieved weathering rates of $10^{-13.12}$ and $10^{-13.75}$ mol $m^{-2}$ $s^{-1}$ for coarse and fine dunite, respectively. These rates are a bit lower than those achieved by our study (the annual average weathering rate for Iowa is 2.13 x $10^{-12}$ mol $m^{-2}$ $s^{-1}$). This may be due to the fact that EW dynamics can depend on many other factors, such as CEC and the seasonality of rainfall that affects the soil moisture signal.
Given these considerations, we modified this part of the manuscript specifying that our dissolution rates are more typical of a field environment rather than those obtained in laboratory conditions, also stating some characteristics of the experiment presented in Amann et al., (2020), such as soil type, pH and average annual rainfall:

*The order of magnitude of weathering rates provided by our model is more typical of the field environment with respect to those achieved in laboratory conditions. Indeed, we achieved weathering rate values similar to those presented in the mesocosm experiment of Amann et al., (2020), which used a loamy sandy soil with pH equal to about 6.6 and a total amount of annual rain of 800 mm $y^{-1}$, similar to the annual average soil pH for Iowa.*

**Referee:** *- lines 348-349: ´suitable calibration´ seems odd in this sentence, perhaps rewrite as "the importance of site representative model input data for the background flux, ..."*

*Rainfall seasonality, irrigation scheme, CEC, soil type, main soil properties and crop phenology have indeed been determined as representative as possible for the four respective sites. In comparison, the soil mineralogy, another very important parameter influencing the olivine dissolution dynamics, chosen for the models is much less site specific or realistic*

**Response**: We rephrased the sentence in lines 348-349 in the following way:

*This aspect stresses the importance of using measurements of soil properties (e.g., CEC, pH) for calibrating the background weathering flux, allowing to obtain more realistic estimates of olivine dissolution dynamics.*

Regarding the second point, as expressed before, we are aware that mineralogical information of soils is less realistic with respect to the other factors, but the calibration of the background weathering flux based on soil pH data should indirectly resolve this aspect, as described in the response to the comment of line 165.

**Referee:** *- lines 352-353: "The overall rather low monthly values of sequestered CO2 for all case studiesare due to the generally low leaching rate, which reflects the low MAP values for all considered sites."*

**Response**: Sentence rephrased as suggested.

**Referee:** *- line 354: "The annual average sequestered CO2 equals 0.62 kg/ha for Sicily, ..."*

**Response**: Sentence rephrased as suggested.

**Referee:** *- lines 355-358: The difference between the Amann et al. (2020) CO2 sequestration values and the ones of this study are on a scale of 1 to 2 orders of magnitude – yet considered comparable to one another. The weathering ratio values obtained for the US and Italian sites differ only 1 order of magnitude from one another – yet deemed different (and this difference explained by the least site representative/realistic model input parameter of soil mineralogy). Please be consistent with interpreting the difference between values. It is true that Amann et al. (2020) added 22kg/m² whereas in these models 10kg/m² was applied, but the rock dust of the former only contains about 90% olivine. How does the soil moisture throughout the year compare between both these studies? How was the CO2 sequestration value calculated in Amann et al. (2020)? What other factors might play a role in the difference of CO2 sequestration rates obtained for these two different study approaches (mesocosm experiment and model)?*

**Response**: Amann et al. (2020) derived an amount of sequestered $CO_2$ higher than the one we estimated for Iowa, that is the site having closer MAP and pH to the experiment condition (i.e., a range of 23 - 49 kg ha$^{-1}$ y$^{-1}$ of the mesocosm experiment against 4.2 kg ha$^{-1}$ y$^{-1}$ obtained by our model). This can be explained by various reasons:

- Amann et al. (2020) applied 22 kg dunite $m^{-2}$ to their mesocosms. The dunite was about 90% olivine of which 92% was forsterite, so they applied 19.8 kg olivine $m^{-2}$ and 18.216 kg forsterite $m^{-2}$, which is almost the double of our application rate.
- The sequestered $CO_2$ in Amann et al., (2020) is calculated on the base of the chemical characteristics of the outlet water, thus on the base of the extra dissolved $Mg^{2+}$ and DIC (Dissolved Inorganic Carbon), while we refer to the leached extra bicarbonates and carbonates produced by olivine dissolution.
- Furthermore, the experiment considers two precipitation regimes, with daily and weekly rainfall, delivering the same total annual precipitation volume, thus scheduled irrigation interventions, which certainly do not reproduce the stochasticity in the temporal distribution of precipitation.

We summarized these reasons in Section 3.4 of the manuscript. The added part is the one below:

*Apart from this aspect, the differences in the achieved carbon sequestration rates may be due to the way in which this is computed. Indeed, while Amann et al., (2020) considers the dissolved magnesium produced by olivine dissolution, we refer to the leached of the extra bicarbonates and carbonates after olivine dissolution. Furthermore, the mesocosm experiment considers two different rain regimes, namely a daily and weekly rainfall, delivering the same total annual precipitation volume. These may be considered as scheduled irrigation interventions, which certainly do not reproduce the stochasticity in the temporal distribution of precipitation.*

**Referee:** *- line 362: ... with a ==corresponding== increase of HCO3-...*

**Response**: Corrected.

**4 Discussion and conclusions**

**Referee:** *- line 372: Analyzing the interactions ==between rainfall== and crop properties...*

**Response**: Corrected.

**Referee:** *- line 378: ... with a ==corresponding== increase of HCO3...*

**Response**: Corrected.

**Referee:** *- lines 379-380: "... by olivine reaction with CO2. Higher soil water contents also mean higher leaching rates and hence better transport of the (bi)carbonate anions away from the active olivine dissolution zone."*

**Response**: Sentence rephrased. Thank you.

**Referee:** *- line 395: ... the one we called CO2,sw ... Please shortly define/explain this parameter instead of just giving the symbol and referring to a previous publication.*
*...In effect, even in this our previous work we obtained...*

*Although I understand the reasoning that (bi)carbonates formed by olivine dissolution but which stay in the ´EW zone´ are seen as a risk to recombine to carbonic acid releasing CO2 back to the atmosphere, and that hence (bi)carbonates leached out from EW zone are interpreted as more reliable measure for CO2 sequestration, it is not so straightforward. Some of the olivine dissolution sourced carbonate anions might precipitate in solid carbonate minerals within the soil (calcite) which is then stable carbon sequestration that can not be traced back in the leached groundwater below. On the other hand, (bi)carbonates dissolved in leached groundwater and hence taken into account for CO2 sequestration calculations, might recombine to carbonic acid and degas CO2 when they resurface or mix with water of different composition, temperature,... The permanence of CO2 sequestered as (bi)carbonates in groundwater through olivine dissolution is difficult to estimate and probably varies from one context to the next. Maybe this suggests that (bi)carbonate anions and DIC are not the best parameters to estimate the amount of captured CO2. Another product from olivine dissolution is Mg2+ cations. In general, weathering of silicate rocks will release base cations into the soil water as well as (bi)carbonates. Please see what is written about this in literature and assess the pros and cons of using (bi)carbonates or cations to estimate sequestered carbon. Is there a possibility within your model to obtain values for cations resulting from olivine dissolution, and to use these data for an alternative calculation of CO2 sequestration?*

**Response**: We explained the meaning of the parameter $CO_{2, sw}$ in the manuscript, as reported below:

*...the one that we called $CO_{2, sw}$ in Cipolla et al., (2021b), that represents the amount of extra bi(carbonate) anions dissolved in soil water due to olivine weathering.*

We agree with you in all you affirm about uncertainties in carbon sequestration definition. Scientific literature, indeed, presents various studies that define it in different ways, representative of the fact that there is not a unique way to assess it. Sometimes, $Mg^{2+}$ released from olivine dissolution is used (Amann et al., 2020, https://doi.org/10.5194/bg-17-103-2020; ten Berge et al., 2012, 10.1371/journal.pone.0042098; Renforth et al., 2015, http://dx.doi.org/10.1016/j.apgeochem.2015.05.016), while other approaches use extra DIC in soil water produced by olivine dissolution as a carbon sequestration metric (Beerling et al., 2020, https://doi.org/10.1038/s41586-020-2448-9). We agree with the fact that even leached bi(carbonate) anions may recombine to carbonic acid and release $CO_2$ into the atmosphere, depending on the chemical and mineralogical characteristics of the soils on their way to the oceans, as well as on biogeochemical processes that can occur along their path.

Our model is certainly able to compute the concentration of $Mg^{2+}$ released by olivine dissolution, as you can see from Cipolla et al., (2021b). However, relying only on the dissolved magnesium in soil water may provide an uncertain estimation of carbon sequestration, given that this cation is removed from soil water by many processes. Indeed, cations such as $Ca^{2+}$ and $Mg^{2+}$ are essential macronutrients for living organisms (White et al., 2010, 10.1093/aob/mcq085). Plant and bacteria uptake them when they are available in the water solution and from soil colloid surface exchange sites. For this reason, leached bi(carbonate) anions provide the most reliable carbon sequestration metric, at least with reference to the domain we adopt in the simulations, since they express the carbon that comes from the reaction of olivine with $CO_2$ and is taken away from the considered domain by the leaching process.

**Referee:** *- line 414: Good to come back to possible more complexity in future models regarding the silicate rock dust that can be used for EW.*

**Response**: Thank you. This is a promising challenge for future works.

**Referee:** *- line 415: basalt is NOT a mineral, it is a rock and hence an assemblage of minerals. See comments for lines 22-25. The reason that basalt has lower Ni and Cr contents compared to olivine is because basalt only partially consists of olivine. Since the topic of potential Ni and Cr contamination resulting of EW of olivine rich rocks is touched upon here, please add a sentence explaining that both these heavy metals occur in olivine crystals and are thus released when the latter are dissolved.*

*Another reason to use basalt is that the other minerals it contains release plant nutrient cations upon dissolution, effectively being a natural fertilizer.*

*All in all, using silicate rock powder consisting of different minerals, instead of just one mineral (olivine, wollastonite) would greatly improve the model´s representation of realistic field situations.*

**Response**: We specified that basalt is a rock that may be also used for EW applications. Furthermore, we stated that the risk of heavy metals release occurs upon olivine dissolution since its crystals contain Ni and Cr. One of the future goals of our research is indeed to study the EW capabilities of other silicate minerals or assemblages of them (i.e., basaltic rocks) since, as you correctly stated, it would represent a more realistic situation and this rock is certainly easier to be found, crushed and used as amendment, with respect to single minerals. The modified sentence can be found below:

*Indeed, many EW experiments have been conducted with wollastonite or using basaltic rocks, among various aspects to avoid or simply reduce the high Ni and Cr content potentially released by olivine during dissolution. These heavy metals are present, in fact, in olivine crystals and are then released upon olivine dissolution.*

**Referee:** *- line 417: The wollastonite EW field trials of Haque et al (2020) are NOT across the world but at three different locations in Canada.*

**Response**: Thank you for highlighting the mistake. We corrected it.

**Referee:** *- lines 417-718: Wollastonite is a calcium silicate – CaSiO3 – that upon reaction with water and CO2 dissolves and forms, among other products, Ca2+ and CO3 2-. This cation and anion can combine within the EW zone to form secondary, pedogenic calcite which is then an easy measure to assess how much wollastonite dissolved, and hence CO2 was sequestrated into this new calcite. In case of olivine dissolution, the released cations are less likely to form new carbonate minerals within the EW zone, only under certain chemical conditions they might. This is one of the reasons why CO2 sequestration from olivine and other silicate rock dusts dissolution is more difficult to measure. (see comments line 395).*

**Response**: We totally agree with you in the sense that carbon sequestration from olivine is more uncertain to quantify with respect to wollastonite. You can refer to our response comments of line 395 for some considerations about carbon sequestration assessment.

**Referee:** *- line 421: Indeed, most lab, pot and mesocosm experiments are carried out under continuous (near) saturation of soil moisture which is not representative of the real life situation. The*

*detailed incorporation of rainfall seasonality and irrigation in the here presented model is therefore one if its greatest merits and strengths towards more realistic EW potential predictions.*

**Response**: Thank you for appreciating and valuing this aspect of our work. Whether the seasonality of rainfall and, in turn, soil moisture is important or not in this process is one of the most debated aspects in scientific literature. We strongly believe, as you affirmed, that considering near saturation conditions with stationary water fluxes is a lot far away from what happens in reality. As we demonstrated with our results, in fact, hydroclimatic fluctuations are one of the most important aspects to predict EW dynamics.

**Referee:** *- line 424-426: Precipitation of secondary minerals as pedogenic carbonates from products of silicate rock powder dissolution in the field is far from well understood and likely not the most common scenario. A more relevant improvement of the here presented excellent EW model would therefore be to go from single mineral olivine (which in reality is never applied as it is not available) to a realistic assemblage of minerals (for example resembling that of a mantle dunite, or a basalt) that takes into account the dissolution rates of the individual minerals and their relative presence in the silicate rock dust.*

**Response**: Once again we thank you for appreciating our work and providing very useful suggestions. We added this important aspect to our discussions, modifying the sentence as follows:

*Therefore, a possible development of this work may consist of a comparison of EW yields under the amendment of different assemblages of silicate minerals (i.e., basaltic rocks) in various areas of the world, taking into account the dissolution rates of the individual minerals and their relative presence in the silicate rock dust, thus providing a more reliable prediction of EW dynamics*

**Figures**

**Referee:** *Figure 3: To allow easier comparison between the average rainfall depth α and rainfall frequency λ (please label both fully on the vertical axes) between the 4 different areas, maintain the same scale for all four diagrams (i.e. λ up to 0.47 and α up to 13.5). Putting the location names in each of the 4 plots would also make it easier to interpret this figure at a glance. In a black and white print out it is not clear which of the two lines is which, so perhaps make one a dotted line and either put in a small legend, or describe in the figure caption which parameter is represented by the full, and which one by the dotted, line. Also fully describe what the α and λ "rainfall parameters" exactly represent.*

**Response**: We agree with you in the sense that the same scale is needed for both the rainfall parameters to easily compare them across the four selected places. For a better comprehension of the figure, we decided to plot the average rainfall depth (α) and frequency (λ) of the four sites under study in a single plot, showing them with different line styles, in order to have a clear plot even in a black and white print out. We also modified the caption, clarifying the meaning of the two rainfall parameters:

*Values of the average rainfall depth (α) and frequency (λ) from January (i.e., month indicated with 1) to December (i.e., month indicated with 12). The reader is referred to Section 2.2.1 for more details.*

You can find the modified figure in the following:

[Figure]

**Referee:** *Figure 4: Please write full name and symbol for both parameters (crop coefficient Kc and added carbon ADD) in both figure caption and alongside the vertical axes. To interpret more easily the graph, perhaps put a) b) c) vertically below one another in the first column, writing ´wheat´ above it and the site name in each of the 3 graphs. And then have d) and e) vertically below one another in the second column, writing ´corn´ above this column and the names of the sites in the respective graphs.*

**Response**: Following your suggestions, we modified the figure:

[Figure]

Furthermore, we modified the caption like this:

*Seasonal variability of the crop coefficient (Kc) and the added carbon from vegetation (ADD) for a) wheat in Sicily, b) wheat in California, c) wheat in the Padan plain and Iowa, d) corn in Padan plain and Iowa and e) corn in Sicily and California. The reader is referred to Section 2.2.3 for more details.*

**Referee:** *Figure 5: Please add the respective symbols for the different variables to the figure caption: soil moisture (S), weathering rate (Wr), Iowa (IA), California (CA), corn in clay loam soil (C-CL), etc. To make this figure easier to read it would be good to have the colour legend just once, write the full parameter name above each column, and the full crop/soil type combination in front of every row.*

**Response**: The figure is now clearer to read. This is the modified version:

[Figure]

**Referee:** *Figure 6: Please add the symbol to the parameter name in the figure caption, and the full nameto the symbol along the Y-axis of the specific graph (crop coefficient Kc, soil moisture S). Letters a), b). c) and d) are missing in the respective panels.*

**Response**: Done! The modified version of the figure is shown below:

[Figure]

**Referee:** *Figure S1: Please make a main manuscript figure occurring after the time-series heat maps for the US sites, and make the same corrections/changes as detailed above for Figure 5. The resolution of the current S1 figure needs to be improved as the Y-axis labels are poorly readable both in print and in the pdf on screen.*

**Response**: We made the same corrections suggested for Figure 5. Now this figure is in the main manuscript after that related to the US case studies. The new version is the following:

[Figure]

**Referee:** *Figure 7: Please adjust in the same way as suggested for Figure 5.*

**Response**: Same corrections as in Figure 5:

[Figure]

**Referee:** *Figure 8: Please adjust in the same way as suggested for Figure 5.*

**Response**: Same corrections as in Figure 5:

[Figure]

**Referee:** *Figure 9: Please add the symbol to the parameter name in the figure caption, and the full name to the symbol along the Y-axis of the specific graph (bare soil evaporation (E), crop transpiration (T)). To allow easier comparison of these parameters between a) wheat sand b) corn, please have both Y-axis the same length (4.25 mm).*

**Response**: Figure and caption have been edited as suggested:

[Figure]

**Referee:** *Figure 10: As before, please write the parameter codes in the figure caption and the full parameter name along with its code above the respective column of heat maps. (soil moisture (S), weathering rate (Wr)). To the left of each of the rows, add the codes of the case studies, and in the figure caption write the case study code with the full description (wheat in clay loam soil in California (W-CL-CA), corn in silty clay loam soil in Iowa (C-SCL-IA).*

**Response**: Figure has been edited as suggested by this and the following comment:

[Figure]

**Referee:** *Figure S2: Include these two rows of each 3 heat maps for the Italian sites in Figure 10, with both full reference and code (wheat in clay loam soil in Sicily (W-CL-SI), corn in silty clay loam in Padan plain (C-SCL-PP)).*

**Response**: This figure does not exist anymore. All its contents have been reported to Figure 10 in the main manuscript, following the Reviewer's suggestions.

**Referee:** *Figure 11: Please put the site name in each of the plots to make this figure easier to read. Do these plots reflect the model outcomes from the input parameters used in section 3.4 (last 4 rows in table 1)? Please write again in the figure caption which soil type and crop, with or without irrigation, is presented here for each of the 4 sites.*

**Response**: The boxplots of sequestered $CO_2$ are related to single runs of the four case studies, considering the single climatic conditions, along with the most frequent crop and soil type. Basically, these simulations are those referred to as "Most frequent scenario (Section 3.4)" shown in table 1. We provided to better clarify this aspect at the beginning of Section 3.4 with this sentence:

*The results here analyzed are those related to the simulations referred to as "Most frequent scenario (Section 3.4)" as shown in table 1, where the considered climatic condition is analyzed along with the most frequent crop and soil type.*

We also clarified this aspect in the caption:

*Box plots representing the seasonality of the $CO_2$ sequestered by leaching of extra $HCO_3^-$ and $CO_3^{2-}$ produced by olivine dissolution, computed over the 10 years subsequent to olivine amendment. The*

*plots are related to a) Wheat in clay loam soil for Sicily (W - CL - SI), b) Corn in silty clay loam soil for Padan plain (C - SCL - PP), c) Wheat in clay loam soil for California (W - CL - CA) and d) Corn in silty clay loam soil for Iowa (C - SCL – IA).*

Finally, the place names to the panels were added as suggested. The new figure is the following:

[Figure]

**Tables**

**Referee:** *Table 2: Please have the same number of digits after the separation point for values of the same parameter (they are different for soil moisture at field capacity and saturation hydraulic conductivity). Please also write which of the four study sites are represented by which soil type.*

**Response**: Done. The new version of the table and the related caption can be found below:

*Table 2 - Properties of the clay loam (Sicily and California) and silty clay loam (Padan plain and Iowa) soils used in the model.*

|  |  | Clay loam (SI and CA) | Silty clay loam (PP and IA) |
|---|---|---|---|
| Soil porosity | n | 0.476 | 0.477 |
| Soil moisture at the hygroscopic point | $s_h$ | 0.394 | 0.319 |
| Soil moisture at wilting point | $s_w$ | 0.453 | 0.373 |
| Soil moisture at incipient stress | $s^*$ | 0.64 | 0.56 |
| Soil moisture at field capacity | $s_{fc}$ | 0.821 | 0.750 |
| Saturation hydraulic conductivity | Ks | 0.212 | 0.147 |
| Pore size distribution index | b | 8.52 | 7.75 |
| Bulk density of soil | $\rho_b$ | 1450 | 1500 |

**Referee:** *Table 3: According to the text (lines 153-155), biomass pool Cb is defined as 1% of the above defined carbon input (C0). Yet in the table the values for Cb are 10 times higher than those for C0? Maybe Cb is here expressed as g/m³ instead of kg/m³ as C0 above? Please also add to this table the model input values for the different study sites of CEC, derived soil mineralogy, soil pH and calculated dissolution rate constants.*
*Perhaps it is possible to combine tables 2 and 3 into one table with all soil type and composition data used in the models?*

**Response**: We corrected the unit of measurement of $C_b$ since it is expressed as g/m$^3$. We also extended table 3 adding the typical pH, CEC, and the achieved dissolution rate constants for the soils under study, which better represent the background weathering component. We prefer to keep separated tables 2 and 3 since the latter presents some parameters that depend on the site under study, rather than only on the soil type (i.e., pH differ from Padan plain and Iowa despite the most frequent soil type is the same). The table in the new form is reported in the following:

*Table 3 - Initial organic carbon content in the litter and humus pools ($C_0$) and the biomass pool ($C_b$) for the four sites under study. Typical values of pH, CEC and background dissolution rate constants are also reported.*

|  | Sicily | Padan plain | California | Iowa |
|---|---|---|---|---|
| $C_0 [kg\ m^{-3}]$ | 13.19 | 18.19 | 19.74 | 11.58 |
| $C_b [g\ m^{-3}]$ | 131.9 | 181.9 | 197.4 | 115.8 |
| pH [-] | 7.2-8.8 | 7.2-8.8 | 7 | 6 |
| CEC [cmol $kg^{-1}$] | 50 | 50 | 35 | 25 |
| $k_{bg}$ [mol $m^{-2} s^{-1}$] | $10^{-5}$ | $10^{-5}$ | $10^{-6}$ | $10^{-8}$ |

**References**

**Referee:** *All references in text are in the references list and vice versa. The only irregularity is the first entry in the references list ´Critical Review …, 1979´ which seems to lack authors but might coincide with the incomplete reference in the text in line 170 ´44 (1979)´.*

**Response**: We corrected it adding the authors to that reference.

**Supplementary material**

**Referee:** *Please include this in the main manuscript text as suggested in comments above*

**Response**: All elements of the supplementary material have been included in the main text.

---

## Referee Report (RR1)

**Review of revised manuscript egusphere-2022-196**

Most of my comments have been addressed, but there are still a few points that need some attention:

- The other reviewer mentioned that some explanation of how to read heatmaps would be useful and I certainly found this to be the case when I looked at them again; it took me a moment to remember that the Y axis represented simulation year. Day of year is on the X axis which seems obvious to me, but it is still worth mentioning.  All these figures have colour bars so an explanation of what the colours mean was less important, at least for me. In any case please say what the axes tickmark values represent, either in the text or (preferably) in the caption of the first heatmap figure, e.g.:
  - Figure 6. Time-series heatmaps (X axes: day of year, Y axes: simulation year) ...
- Please provide a reference for the laboratory rates being considered in the following revised text. One good possibility would be Bandstra et al. 2008 (in Brantley, Kubicki and White *Kinetics of Mineral Dissolution*):

  > "The order of magnitude of weathering rates provided by our model is more typical of the field environment with respect to those achieved in laboratory conditions (e.g., Bandstra et al. 2008)."

- The immediately-following revised text stating that the modelled rates are "similar" to those of Amann et al. (2020) is misleading because the modelled weathering rates are an order of magnitude faster than those of Amann et al.  Your rates  are a lot more similar to those of Renforth et al. (2015, you already cite them), although their study used soil without plants. It is best to state that both the modelling and mesocosm studies achieved rates that are slower than those observed in the laboratory, and then give all the values for readers to see for themselves, e.g.

  > "This was also the case for the mesocosm studies of Renforth et al. (2015) and Amann et al. (2020). Our rates (10^-11.67 Iowa, 10^-11.79 California, 10^-12.50 Padan and 10^-12.32 Sicily, mol Olivine/m2/s) are comparable to theirs:  10^-12.7 to 10^-11.8 mol Olivine/m2/s (Renforth et al 2015), 10^-13.12 and 10^-13.75 mol Olivine/m2/s (Amann et al. 2020)."

Note that the spread of dissolution rates observed at similar pH can approach or exceed an order of magnitude even at the same experimental scale, as can be seen in the laboratory data shown for olivine by Bandstra et al. (figure on page 809 in the Appendix of Brantley, Kubicki and White 2008).

---

## Referee Report (RR2)

**Review of REVISED Cipolla et al EGUsphere-2022-196**

by Ingrid Smet (ingrid.smet@fieldcode.com)

**General comments**

The authors carefully considered each and every one of my (many) review comments, thereby accepting nearly all suggested changes, answering my scientific questions whilst also clarifying these topics in the new manuscript where appropriate, and providing sound reasons for those suggested changes they did not implement.

Whereas the first manuscript was already an interesting and important publication, the revised manuscript is better structured, more easily understandable and more complete, with extra information and better readable figures.

As I have no background in computational modelling, I greatly appreciate the extra information the authors provided me in their responses and which allowed me to better understand the model, especially the mineralogy part (both in soil and EW material) which is my field of expertise.

In particular I had not quite grasped that the background weathering flux for the 4 locations was not just based on the main mineral of their bedrock (quartz or calcite) but that actual values of soil CEC were also considered as well as a calibration with soil pH. Therefore, my previous worry that the background weathering fluxes might be a lot less realistic than the other model input data of climate and vegetation, turned out to be misplaced.

I am excited to read that the authors appreciate my suggestion for further development of their model to implement multi-mineral EW materials, which are more realistic. Although I understand that modelling EW with a rock dust consisting of 100% Mg-endmember olivine allows for other parameters to vary – and their influence on EW efficiency to be assessed – in the real world there is no 100% olivine rock consisting exclusively of $Mg_2SiO_4$ to be found. Some thoughts:

- Only in case of rather pristine mantle rocks (dunite, peridotite) there might be up to 90-95% olivine with minor pyroxene, Fe-Ti-oxides, … as other mineral phases. For these rock types one could indeed just take into account the olivine fraction of the rock for calcuations.

- In case of altered mantle material, that is more commonly found, olivine percentages can be 30-70% and in this case it becomes interesting to also take into account the other main mineral phases that dissolve within relevant time frames.

- Finally, a rock such as basalt contains up to ca. 10% of olivine, so the other main mineral phases now really need to be taken into account. It furthermore also consists of volcanic glass, another quickly weathering material (no mineral as no crystal structure) which would be interesting to include in the calculations.

*Specific comments*

I did not have the time to read through the revised manuscript document in as much detail as I did with the authors' responses document, but when I read through it I observed the following potential typos/missing words (**bold** is suggested change):

- line 20: …such as **Ca-Mg** silicates…

- line 27: …can be **found in** igneous rocks… and mostly **in**… (extracting olivine from rocks is not done except for large, high quality olivine crystals then used as gemstone 'peridote')

- line 44: … conditions **whilst still being** a closed and controlled system… (this paragraph I wrote in my initial review was a comment to explain these different EW experiments, not a grammarly complety correct text to include as is in the manuscript)

- line 61: … in laboratory **experiments**, such…

- line 92: … when required **based** on crop…

- line 128: … to be mixed **throughout** the…

- line 162: … phenology **and despite** the modest MAP differences… (I think this is what is meant here?)

- Figure 3: In the main text and in the figure caption the 'average rainfall depth ($\alpha$) and frequency ($\lambda$)' are mentioned, but the Y-axis in the figure itselfs mentions 'average storm depth ($\alpha$) and average storm frequency ($\lambda$)'

- lines 252-253: … the period**s** related to… and late **season**,respectively)…

- line 257: … **derived** from plant metabolic…

- lines 260-270: two different ways of writing the dates, with and without a comma (for July**,** 6th for corn - for July 6th for corn) are used intermittently within this paragraph

- line 312: … of multiple irrigation **events** are…

- lines 360-362: …  Apart from some spikes, occurring on some specific days, averaging the grain scale weathering rate ratio, we achieved that the clay loam soil results in a weathering rate about twice as high as what obtained with the silty clay loam soil, at all four locations….

This sentence is very strange to me, perhaps the word 'achieved' is not correct here, in any case I do not understand the structure of the first part of the sentence. Please rephrase to make more clear.

- line 368: … acidifying effect resulting from the displacement of…

- line 414: … two Italian sites, Sicily and Padan plain, which…

- line 444: … that has the slowest background weathering flux…

- lines 451-452: …more than double the olivine we added in our study (i.e., 22 kgm−2 of dunite corresponding to about 18 kg m−2 of olivine)

Although should it not be 'almost double' the olivine, since this study added 10kg/m² and you calculate that given its olivine contents the dunite represented about 18kg/m²?

- line 455: … the leached concentrations of the…

- lines 461-462: … A great amount of rainfall, contextually occurring to low transpiration losses, leads… Not sure what is meant here? Contextually occurring together with low transpiration losses? Contextually leading to low transpiration losses? Please rephrase to clarify.

- lines 485-488:  "Taking into account the case of Iowa, which resulted in the highest carbon sequestration rate and is characterized by a cropland area covered by the corn of about 56,000 km2, the annual average sequestered CO2 could reach the value of about 0.023 Mt y−1, if the whole cropland area were amended with olivine. Sicily, instead, may sequester on average a mass of 0.0002 Mt y−1, if amending the total cropland area cultivated with wheat of about 265,000 ha.''

Maybe it is good to repeat here that these annual sequestration rates are calculated based on a once in 10 year 100ton/ha application of 100% pure Mg2SiO4 – for those readers that did not go through the entire paper but from abstract to final discussion and conclusions.

Since there are so many 0s in the sequestered CO2 when expressing it as Mton/year, it might be easier on the eye (and to understand the amount) when expressed in kilo tons? Or even as 23,000 tons and 200 tons?

- line 498: … In effect, **even in**  **our** previous work we obtained about…

- line 512: … products (e.g., **$NO_3^-$** ), …     (superscript for '-')

- lines 524-525: Haque et al. (2020) carried out a wollastonite EW experiment on three farms with different plants, located in three **separate sites**  in Canada.

- line 527: Kelland et al. (2020) (i.e., 2-4 $tCO_2ha^{-1}$) … please also mention in what time frame, I presume per year? Then it can become

'greater **annual** carbon sequestration rates (…) **compared** to…'

- line 537: Another relevant aspect to consider when planning an EW intervention is **the** economic feasibility **on** itself

- line 544: … biochar, leading to **consideration of** EW as a reasonable…

- line 547: … sequestration potential **and minimal** related costs…

Regarding these lasts comments of the need to combine EW potential with minimizing costs for each location specifically, this can be addressed in model calculations by using multi-mineral compositions reflecting real rocks potentially suitable for EW that are found near those locations. For Europe there is a paper of Kremer et al, 2019 (https://www.mdpi.com/2075-163X/9/8/485) identifying natural rocks suitable for EW.

---

## Author Response (AR2)

Replies to the comments on "**Effects of precipitation seasonality, irrigation, vegetation cycle and soil type on enhanced weathering – Modelling of cropland case studies across four sites**".

G. Cipolla, S. Calabrese, A. Porporato, L.V. Noto

**Review of revised manuscript No: egusphere-2022-196**
* * *
**General comments:**

**Referee:** *Most of my comments have been addressed, but there are still a few points that need some attention:*

- *The other reviewer mentioned that some explanation of how to read heatmaps would be useful and I certainly found this to be the case when I looked at them again; it took me a moment to remember that the Y axis represented simulation year. Day of year is on the X axis which seems obvious to me, but it is still worth mentioning. All these figures have colour bars so an explanation of what the colours mean was less important, at least for me. In any case please say what the axes tickmark values represent, either in the text or (preferably) in the caption of the first heatmap figure, e.g.: Figure 6. Time-series heatmaps (X axes: day of year, Y axes: simulation year) ...*

**Response**: Good suggestion! We specified what the axes tickmark values represent in the caption of Figure 6.

- *Please provide a reference for the laboratory rates being considered in the following revised text. One good possibility would be Bandstra et al. 2008 (in Brantley, Kubicki and White Kinetics of Mineral Dissolution): "The order of magnitude of weathering rates provided by our model is more typical of the field environment with respect to those achieved in laboratory conditions (e.g., Bandstra et al. 2008)."*

**Response**: Thank you for the insight. Done.

- *The immediately-following revised text stating that the modelled rates are "similar" to those of Amann et al. (2020) is misleading because the modelled weathering rates are an order of magnitude faster than those of Amann et al. Your rates are a lot more similar to those of Renforth et al. (2015, you already cite them), although their study used soil without plants. It is best to state that both the modelling and mesocosm studies achieved rates that are slower than those observed in the laboratory, and then give all the values for readers to see for themselves, e.g. "This was also the case for the mesocosm studies of Renforth et al. (2015) and Amann et al. (2020). Our rates ($10^{-11.67}$ Iowa, $10^{-11.79}$ California, $10^{-12.50}$ Padan and $10^{-12.32}$ Sicily, mol Olivine/m2/s) are comparable to theirs: $10^{-12.7}$ to $10^{-11.8}$ mol Olivine/m2/s (Renforth et al 2015), $10^{-13.12}$ and $10^{-13.75}$ mol Olivine/m2/s (Amann et al. 2020)."*
*Note that the spread of dissolution rates observed at similar pH can approach or exceed an order of magnitude even at the same experimental scale, as can be seen in the laboratory data shown*

*for olivine by Bandstra et al. (figure on page 809 in the Appendix of Brantley, Kubicki and White 2008).*

**Response**: This is a very good point. We rephrased the sentence as suggested.
* * *
Replies to the comments on "**Effects of precipitation seasonality, irrigation, vegetation cycle and soil type on enhanced weathering – Modelling of cropland case studies across four sites**".

G. Cipolla, S. Calabrese, A. Porporato, L.V. Noto

**Review of revised manuscript No: egusphere-2022-196**
* * *
**General comments:**

**Referee:** *The authors carefully considered each and every one of my (many) review comments, thereby accepting nearly all suggested changes, answering my scientific questions whilst also clarifying these topics in the new manuscript where appropriate, and providing sound reasons for those suggested changes they did not implement.*
*Whereas the first manuscript was already an interesting and important publication, the revised manuscript is better structured, more easily understandable and more complete, with extra information and better readable figures.*
*As I have no background in computational modelling, I greatly appreciate the extra information the authors provided me in their responses and which allowed me to better understand the model, especially the mineralogy part (both in soil and EW material) which is my field of expertise.*
*In particular I had not quite grasped that the background weathering flux for the 4 locations was not just based on the main mineral of their bedrock (quartz or calcite) but that actual values of soil CEC were also considered as well as a calibration with soil pH. Therefore, my previous worry that the background weathering fluxes might be a lot less realistic than the other model input data of climate and vegetation, turned out to be misplaced.*
*I am excited to read that the authors appreciate my suggestion for further development of their model to implement multi-mineral EW materials, which are more realistic. Although I understand that modelling EW with a rock dust consisting of 100% Mg-endmember olivine allows for other parameters to vary – and their influence on EW efficiency to be assessed – in the real world there is no 100% olivine rock consisting exclusively of Mg2SiO4 to be found. Some thoughts:*

- *Only in case of rather pristine mantle rocks (dunite, peridotite) there might be up to 90-95% olivine with minor pyroxene, Fe-Ti-oxides, ... as other mineral phases. For these rock types one could indeed just take into account the olivine fraction of the rock for calcuations.*
- *In case of altered mantle material, that is more commonly found, olivine percentages can be 30-70% and in this case it becomes interesting to also take into account the other main mineral phases that dissolve within relevant time frames.*
- *Finally, a rock such as basalt contains up to ca. 10% of olivine, so the other main mineral phases now really need to be taken into account. It furthermore also consists of volcanic glass, another quickly weathering material (no mineral as no crystal structure) which would be interesting to include in the calculations.*

**Response**: We thank the Reviewer for appreciating our work and for defining it *better structured, more easily understandable and more complete* after our responses to the very worthwhile comments. We also appreciate the thoughts about the use of multi-mineral EW materials; we will surely take them into account in a future application of the model considering this more realistic condition.
* * *
**Specific comments:**

**Referee:** *I did not have the time to read through the revised manuscript document in as much detail as I did with the authors' responses document, but when I read through it I observed the following potential typos/missing words (bold is suggested change):*

- *line 20: …such as **Ca-Mg** silicates…*
- *line 27: …can be **found in** igneous rocks… and mostly in… (extracting olivine from rocks is not done except for large, high quality olivine crystals then used as gemstone 'peridote')*
- *line 44: … conditions **whilst still being** a closed and controlled system… (this paragraph I wrote in my initial review was a comment to explain these different EW experiments, not a grammarly comply correct text to include as is in the manuscript)*
- *line 61: … in laboratory **experiments**, such…*
- *line 92: … when required **based** on crop…*
- *line 128: … to be mixed **throughout** the…*
- *line 162: … phenology **and despite** the modest MAP differences… (I think this is what is meant here?)*
- *Figure 3: In the main text and in the figure caption the 'average rainfall depth (α) and frequency (λ)' are mentioned, but the Y-axis in the figure itselfs mentions 'average storm depth (α) and average storm frequency (λ)'*
- *360lines 252-253: … the periods related to… and late **season**, respectively)…*
- *line 257: … **derived** from plant metabolic…*
- *lines 260-270: two different ways of writing the dates, with and without a comma (for July, 6$^{th}$ for corn - for July 6th for corn) are used intermittently within this paragraph*
- *line 312: … of multiple irrigation **events** are…*
- *lines 360-362: … "Apart from some spikes, occurring on some specific days, averaging the grain scale weathering rate ratio, we achieved that the clay loam soil results in a weathering rate about twice as high as what obtained with the silty clay loam soil, at all four locations"…. This sentence is very strange to me, perhaps the word 'achieved' is not correct here, in any case I do not understand the structure of the first part of the sentence. Please rephrase to make more clear.*
- *line 368: … acidifying effect resulting from the displacement of…*
- *line 414: … two Italian sites, Sicily and Padan plain, which…*
- *line 444: … that has the slowest background weathering flux…*
- *lines 451-452: …"more than double the olivine we added in our study (i.e., 22 kgm−2 of dunite corresponding to about 18 kg m−2 of olivine)" Although should it not be 'almost double' the olivine, since this study added 10kg/m² and you calculate that given its olivine contents the dunite represented about 18kg/m²?*
- *line 455: … the leached concentrations of the…*
- *lines 461-462: … A great amount of rainfall, contextually occurring to low transpiration losses, leads… Not sure what is meant here? Contextually occurring together with low transpiration losses? Contextually leading to low transpiration losses? Please rephrase to clarify.*
- *lines 485-488: "Taking into account the case of Iowa, which resulted in the highest carbon sequestration rate and is characterized by a cropland area covered by the corn of about 56,000 km2, the annual average sequestered CO2 could reach the value of about 0.023 Mt y−1, if the whole cropland area were amended with olivine. Sicily, instead, may sequester on average a mass of 0.0002 Mt y−1, if amending the total cropland area cultivated with wheat of about 265,000 ha.''  Maybe it is good to repeat here that these annual sequestration rates are calculated based on a once in 10 year 100ton/ha application of 100% pure Mg2SiO4 – for those readers that did not go through the entire paper but from abstract to final discussion and conclusions. Since there are so many 0s in the sequestered CO2 when expressing it as*

*Mton/year, it might be easier on the eye (and to understand the amount) when expressed in kilo tons? Or even as 23,000 tons and 200 tons?*

- *line 498: … In effect, **even in**  **our** previous work we obtained about…*
- *line 512: … products (e.g., **NO3 – ), …** (superscript for '-')*
- *lines 524-525: Haque et al. (2020) carried out a wollastonite EW experiment on three farms with different plants, located in three **separate sites** in Canada.*
- *line 527: Kelland et al. (2020) (i.e., 2-4 tCO2ha−1) … please also mention in what time frame, I presume per year? Then it can become 'greater **annual** carbon sequestration rates (…) **compared** to…'*
- *line 537: Another relevant aspect to consider when planning an EW intervention is **the** economic feasibility **on** itself*
- *line 544: … biochar, leading to **consideration of** EW as a reasonable…*
- *line 547: … sequestration potential **and minimal** related costs…*

*Regarding these lasts comments of the need to combine EW potential with minimizing costs for each location specifically, this can be addressed in model calculations by using multi-mineral compositions reflecting real rocks potentially suitable for EW that are found near those locations. For Europe there is a paper of Kremer et al, 2019 (https://www.mdpi.com/2075-163X/9/8/485) identifying natural rocks suitable for EW.*

**Response**: Once again, we appreciate the very in-depth analysis conducted by the Reviewer on our work. We corrected the above-highlighted typos/missing words and rephrased some sentences that were unclear. We also corrected the axis of Figure 3 as suggested.

In particular, the sentence related to lines 360-362 was rewritten as:

*Averaging the grain scale weathering rate ratio (i.e., the curves shown in the last row of the figure), the clay loam soil results in a weathering rate about twice as high as what is obtained with the silty clay loam soil, at all four locations.*

In the sentence in lines 485-488 we better specified that carbon sequestration rates were *calculated as the average over 10 years of 100% pure $Mg_2SiO_4$ amendment with a rate of 100 t $ha^{-2}$*. We also expressed carbon sequestration rates in kilo tons.